# Gliovascular transcriptional perturbations in Alzheimer's disease reveal molecular mechanisms of blood brain barrier dysfunction

To uncover molecular changes underlying blood-brain-barrier dysfunction in Alzheimer's disease, we performed single nucleus RNA sequencing in 24 Alzheimer's disease and control brains and focused on vascular and astrocyte clusters as main cell types of blood-brain-barrier gliovascular-unit. The majority of the vascular transcriptional changes were in pericytes. Of the vascular molecular targets predicted to interact with astrocytic ligands, *SMAD3*, upregulated in Alzheimer's disease pericytes, has the highest number of ligands including *VEGFA*, downregulated in Alzheimer's disease astrocytes. We validated these findings with external datasets comprising 4,730 pericyte and 150,664 astrocyte nuclei. Blood *SMAD3* levels are associated with Alzheimer's disease-related neuroimaging outcomes. We determined inverse relationships between pericytic SMAD3 and astrocytic VEGFA in human iPSC and zebrafish models. Here, we detect vast transcriptome changes in Alzheimer's disease at the gliovascular-unit, prioritize perturbed pericytic *SMAD3*-astrocytic *VEGFA* interactions, and validate these in cross-species models to provide a molecular mechanism of blood-brain-barrier disintegrity in Alzheimer's disease.

Impairment of the blood-brain-barrier (BBB) is a key feature in Alzheimer's disease (AD), which is thought to lead to entry of neurotoxic substances from blood to the brain resulting in inflammatory response and reduced cerebral blood flow[1]. Accumulation of amyloid β (Aβ) deposits around cerebral vasculature is thought to be both a cause and consequence of BBB impairment[2,3], which in turn is an early biomarker of cognitive dysfunction[4], can predict cognitive decline[5], and contributes to AD pathogenesis and progression[2,6]. However, precise transcriptional changes in the gliovascular unit (GVU)[7–9] of the BBB in AD and molecular interactions between the main GVU cell types, namely brain vascular cells and astrocytes remain to be established at systems level.

Single cell RNA sequencing (RNAseq) enables researchers to obtain transcriptomes of individual intact cells (scRNAseq) or nuclei (snRNAseq)[10,11]. This approach has been utilized to profile cell types (and subtypes) in AD and healthy brains, identify cellular states and cell activation, describe vulnerable cell populations, and elucidate perturbed genes and pathways in specific cell types in AD[12–23].

To date, most single-cell transcriptomic studies of AD brains focused on neuronal cells and more abundant glial cells. Relatively little is known about transcriptional changes in vascular cells, namely endothelia and pericytes, and their interaction with other central nervous system (CNS) cells in the GVU[1,24]. Recent snRNAseq studies began to reveal transcriptional profiles of human cerebrovasculature[18,21,22,25] and detected differentially expressed genes (DEGs) in AD either in enriched vascular[22] or un-enriched nuclei[18,21]. Despite these advances, studies that systematically interrogate and prioritize transcriptional perturbations in the GVU, followed by

✉e-mail: taner.nilufer@mayo.edu

experimental validations of interacting GVU molecules and their effects on the BBB are necessary to identify high-confidence therapeutic target or biomarker candidates of BBB dysfunction.

In this study, we apply a systematic approach to detect, prioritize, validate and replicate GVU transcriptional perturbations in post-mortem AD brains, test the top perturbed vascular transcript, *SMAD3*, for its associations with AD-related antemortem outcomes, perform in vitro validations of *SMAD3* interactions with its predicted astrocytic molecular partner *VEGFA* in iPSC-derived pericytes and conduct in vivo experimental validations of SMAD3-VEGFA interactions and their consequences on BBB integrity in a well-established zebrafish model[26–28]. Our findings provide information on brain vascular expression changes at a single nucleus level in AD, uncover vascular-astrocytic interactions in the GVU, provide cross-species experimental validations for pericytic *SMAD3*-astrocytic *VEGFA* perturbations as a mechanism that may contribute to BBB disintegrity in AD.

## Results

### SnRNAseq brain transcriptome profiling

In 12 donors with neuropathologic AD and 12 age- and sex-matched controls (Fig. 1a, Supplementary Fig. 1) we obtained snRNAseq profiles from temporal cortex tissue (TCX) using 10x Genomics platform, which yielded 87,493 single nuclei transcriptomes (Fig. 1b, Supplementary Data 1). Most nuclei isolation methods rely on sucrose gradient[29] or fluorescence-activated nuclear sorting (FANS)[30] for optimal nuclear purity and quality; although detection of rare cell types remain relatively limited. We optimized a nuclei isolation method that enables detection of all known major brain cell types with high purity including rarer cell types (Supplementary Figs. 2, 3). Quality control (QC) and filtration steps were applied based on number of genes, unique molecular identifiers (UMIs) per nuclei and predicted doublets (Supplementary Fig. 4), resulting in 78,396 high quality nuclei in 35 clusters that were annotated for their types according to published cellular markers[31]. Heatmap visualization using well-established cell type markers further confirmed the cell type assignment for the clusters (Fig. 1c, d). All clusters include nuclei from > 20 individuals, i.e. > 80% of cohort, except the two smallest clusters which contain 206 and 105 nuclei (Supplementary Data 2, 3). The clusters represent eight cell types (Fig. 1c) as follows: 14 excitatory neuronal (41% nuclei), 9 inhibitory neuronal (20%), 3 oligodendrocytic (22%), 3 astrocytic (8%), 3 vascular (3%), 2 microglial (3%) and 1 oligodendrocyte progenitor nuclei clusters (3%). We tested the associations of each cell cluster proportion with diagnosis, age, sex, *APOEε*4 and neuropathology measures (Supplementary Figs. 5, 6, Supplementary Data 4–6). An excitatory neuronal cluster 23 (cl.23) has a lower proportion of cells in AD cases, is likewise negatively associated with both Braak stage and Thal phase and is negatively associated with *APOEε*4; inhibitory neuronal cl.10 is also negatively associated with Braak stage. An inhibitory neuronal cl.7 is positively and an oligodendrocytic cl.14 is negatively associated with

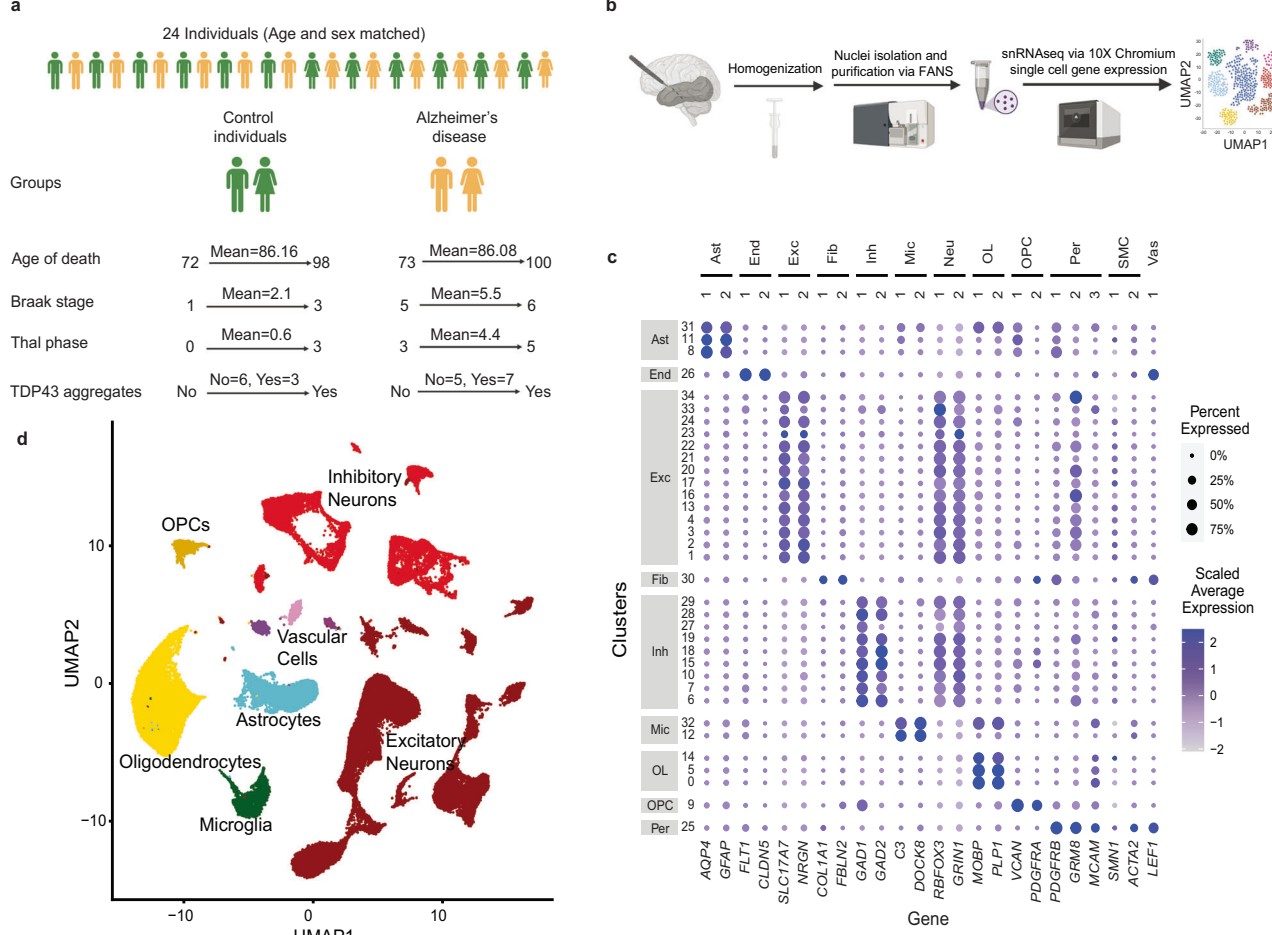

**Fig. 1 | Summary of the snRNAseq approach utilized in this study. a** Post-mortem temporal cortex tissue from 24 individuals that comprise sex and age matched AD and control individuals were used in this study. **b** Development and optimization of nuclei isolation protocol for snRNAseq platform. **c** Well-established cell type markers were used to annotate nuclei clusters and **d** major brain cell types were visualized in UMAP plots. Figure 1/panels **a** and **b** Created with BioRender.com released under a Creative Commons Attribution-NonCommercial-NoDerivs license.

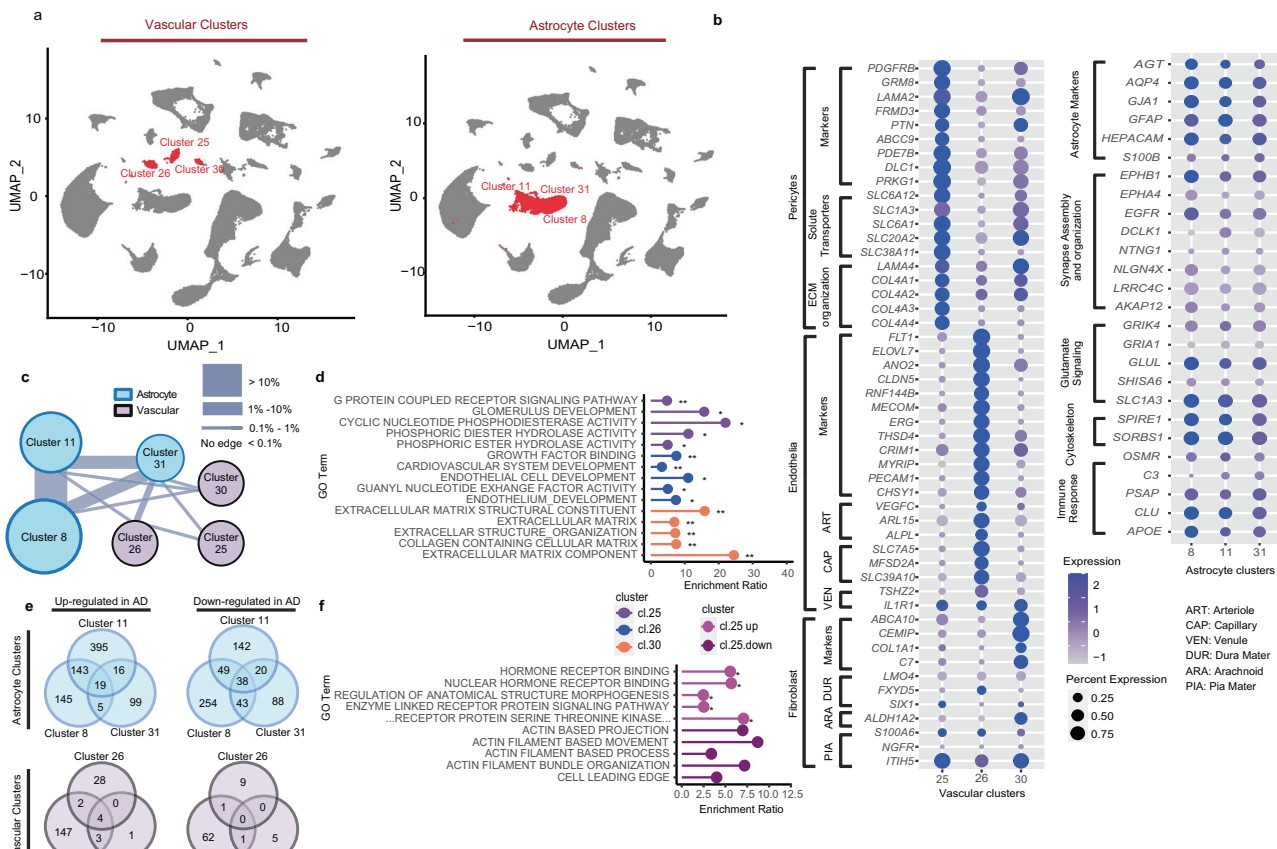

**Fig. 2 | Vascular and astrocytic snRNAseq analyses reveal unique vascular clusters of which pericytes are the most perturbed in AD brains. a** Three vascular and three astrocytic clusters were demonstrated in UMAP plots. **b** We identified three distinct vascular clusters which could be classified as pericytes (cl.25), endothelia (cl.26) and perivascular fibroblasts (cl.30), owing to the unique expression profiles of their highly expressed signature genes. Unlike the vascular clusters, the astrocytic clusters were less distinct from each other. **c** The constellation plot displays the relatedness of the 3 vascular and 3 astrocyte (see Fig. 3) clusters, based on post-hoc classification of cells. The thickness of the connecting line between any two clusters was determined by the percent of cells that are ambiguously assigned. Astrocyte clusters demonstrated greater relatedness as shown by the thick connecting lines (-1-10%). Vascular clusters, on the other hand, demonstrated more distinct cell populations with thin connecting lines (-0-1%).

**d** Top Enriched GO terms of signature genes in each vascular cluster show distinct functions. *: enrichment FDR < 0.05. **: enrichment FDR < 0.001. **e** We also identified DEGs in these vascular clusters; the largest numbers of which were in the pericyte cl.25 (1562 up, 64 down), followed by endothelial cl.26 (34 up, 10 down) and perivascular fibroblasts cl.30 (8 up, 6 down). Pericyte cluster showed the highest number of DEGs in AD further implicating these cells. **f** Top GO Term Enrichment analysis was summarized for pericyte cluster cl.25, which shows pathways involved in cell-to-cell communication are upregulated. The full name of the fifth GO term from the top is, "NEGATIVE REGULATION OF TRANSMEMBRANE RECEPTOR PROTEIN SERINE THREONINE KINASE SIGNALING". Source data are provided as a Source Data file. Figure 2/panels **c** and **e** Created with BioRender.com released under a Creative Commons Attribution-NonCommercial-NoDerivs license.

---

*APOE*ε4. No other cell clusters had nuclei proportion associations with the tested variables.

## AD associated genes and pathways are detected in distinct brain vascular clusters

In this study, we focused on the transcriptional landscape of brain vascular and astrocyte clusters, to discover transcriptional perturbations in these cells of the GVU[7–9]. Three vascular nuclei clusters were identified – cl.25, cl.26 and cl.30, containing 926 (AD:424, control:502), 739 (AD:313, control:426), and 488 (AD:237, control: 251) nuclei, respectively (Fig. 2a, Supplementary Fig. 7). All three clusters express BBB-specific transcription factor *LEF1*[32] (Fig. 2b Supplementary Fig. 7B), which is not expressed in any other clusters. Smooth muscle cell (SMC) markers of peripheral vasculature *RBP1*[33], *SMN1*[34] have limited expression in the brain vascular clusters (Supplementary Fig. 7). Pericytes (cl.25) displayed expected expression pattern of markers involved in solute transport and ECM organization (Fig. 2b). Endothelia (cl. 26) expressed known markers highly expressed in arteriole, capillaries, and venules (Fig. 2b). Cl.30 was consistent with a perivascular fibroblast expression signature.

Three astrocytic clusters cl.8, cl.11 and cl.31 were identified, encompassing 3343 (AD:1862, normal:1481), 2439 (AD:1188, normal:1251) and 383 (AD:246, normal:137) nuclei, respectively (Fig. 2a). The three vascular clusters are well separated in the reduced dimension UMAP plot, whereas three astrocytic clusters are close to each other (Fig. 2a). Using random forest classification to identify any ambiguous or intermediate cells between each of the two clusters, we determined that merely <0.5% of these vascular cells were ambiguous (Fig. 2c, Supplementary Data 7), which further highlights their distinct gene expression. Unlike the vascular clusters, the astrocytic clusters were less distinct from each other as illustrated by their percentage of ambiguous or intermediate cells (1–10%, Supplementary Data 7, Fig. 2c).

To understand the unique biological functions of the vascular and astrocyte clusters, we identified their signature genes that were detected in at least 50% of the cells and that had significantly higher expression in the cluster of interest. Signature genes of a cluster had average expression >= 2.0X higher than that of the other two clusters with a Bonferroni-corrected p-value < 0.05. This resulted in 102, 174 and 80 signature genes for cl.25, cl.26 and cl.30 respectively

(Supplementary Data 8), using which we performed gene ontology (GO) enrichment analyses (Fig. 2d, Supplementary Data 9–11). Cl.25 has an expression signature consistent with pericytes with high expression of pericytic markers *PDGFRB*[35,36], and *GRM8*[22] (Fig. 2b) and GO term enrichment for signaling pathways (Supplementary Data 9). Further, cl.25 has high expression of genes related to nutrient and ion transport (*SLC12A7*[37], *SLC6A12*[38], *SLC19A1*[39]), and formation of blood brain barrier (*COL4A1*[40], *CDH6*[41], *SNTB1*[42]). As expected, endothelial Cl.26 is enriched in endothelial GO terms (Fig. 2d, Supplementary Data 10) and highly expresses endothelial damage associated genes such as *VWF*[43], *ABCG2*[44], *ABCB1*[45] and angiogenesis associated genes such as *ENG*[46], *TGM2*[47], and *ERG*[48] (Fig. 2b, Supplementary Data 8). Finally, cl.30 has high expression of fibroblast markers such as *ABCA9, CEMIP*, and *C7*[22] (Fig. 2b) with enrichment of GO terms for the extracellular matrix (Supplementary Data 11, Fig. 2d). In summary, cl.25, cl.26 and cl.30 have unique expression profiles consistent with pericyte, endothelia, and perivascular fibroblast clusters, respectively.

Signature genes were identified for astrocytic clusters as done for the vascular clusters. There were 20, 12, and 274 signature genes for cl.8, cl.11, and cl.31, respectively (Supplementary Data 12). GO term analysis could only be conducted for cl.31 signature genes which showed enrichment for synaptic signaling, and myelination terms (Supplementary Data 13), suggesting that this may either be a mixed cluster, or one involved in astrocyte-neuron and oligodendrocyte interactions.

We next performed DEG analyses for each GVU cluster to compare their gene expression in AD vs. control tissue using the MAST R package[49]. Imposing a q value < 0.05, an absolute log (fold change) > 0.1 and detection of gene expression in >= 20% AD or control cells, 220 (156 up, 64 down), 44 (34 up, 10 down), and 14 (8 up and 6 down) DEGs were identified in pericyte cl.25, endothelial cl.26 and perivascular fibroblast cl.30, respectively (Fig. 2e, Supplementary Data 14). Four genes are up-regulated in AD across all three vascular clusters (*INO80D, LINGO1, RASGEF1B, SLC26A3*) and no down-regulated genes are shared. Most DEGs were detected in cl.25, supporting pericytes being the most perturbed vascular cluster in AD. The limited number of overlapping DEGs in any two clusters (Fig. 2e) further confirmed that these clusters are distinct from each other and likely have different biological roles. Notably *PLCG2* is 1 of 7 genes upregulated in both pericyte and perivascular fibroblast clusters, while another gene implicated by AD genetic studies, *MEF2C*, is downregulated in cl.25. While *PLCG2* is predominantly expressed in microglia[50], our data and others[22] also implicate its upregulation in brain vasculature which may likewise be relevant for AD pathogenesis.

GO enrichment analyses were performed for those vascular cluster DEGs that had sufficient numbers, i.e. genes up or down in AD in cl.25, and genes up in cl.26 (Supplementary Data 15–17). The top 5 GO terms from pericyte cl.25 are shown in Fig. 2f. The top perturbed genes and their enriched GO terms are growth factor related genes upregulated (*FLT1, SMAD3, STAT3*) (Supplementary Data 15) and cytoskeleton related genes downregulated (*DMD, MYO1B*) in pericyte cl.25 (Supplementary Data 16). Of these, *STAT3* is also up in endothelial cl.26, which additionally harbors upregulated angiogenesis related genes (*ANGPT2, INSR*) (Supplementary Data 17). These findings support AD-related expression changes in distinct brain vascular cells.

Genes in astrocytic clusters that were differentially expressed between AD and control brains were identified. Cl.8, cl.11 and cl.31 contain 696 (312 up, 384 down), 822 (573 up, 249 down), 328 (139 up, 189 down) DEGs, respectively (Fig. 2e, Supplementary Data 18). Top GO terms of DEGs upregulated in AD within cl.8 and cl.11 include actin cytoskeleton and cell differentiation related terms, whereas for cl.31 the top enriched terms in upregulated DEGs are related to cytoskeleton, neurogenesis, and ensheathment of neurons (Supplementary Data 19–21). For DEGs that are downregulated in AD in the astrocytic clusters, the top enriched GO terms include cell signaling,

neurogenesis and cilia/motility related processes (Supplementary Data 22–24). Unlike vascular DEGs, about 23% of the astrocytic DEGs are shared in two or more clusters. GO enrichment analyses of these commonly perturbed genes in AD astrocytes demonstrate enrichment of cytoskeleton and neurogenesis-related terms for upregulated, and cilium and calcium transport related terms for downregulated genes. Comparison of the astrocytic cluster DEGs in our study to a previously published study that focused on astrocytes[18] revealed significant overlap, as well as unique genes (Supplementary Fig. 8). Differences may be attributed to several factors such as brain region and donor sampling, whereas commonalities likely represent disrupted astrocytic processes that are robust to these. Our findings support widespread transcriptome perturbations in AD astrocytes. There are many shared DEGs between astrocytic clusters which underscore more similar transcriptional changes in AD for this cell type in comparison to those for the brain vascular clusters.

### Ligand-target interactions between astrocytic and vascular AD-associated genes

Cell biology studies of the GVU have discovered multiple interactions between astrocytes and brain vasculature that are mediated through ligand-target interactions. Further systematic efforts are needed to discover the vast and complex molecular relationships between these cells of the BBB[7–9]. We aimed to identify a prioritized set of vascular targets that are regulated by astrocyte ligands and consequently influence brain vascular functions at the GVU of the BBB. To accomplish this, we used transcriptome data from the brain vascular and astrocyte clusters and the NicheNet[51] analytic platform that utilizes prior knowledge of such interactions. As our goal was to determine those vascular target-astrocyte ligand pairs that are most perturbed in AD, we confined our analyses to the significant vascular and astrocytic DEGs. Using NicheNet[51] and focusing on significant DEGs in astrocytic clusters, we identified a combined pool of 40 unique potential ligand genes that have corresponding targets in one or more of the vascular clusters (Supplementary Data 25, Supplementary Fig. 9). There were 22, 4, and 2 predicted vascular targets in the pericyte cl.25, endothelial cl.26, and perivascular fibroblast cl.30, respectively, comprising 26 unique target genes (Supplementary Fig. 10).

These 26 brain vascular target candidates include genes with diverse biological functions (Fig. 3a), including cytoskeleton and ECM-related (*TIMP3*[52], *AHNAK*[53], *SLC38A2*[54], *STARD13*[55]); growth factor-related (*STAT3, SMAD3, TGFB1, TFPI, EGFR, FGFR1, PDGFA*)[56–58]; glucocorticoid-related and anti-inflammatory (*NR3C1, TSC22D3*)[59]; angiogenesis (*ANGPT2*)[60,61]; as well as *ECE1*, an AD-related gene that is involved in Aβ clearance and vasoconstriction[62,63]. There were established AD genes amongst the top astrocytic ligands namely *APOE* corresponding to predicted vascular target *TSC22D3*[59] with high estimated regulation strength and *APP* with high regulation strength for *ECE1*[62,63]. (Fig. 3a, Supplementary Fig. 11).

We next sought to validate a subset of the 26 predicted vascular targets, using quantitative PCR (qPCR). We selected genes representative of the biological functions associated with these 26 genes (Supplementary Fig. 11, Supplementary Data 26). Genes selected for validations include *SMAD3* and *STAT3* which are growth-factor related signaling molecules[57,64,65]. Amongst all 26 vascular genes, *SMAD3* has the strongest predicted interactions with astrocytic ligands, is upregulated in pericyte cl.25 (Fig. 3b, Supplementary Data 26) and is one of the most frequently observed genes in the GO terms enriched for this cluster (Supplementary Data 15). *STAT3*, also a strong vascular target, is upregulated in both cl.25 and endothelial cl.26 (Supplementary Data 26). Other selected genes were *AHNAK, ANGPT2, ECE1* and *TSC22D3*. *AHNAK*, the second most strongly connected vascular target encoding a structural protein involved in BBB integrity[53], is an upregulated DEG in pericyte cl.25 (Supplementary Data 26). *TSC22D3* and *ECE1* that are also upregulated in pericytes have known Aβ-related

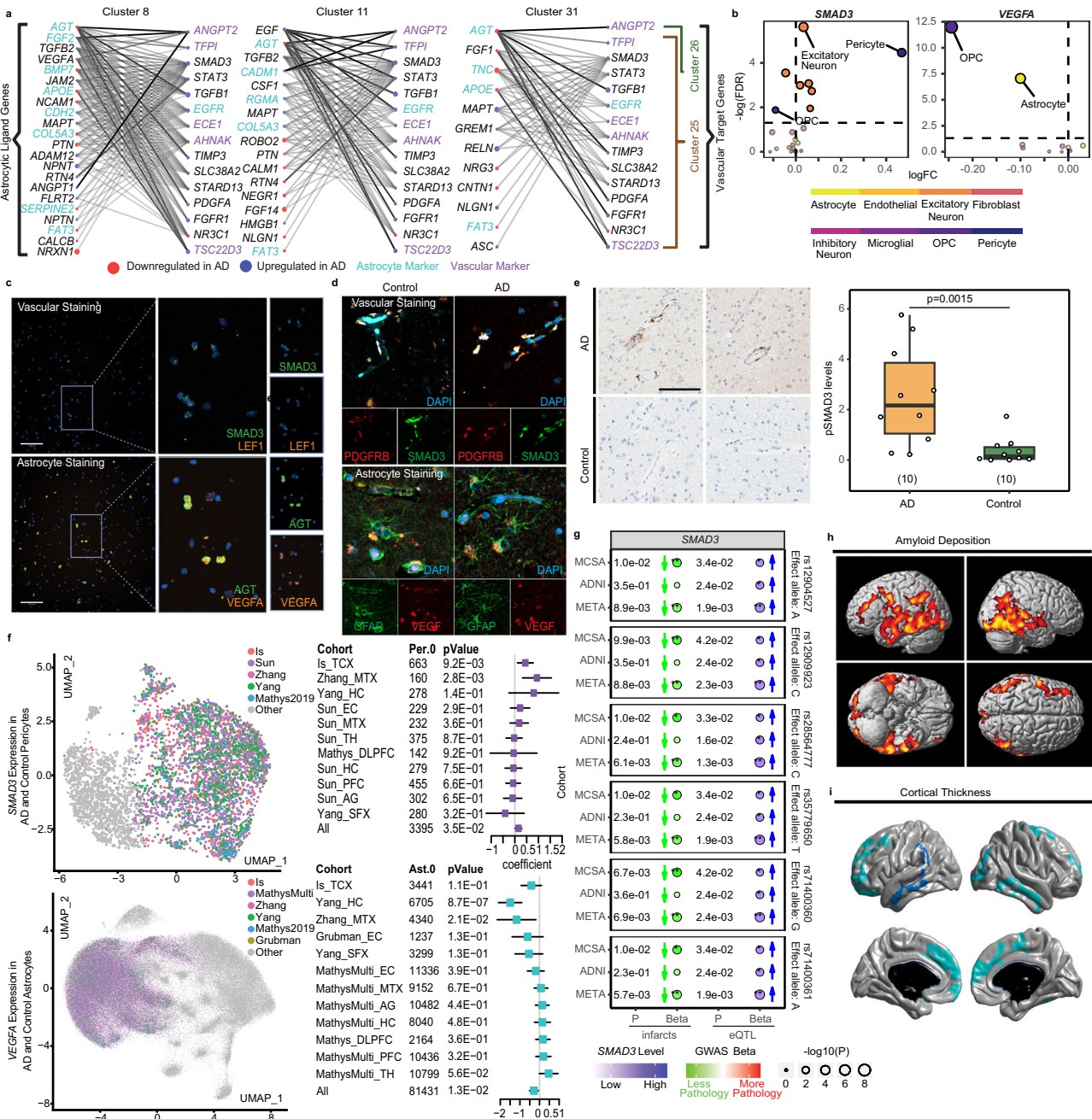

**Fig. 3 | Discovery, prioritization, validation, and replication of perturbed GVU vascular target-astrocyte ligand pair SMAD3-VEGFA. a** Strength and direction of NicheNet vascular target-astrocyte ligand interactions. Left: predicted ligands in astrocyte clusters. Right: predicted targets in vascular clusters. Edge: regulation strength between ligands and target genes; Cyan astrocyte, purple: endothelial markers. Direction of change in AD is denoted as blue for up and red for down-regulation. **b** Of the perturbed vascular targets in AD brains, *SMAD3*, which is upregulated in AD pericytes and has strong astrocytic connections, is prioritized. Of the astrocytic ligands, *VEGFA*, which is downregulated in AD and has strong predicted interactions with *SMAD3*, is prioritized. **c, d** We validated expression of SMAD3 in vascular cells and VEGFA expression in astrocyte cells through RNAscope (scale bar:100 μm) and immunofixation (scale bar:10 μm). **e** Immunohistochemistry results showed significantly higher phospho-SMAD3 immunoreactivity in AD compared to controls in pericytes ($p < 0.01$, $n = 10$ per diagnosis). **f** *SMAD3* and *VEGFA* brain expression changes in external brain snRNAseq studies. Pericytes (purple) and astrocytes (cyan) were from multiple studies and were clustered (Gray dots: other nuclei). In forest plots, the square indicates the coefficient, which is the natural log(fold change). The left bar: 2.5% confidence interval; the right bar: 97.5% confidence interval. (Ast: astrocytes, Per: pericytes, TCX: temporal cortex, MTX: midtemporal cortex, EC: entorhinal cortex, DLPFC: dorsolateral prefrontal cortex, PFC: prefrontal cortex, SFX: superior frontal cortex, Hippocampus: HC, AG: angular gyrus, TH: thalamus) **g** 6 intronic variants associated with higher blood expression levels of *SMAD3* (eQTL) were also associated with decreased brain infarcts in ADNI, MCSA, and meta-analyzed cohorts. P-values and direction of effects from the infarct GWAS and the eQTL analysis in MCSA, ADNI, and meta-analysis (random effects) are shown. **h, i** Whole-brain association analysis of blood *SMAD3* levels with brain Aβ deposition and cortical thickness in the ADNI cohort. Color scales indicate regions where higher blood *SMAD3* were associated with less brain amyloid-β deposition and less brain atrophy, respectively. Statistical maps were thresholded for a multiple testing adjustment to a corrected significance level of 0.05. Source data are provided as a Source Data file.

functions[62,63,66], predicted astrocytic ligands that are AD genes (Fig. 3a), and anti-inflammatory and vasoconstrictive properties, respectively. Finally, *ANGPT2* that is the most significantly upregulated DEG in endothelial cl.26, is a strong target in this cluster (Fig. 3a, Supplementary Data 26). *ANGPT2* is involved in angiogenesis[61] and like *SMAD3* and *STAT3*, is a signaling molecule.

Using nuclei isolated from the same brain region (temporal cortex) of the same donors, we collected RNA from bulk nuclei and measured the expression of these genes using qPCR. We found that all 6 genes had higher expression in the AD cases than in the controls, with all but 1 (*ANGPT2*, p = 0.066) reaching significance, thus validating our prior findings in the snRNAseq data (Supplementary Fig. 12A, Supplementary Data 27). For completeness, expression levels of the six prioritized vascular target genes were also tested for associations with AD-related neuropathologies (Supplementary Fig. 12B), age, sex and *APOEε*4 (Supplementary Fig. 12C). The results are detailed in Supplementary Information, where neuropathology associations are consistent with that expected from AD-related DEG results, and some associations were also detected with sex and age, but not with *APOEε*4.

## Validation and replication of the prioritized pericyte target-astrocyte ligand pair *SMAD3-VEGFA* in human AD and control brains

Of the 6 prioritized and qPCR-validated vascular genes, we focused on *SMAD3*, a signaling molecule with known vascular functions[1,67], which shows upregulation in AD pericytes and has the strongest level of astrocyte ligand interactions (Supplementary Fig. 13, Supplementary Data 25, 26). The ligands of *SMAD3* in astrocyte clusters include several known AD-related genes that are also DEGs in our study such as *APOE*, *APP*, *PSEN1*, and *MAPT* (Supplementary Fig. 13). Further, *SMAD3* expression change in AD has the strongest effect size in the pericyte cluster (β = 0.47, q = 3.42e-05) (Fig. 3b). The other cells, where *SMAD3* showed significant difference between AD and controls, were six excitatory neuron clusters (cluster1: β = −0.045, q = 0.00028; cluster2: β = 0.057, q = 0.00084; cluster3: β = 0.034, q = 2.40e-06; cluster4: β = 0.063, q = 0.01117; cluster17: β = 0.071, q = 0.00186; cluster24: β = 0.019, q = 0.00101), and the OPC cluster (Cluster9: β = −0.090, q = 0.01320). However, the effect sizes of *SMAD3* changes in AD within these clusters were smaller than that for pericytes (Fig. 3b). These findings suggest that *SMAD3* is well-connected with astrocytic ligands and has AD-related expression changes that are most pronounced in pericytes. Consequently, we selected *SMAD3* for downstream replication and experimental validations.

Amongst the astrocytic ligands of *SMAD3*, we prioritized *VEGFA*, an angiogenic growth factor that is involved in multiple processes in the human brain that include synaptic plasticity, memory formation, cognition, and the progression of AD[68-70]. *VEGFA* is mainly expressed by astrocytes (Supplementary Fig. 14) and its expression is significantly downregulated in astrocytic cl.8 and OPC cluster (Fig. 3b). Astrocytic *VEGFA* is one of the most well-connected predicted ligands for pericytic *SMAD3* in our analyses (Fig. 3a, Supplementary Data 25). Prior in vitro studies determined that SMAD3 mediates TGFβ-signaling related effects on VEGFA[71,72]. Therefore, we prioritized astrocytic *VEGFA* and its predicted pericytic partner *SMAD3* as the GVU molecular pair, for further validations and replications in human brains.

We first validated expression of *SMAD3* in vascular cells and *VEGFA* in astrocytes. Using *LEF1* and *AGT* as vascular and astrocytic markers, respectively (Supplementary Figs. 7b, 14), we performed co-staining with RNAscope. We isolated bulk nuclei from human temporal cortex of 9 AD and 9 control patients and co-stained with astrocytic (*VEGFA-AGT*) and vascular (*SMAD3-LEF1*) RNAscope probe pairs (Fig. 3c). We captured and visualized images via Operetta CLS high content imager (Perkin Elmer) and utilized our custom RNAscope pipeline on Cell Profiler (version 4.2.5) to analyze a total number of 50,946 DAPI[+] nuclei (AD: 25,055, Control: 25,891) for astrocytic

staining and 62,072 DAPI[+] nuclei (AD: 32,442, Control: 29,630) for vascular staining. Out of these, 5370 DAPI[+] + AGT[+] nuclei (AD: 2,769 Control: 2,601) and 1,486 DAPI[+] + LEF1[+] (AD: 755 Control: 731) nuclei were annotated as astrocytic and vascular, respectively. Proportions of astrocytes (~10%) and vascular nuclei (~2%) in all bulk nuclei assessed with RNAscope are similar to those detected in our brain snRNAseq results. We observed that VEGFA[+] staining in AGT[+] nuclei ranged from 10 % to 68 % (Median = 40.89) and SMAD3[+] staining in LEF1[+] nuclei ranged from 14% to 62% (Median = 41.78) (Supplementary Data 28, 29).

We also performed immunofluorescence (IF) in the temporal cortex of 2 AD and 2 control donors who were not part of our snRNAseq study. Using astrocytic GFAP and pericytic PDGFRB markers, we observed co-expression with VEGFA and SMAD3, respectively, (Fig. 3d). Thus, we validated pericytic SMAD3 and astrocytic VEGFA expression in human brain tissue both at the RNA and protein level.

We also measured phospho-SMAD3 immunoreactivity in an additional 10 AD and 10 Control donors from Mayo Clinic Brain Bank to validate changes in this active signaling form of SMAD3 protein[73]. Using a custom analysis pipeline[74], we identified immunopositive pixels in the immunoreactive area. Pericytes were distinguished by their unique morphology and localization around blood vessels. AD pericytes showed significantly increased (p < 0.01) pSMAD3 reactivity compared to control subjects. (Fig. 3e, Supplementary Data 30).

We also sought replication of our brain snRNAseq findings of upregulated pericytic *SMAD3* and downregulated astrocytic *VEGFA* in AD in external, independent snRNAseq datasets from multiple brain regions (Fig. 3f, Supplementary Data 31–34, Supplementary Figs. 15, 16). For pericytes, we integrated the midtemporal cortex (MTX) of Zhang et al.(GSE188545)[23], six brain regions of Sun et al.[21], dorsolateral prefrontal cortex (DLFPC) of Mathys et al.[20], and superior frontal cortex (SFX) and hippocampus (HC) of Yang et al.[22]. Sun et al.[21]. and Yang et al.[22]. had over-representation of prefrontal cortex (PFC) and selected vascular nuclei, respectively, and were therefore downsampled (Supplementary Data 31). In total, 4,730 pericytic nuclei were clustered into two subclusters (Supplementary Data 32, Supplementary Fig. 15). Per.0 cluster demonstrates upregulation of solute transport genes, whereas Per.1 shows extracellular matrix organization gene upregulation (Supplementary Data 34, Supplementary Fig. 15). Per.0 and Per.1 resemble T-pericytes and M-pericytes from Yang et al.[22], respectively. Per.0 has >2.5X greater number of nuclei (n = 3395) than Per.1 (n = 1335), and also contains the majority of nuclei of pericytic cl.25 from our data (Fig. 3f, Supplementary Data 32). In Per.0, *SMAD3* was significantly up-regulated in AD donors in our study, Zhang et al.[23]. and all cohorts combined, with a trend of up-regulation in HC of Yang et al.[22], and EC and MTX of Sun et al.[21]. (Fig. 3f, Supplementary Data 32). There is no significant down-regulation of *SMAD3* in any of the cohorts. Interestingly, the pericytes from our study showed significant up-regulation in AD *SMAD3* expression in both integrated pericyte subclusters, although this association was not significant in all cohorts combined for Per.1 (Supplementary Data 32).

For astrocytes, we integrated our dataset, Zhang et al.[23], Grubman et al.[15], Yang et al.[22], DLFPC from Mathys et al.[14], and six brain regions from Mathys et al.[20]. In total, 150,664 astrocytic nuclei clustered into 14 subclusters, the smallest one contained only two nuclei (Supplementary Data 33, Supplementary Fig. 16). Notably, the largest astrocyte subcluster 0 (Ast.0) contained the majority of astrocytic nuclei from the integrated datasets (n = 81,431) as well as that of astrocytic cluster 8 of our own data where *VEGFA* is downregulated in AD (Supplementary Data 33). Ast.0 is enriched for genes involved in synaptic assembly and organization compared to other clusters (Supplementary Data 34, Supplementary Fig. 16). These nuclei display a gene expression profile similar to those in the Mathys et al. GRM[+] astrocyte subcluster[20]. In Ast.0, *VEGFA* was significantly downregulated in AD participants in Yang et al.[22] HC region, Zhang et al.[23] and all cohorts combined, with a trend of downregulation in our study, Grubman et al.[15], Yang et al.[22] SFX

region, Mathys et al[20] EC region (Fig. 3f, Supplementary Data 33). There is no significant upregulation of *VEGFA* in any of the cohorts. *VEGFA* is also down-regulated in AD in the combined datasets for the second largest astrocyte subcluster comprising 42,880 nuclei and the entire astrocytic cluster of this integrated dataset (Supplementary Data 33).

In summary, our postmortem analyses of brain snRNAseq data discovered perturbed vascular and astrocytic transcript pairs, of which pericytic *SMAD3* (up in AD) and astrocytic *VEGFA* (down in AD) were prioritized. These findings were validated with orthogonal quantitative PCR, RNAscope and immunohistochemistry studies and replicated in external human brain snRNAseq data.

## Association of blood *SMAD3* gene expression levels with infarcts, Aβ deposition and cortical atrophy

We next aimed to determine whether brain *SMAD3* expression perturbations detected in vascular cells from deceased AD patients could also be captured in blood samples of living patients. Our goal was to detect whether brain perturbations of vascular molecules could also be detected peripherally and whether these peripheral levels associate with vascular and other AD-related outcomes. We analyzed existing blood *SMAD3* expression, genetic and imaging data from two longitudinal antemortem cohorts, Mayo Clinic Study of Aging (MCSA)[75] and Alzheimer's Disease Neuroimaging Initiative (ADNI)[76]. First, we hypothesized that genetic variants that influenced *SMAD3* expression levels could also impact brain vascular disease burden. To test this, we used the neuroimaging variable of infarcts as a surrogate for vascular disease burden[77], obtained from MCSA ($n = 1508$) and ADNI ($n = 1080$). We tested the association of infarcts with 588 genetic variants in the *SMAD3* locus in each cohort and subsequently performed meta-analysis. These variants were also tested for association with blood *SMAD3* levels in 395 MCSA and 645 ADNI participants[78].

Random effects meta-analysis (Fig. 3g) of genetic associations with infarcts and with blood *SMAD3* levels in ADNI and MCSA revealed 6 intronic *SMAD3* variants (rs71400360, rs12904527, rs12909923, rs71400361, rs35779650, rs28564777) that had nominally significant associations ($p < 0.05$) with both lower risk of brain infarcts and with higher blood levels of *SMAD3*. Results for sex- and *APOE*-ε4-stratified association analyses revealed similar directions of effect as in unstratified analyses (Supplementary Data 35, Supplementary Information).

Additionally, using microarray-based blood *SMAD3* expression levels, amyloid β (Aβ) positron emission tomography (PET) scan and magnetic resonance imaging (MRI) available from the same ADNI patients ($n = 638$), we performed whole-brain association analysis of blood *SMAD3* levels with brain Aβ deposition and cortical thickness (Supplementary Fig. 17, Supplementary Data 36). We determined that higher blood *SMAD3* levels are associated with less brain amyloid (Fig. 3h) and less cortical atrophy (Fig. 3i), especially in the temporal, parietal, and frontal lobes (corrected *p*-value < 0.05). In summary, our antemortem analyses revealed associations of *SMAD3* locus genetic variants with both higher blood *SMAD3* levels and lower brain infarcts. Further, higher blood *SMAD3* levels associated with less amyloid and cortical atrophy on antemortem imaging. Collectively, these findings demonstrate that blood *SMAD3* levels may be reflective of brain vascular disease, Aβ and neurodegeneration.

## In vitro validations of *SMAD3*-*VEGFA* interactions

Our human brain snRNAseq data analyzed by NicheNet[51] predicted interactions with pericytic targets and astrocytic ligands, of which we prioritized SMAD3-VEGFA molecular pair perturbed in AD brains, where former is also associated with antemortem AD outcomes. To validate molecular interactions of *SMAD3* and *VEGFA* in vitro, we utilized human iPSC-derived pericytes from AD and control participants (Fig. 4a, Supplementary Fig. 18). To minimize any sex and *APOE*-ε4-related variability, we utilized well-characterized iPSCs from 2 AD and 2 control female participants with *APOE*-ε4/ε4 genotypes for pericyte

differentiation[79] followed by treatments to activate or inhibit VEGFA signaling[80–83]. We validated pericyte differentiation through staining of pericyte markers using flow cytometry, immunocytochemistry (ICC), and RT-qPCR (Fig. 4b–d). We validated pericytic *SMAD3* expression using RNAscope (Supplementary Fig. 19) and assessed the impact of VEGF (encoded by *VEGFA*), VEGF receptor-2 (a.k.a. VEGFR2 or KDR) inhibitor cocktail and Aβ treatment on *SMAD3* expression with RT-qPCR at 6, 12, and 24 h following each treatment (Supplementary Data 37, Supplementary Fig. 20). Compared to the matched pericytes treated only with media, pericytes treated with VEGF had a treatment-duration-dependent reduction of *SMAD3* expression (Fig. 4e), with significant decrease at 24 h post VEGF treatment ($p = 1.17E-3$). In separate assessment of AD and control pericytes, we observed *SMAD3* reductions upon VEGF treatment at 24 h in both diagnostic groups (Supplementary Fig. 21). There was an acute increase in *SMAD3* expression after 6 h of VEGF treatment only in AD pericytes (Supplementary Fig. 21), though this was not sustained in later timepoints. *SMAD3* reductions were observed using 3 different concentrations of VEGF at 50, 100 and 200 μM (Supplementary Fig. 22).

Consistent with these findings, VEGF receptor KDR inhibitor cocktail treatment significantly elevated *SMAD3* expression compared to vehicle-treated conditions at all time points both in combined analyses and those done separately for AD and controls (Fig. 4f, Supplementary Fig. 21B). We did not observe any significant change in *SMAD3* expression after Aβ treatment at any treatment duration (Fig. 4g, Supplementary Fig. 21C).

In summary, our in-vitro analyses validated *VEGFA*-*SMAD3* interactions in human iPSC-derived pericytes. Treatment of human pericytes with VEGF (encoded by *VEGFA*) reduces *SMAD3*, and blocking VEGF signaling increases *SMAD3*.

## In vivo validations of the impact of *SMAD3*-*VEGFA* interactions on the blood-brain-barrier (BBB) integrity in an experimental zebrafish model

To determine whether regulation of SMAD3 signaling by VEGF is conserved in an in vivo model system and whether this interaction has any impact on the BBB integrity, we used well-established zebrafish models. First, we tested the effect of amyloid β (Aβ)[84,85] in astrocytic *vegfaa* (*VEGF* ortholog) and pericytic *smad3* levels in the adult telencephalon of double reporter transgenic zebrafish line – Tg(her4:DsRed)[86] and Tg(fli1a:eGFP)[26] (Supplementary Fig. 23A). We generated scRNAseq profiles of astrocytes and vascular cells from the brains of PBS- and Aβ- injected zebrafish models after FANS (Supplementary Fig. 24). After QC and clustering (Supplementary Fig. 25), we annotated the brain cell types with commonly used zebrafish brain cell type markers (Supplementary Fig. 23B). Six astroglial cell clusters were identified – cl.0, cl.1, cl.2, cl.8, cl.10, and cl.12, containing 8,529 cells (Aβ42: 3,249, PBS: 5,280) (Supplementary Data 38, 39). The expression of *vegfaa* is significantly lower in the astroglial clusters in the adult zebrafish brain injected with Aβ (Supplementary Fig. 23C). In addition, we obtained scRNAseq profile of zebrafish vascular cells - cl.27, cl.28, and cl.32, containing 638 cells (Aβ42: 320, PBS: 318) (Supplementary Data 38, 39). Out of these vascular clusters, only cl.27 expressed pericyte markers. This cluster comprised only 100 cells (Aβ42: 47, PBS: 53), representing only a small portion of vascular cells, consistent with the rarity of this cell type. There was a tendency towards increased *smad3a* expression in pericytes after Aβ injection, though this did not reach statistical significance (Supplementary Fig. 23C).

We next tested the effect of blocking VEGF signaling on SMAD3 and BBB integrity. We pharmacologically treated transgenic zebrafish model Tg(*kdrl*:GFP), with Vegfr2 blockers to reduce Vegf signaling (Fig. 5a). Our structural comparison between human and zebrafish Vegfr2 predicts that the catalytic domain is highly conserved between both species and drugs will be similarly effective in both (Supplementary Fig. 26). Vegf activates pERK signaling in zebrafish to promote

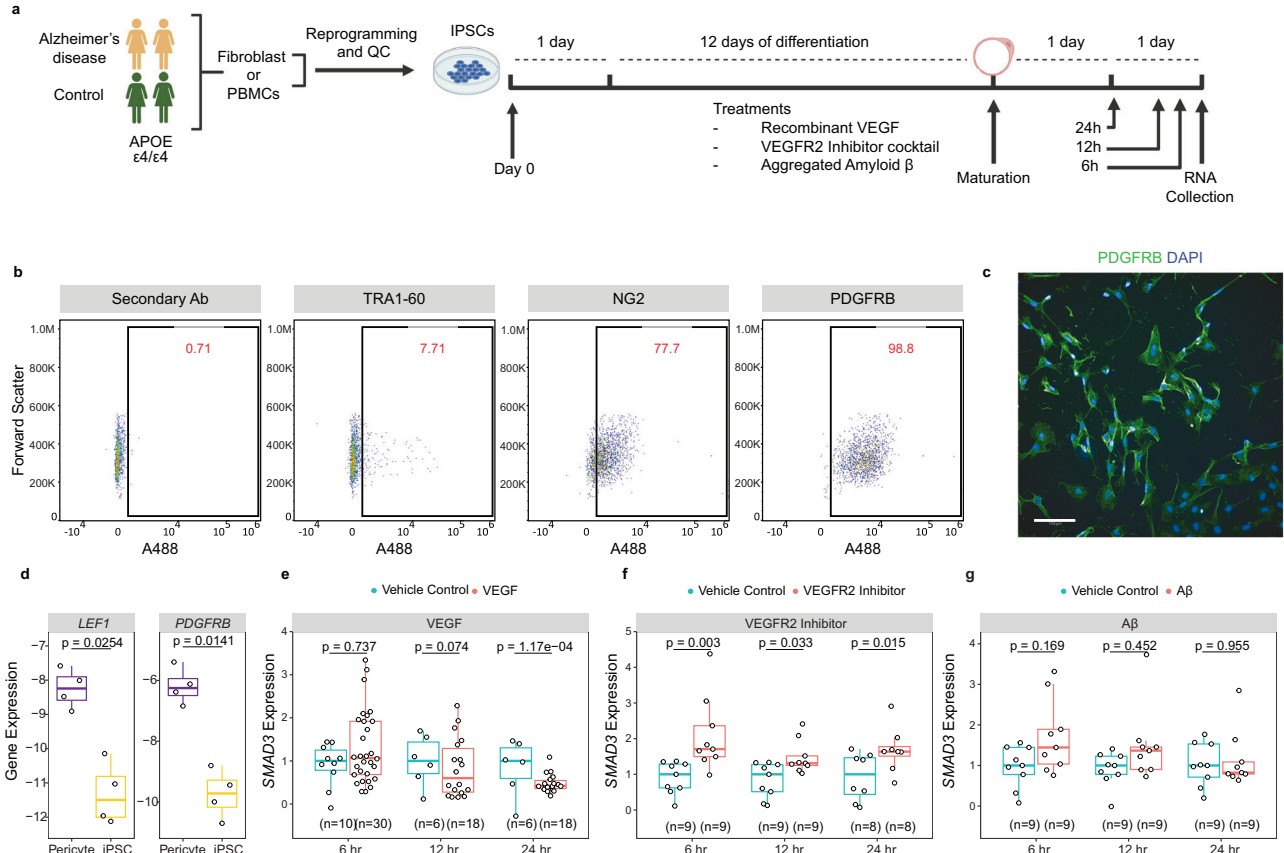

**Fig. 4 | VEGF regulates of SMAD3 expression levels in pericytes. a** We differentiated 2 AD and 2 control patient-derived iPSCs to pericytes as previously described[79], treated the differentiated pericytes with recombinant VEGF, VEGFR2 (KDR) inhibitor cocktail, and aggregated Aβ and analyzed the impact on *SMAD3* expression at three time points (6, 12, and 24 h). **b–d** Validation of pericyte differentiation was performed via flow cytometry, immunocytochemistry, and RT-qPCR. We observed decreased expression of iPSC pluripotency marker, TRA-10, and increased expression of pericytic PDGFRB and NG2 through FACS in the differentiated pericytes. We also visualized and confirmed pericytic PDGFRB expression through ICC (scale bar:100 μm) and observed upregulation of pericyte and vascular markers after differentiation. Statistics: two-sided paired t-test, *n* = 4 biologically independent samples; within each experiment, *n* = 6 technical replicates.

**e** We observed significant decrease in *SMAD3* expression after 24 h of VEGF treatment. **f** Consistently, VEGFR2 inhibitor cocktail treatment caused significant increase in *SMAD3* expression at all time points. **g** Aggregated Aβ treatment did not cause significant change in *SMAD3* expression (*n* = 5 per each duration). Statistics derived from biologically independent replications (different iPSC lines and differentiation batches). All boxplots represent the first quartile, the median, and the third quartile. The upper whisker indicates the maximum value no further than 1.5 times the inter-quartile range from the third quartile. The lower whisker indicates the minimum value no further than 1.5 times the inter-quartile range from the first quartile. Source data are provided as a Source Data file. Figure 4/panel a Created with BioRender.com released under a Creative Commons Attribution-NonCommercial-NoDerivs license.

angiogenesis[87] and we confirmed the inhibition of Vegf signaling upon Vegfr2 blocker treatment by analyzing pERK/GFP colocalization (Supplementary Fig. 27A). We detected a significant decline in pERK/GFP colocalization after Vegfr2 blocker treatment (*n* > 3) (Fig. 5c). We next assessed the impact of Vegfr2 blocking on Smad3 signaling. We counted the number of the active signaling molecules pSMAD3+ endothelial (GFP+/DAPI+) and pericyte cells (GFP+/DAPI+) after blocking Vegf signaling. Pericytes can be distinguished by their unique cellular localization and morphology in the vasculature (Fig. 5b). Vegfr2 blockage increases the percentage of pSMAD3+ endothelial cells and pericyte cells (Fig. 5b), consistent with an activation in Smad3 signaling. We further analyzed the impact of Vegfr2 blockage on the BBB integrity. For this purpose, we evaluated zebrafish brain vasculature tight junction protein (ZO-1) and GFP colocalization (Supplementary Fig. 27B). Blocking Vegfr2 caused significant decrease in the colocalization of ZO-1 and GFP, highlighting dysfunctional vasculature (Fig. 5d). Correlation between random measurement points between colocalization analyses and ZO-1/GFP analysis indicated positive correlation between pERK/GFP and ZO-1/GFP (Fig. 5e), suggesting that the reduction of Vegf signaling is correlated with BBB disintegrity.

In summary, in vivo results validated astroglial *vegfaa* reductions upon Aβ42 treatment in the zebrafish model. Furthermore, our results provide a mechanistic link between *SMAD3-VEGF* interactions and their potential role in BBB disintegrity in AD.

## Discussion

Single cell and single nuclei approaches have been instrumental in revealing the molecular perturbations in AD, however most of these studies have been focused on abundant brain cell types[13–15,17]. Despite the known breakdown of BBB in AD[1,2,6,88,89], there is relative paucity of sn/scRNAseq studies focusing on brain vascular cells in AD, likely due to their low frequency. More recently, several studies evaluated vascular transcriptome changes in AD at single cell magnification. Lau et al.[18]. study obtained gene expression profiles from ∽2400 endothelial nuclei in 12 AD and 9 control brain samples. Yang et al.[22]. developed a vascular nuclei enrichment method that allowed them to profile around ∽144,000 nuclei from 9 AD and 8 control brain samples, which identified several genes and pathways. A recent study, by Sun et al.[21], from six brain regions of a large cohort that comprise 220 ADs and 208 age matched controls, identified several vascular cell type and

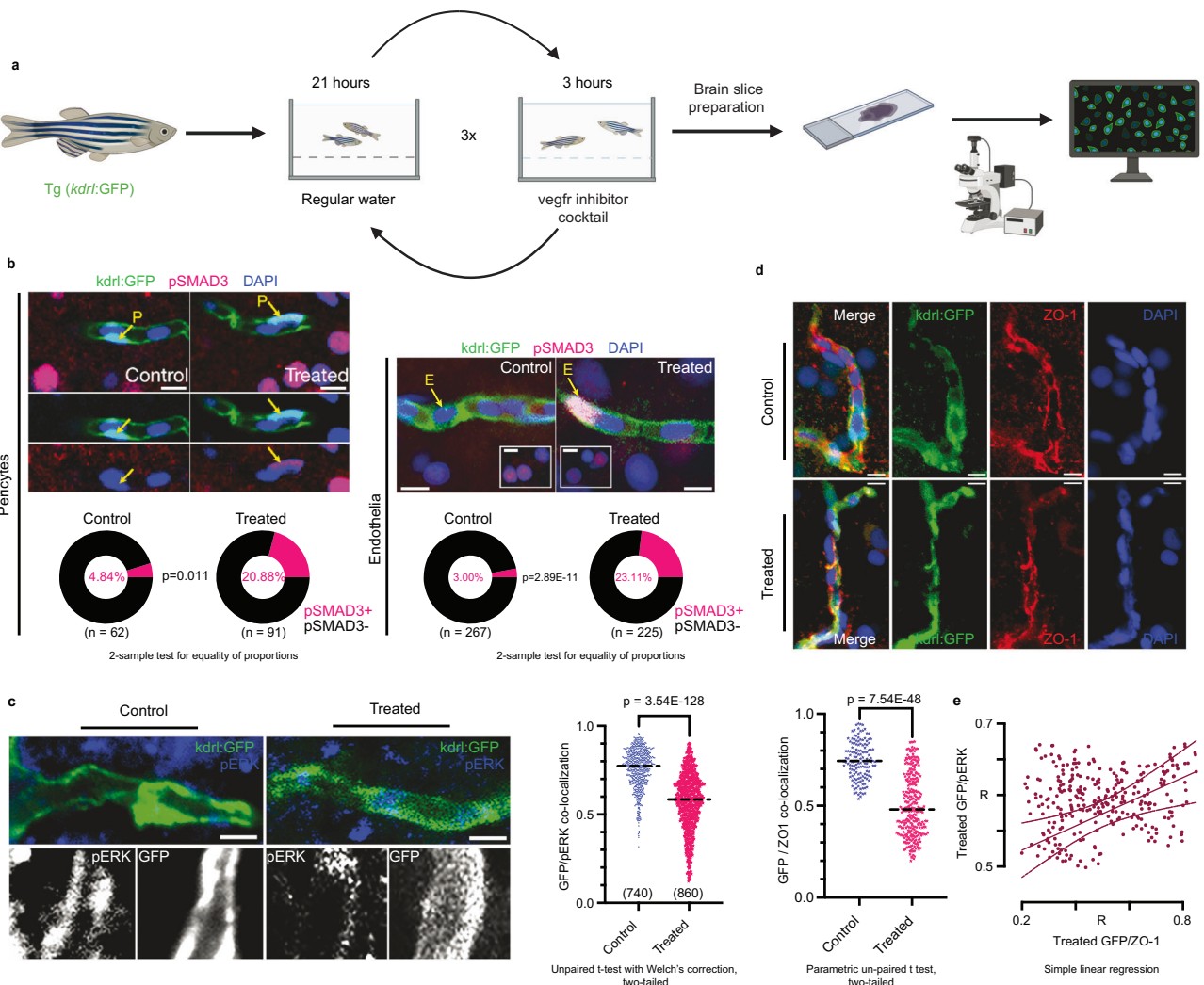

**Fig. 5 | SMAD3-VEGF interactions influence blood-brain-barrier integrity in a zebrafish amyloidosis model. a** We pharmacologically treated transgenic zebrafish model Tg(*kdrl*:GFP) with vegfr2 blockers to reduce VEGF signaling. **b** Double immunostaining for GFP and pSMAD3 coupled to DAPI nuclear counterstain. Lower panels indicate percentage of pSMAD3⁺ cells. In endothelia panel, insets indicate neuronal pSMAD3. Inside brackets the number of analyzed cells are shown. Vegfr2 blockage increased the percentage of pSMAD3⁺ endothelial cells and pericytes (GFP⁺/DAPI⁺). **c** pERK and GFP double immunostaining coupled to DAPI nuclear counterstain in control and vegfr2 blocker treated zebrafish models. Inside brackets the number of analyzed spots are shown. Vegfr2 blocking decreased pERK/GFP colocalization in zebrafish models. **d** Double immunostaining for ZO-1

and GFP in control and vegfr2 blocker-treated zebrafish models coupled to DAPI nuclear counterstain. Treatment caused decreased colocalization of ZO-1/GFP, indicating impaired integrity in zebrafish brain vasculature. Correlation graph between random measurements between pERK/GFP vs ZO-1/GFP indicated strong association. R indicates the correlation coefficient. Scale bars equal 5 µm (**b**) and 10 µm (**c**,**d**).**e** Colocalization coefficients of pERK/GFP and ZO-1/GFP are correlated. With decreased pERK, ZO-1 also reduces, similarly, high pERK expressing vascular cells also has high ZO-1 levels. GFP always marks the vasculature, and therefore is common to both separate correlations. Source data are provided as a Source Data file. Figure 5/panel a Created with BioRender.com released under a Creative Commons Attribution-NonCommercial-NoDerivs license.

subtype specific genes from around ~22,500 cerebrovascular cells. Although these studies elevate our understanding on the complexity of vascular dysfunction in AD, there remains a knowledge gap about the potential translational impact of these findings. Systematic studies that interrogate transcriptional perturbations in the gliovascular unit (GVU) of the BBB; uncover potential molecular interactions between the different cell types of the GVU; prioritize these molecules through analytic approaches; validate and replicate them using orthogonal methods and external datasets; and finally perform experimental validations in model systems are necessary. Such systematic studies are instrumental in translating big data to knowledge to high-confidence molecular targets in complex pathophysiologic events, such as BBB breakdown in AD.

Our study utilizes a systematic approach to discover, prioritize, replicate and experimentally validate GVU interacting molecular pairs

the perturbations of which may contribute to BBB disintegrity in AD. In postmortem studies, we discovered vascular transcriptional changes in AD, identified their predicted astrocytic molecular partners, prioritized pericytic *SMAD3* upregulation and astrocytic *VEGFA* down-regulation in AD for follow-up, validated these perturbations using orthogonal approaches and replicated in external brain snRNAseq datasets. In antemortem studies, we demonstrated associations of *SMAD3* locus genetic variants with both higher blood levels of this gene and lower frequency of brain infarcts, as well as correlations of high blood *SMAD3* with lower brain amyloid β and less cortical atrophy. In in vitro studies, we validated molecular interactions between VEGFA and pericytic SMAD3 by showing an inverse relationship between them through either activation or inhibition of the VEGFA pathway that decreased or increased pericytic *SMAD3*, respectively. In in vivo studies, we used well-established zebrafish models to demonstrate *VEGFA*

ortholog (*vegfaa*) astroglial downregulation upon Aβ treatment; and both activated pSMAD3 upregulation and BBB breakdown upon blockage of VEGFA/vegfaa signaling. Our systematic approach combining human post-mortem and antemortem data with cross-species model systems can be used as a roadmap in future studies to enable prioritizations of molecules from big-omics datasets down to a manageable number of targets for downstream experimental validations.

In addition to the high confidence, experimentally validated pericytic *SMAD3*-astrocytic *VEGFA* interactions and perturbations in AD with consequences in BBB disintegrity, our study also provides detailed information on human brain vascular and astrocytic transcriptional perturbations and other predicted molecular pairs which can be followed up in future studies. In our postmortem studies, using snRNAseq, from temporal cortex brain tissue of 12 AD and 12 control donors, we obtained 78,396 post-QC nuclei, of which 2153 were classified as brain vascular cells. Our snRNAseq data from brain vascular nuclei enabled the following observations. First, we identified three distinct vascular clusters which could be classified as pericytes (cl.25), endothelia (cl.26) and perivascular fibroblasts (cl.30), owing to the unique expression profiles of their highly expressed signature genes. Second, we identified DEGs in these vascular clusters; the largest numbers of which were in the pericyte cl.25 (156 up, 64 down), followed by endothelial cl.26 (34 up, 10 down) and perivascular fibroblast cl.30 (8 up, 6 down). The limited number of overlapping DEGs amongst the brain vascular clusters underscore their distinct nature. Also, having relatively higher up and down regulated DEGs indicated potential selective vulnerability of AD pericytes compared to other vascular cell types. Third, enriched GO terms amongst the vascular cluster DEGs highlight perturbed biological processes in brain vasculature. Upregulated pericyte cl.25 had many signaling molecules such as *SMAD3*[57] and *STAT3*[57,64,65], whereas downregulated genes in this cluster had cytoskeletal genes such as *DMD*[90], with enrichment for hormone receptor binding and actin-based processes, respectively. Endothelial genes (cl.26) upregulated in AD include angiogenesis related genes such as *ANGPT2*[61]. These findings demonstrate the vast transcriptional changes in brain vascular cells in AD and nominate molecules and pathways that may propagate the known BBB dysfunction and breakdown in this condition[1,91].

Given the known interactions and proximity between astrocytes and brain vascular cells, i.e. endothelia and pericytes, at the BBB[1,67], we sought to discover those molecules that have strong interactions between these key cell types of the GVU. For this purpose, we used the analytic approach of NicheNet[51], a computational method that uses prior knowledge on signaling and gene expression networks to predict ligand-target relationships of interacting cells based on their expression data. Astrocytes already have known ligands, such as *APOE*, that bind targets on pericytes with downstream signaling changes that influence pericyte function[67]. Thus, we used NicheNet[51] and our snRNAseq data to discover and prioritize brain vascular targets that are influenced by astrocytic ligands. We restricted our analyses to those genes that are significant DEGs in these cell types in our data in order to identify those vascular target-astrocyte ligand pairs that are most perturbed in AD and therefore most likely to influence BBB dysfunction.

Our astrocyte ligand-vascular target analysis revealed strong predicted interactions and 24 brain vascular target candidates most of which had biological functions involving signaling, angiogenesis, and cytoskeleton structure. We selected 6 predicted vascular target genes representing each functional category for validation of their differential expression in our snRNAseq cohort using an orthogonal gene expression measurement approach of qPCR. All 6 genes (*ANGPT2*, *AHNAK*, *ECE1*, *TSC22D3*, *STAT3*, *SMAD3*) were validated for their differential expression in AD vs. control nuclei. Additionally, all genes had positive associations with AD-related neuropathologies, consistent with their higher levels in AD brains.

Of the signaling molecules[57,64,65] identified in our study, *SMAD3* and *STAT3* also have roles in vascular function[65,67]. Both of these genes were significantly upregulated in AD brains in the pericyte cl.25 and had strong interactions with astrocyte ligands, some of which are known AD risk genes, namely *APOE*, *APP*, *PSEN1* and *MAPT*, previously shown to lead to BBB dysfunction in model systems[92]. Both *SMAD3* and *STAT3*, which have two of the highest number and strength of astrocyte ligand interactions amongst all vascular targets in our study, are signaling molecules downstream of many ligands including TGF-β and VEGFR2-binding growth-factors, respectively[93]. Crosstalk between SMAD3 and STAT3 has been demonstrated in numerous conditions[57,93], especially in cancer. To our knowledge, neither of these signaling molecules with variable functions have been investigated in human AD brains nor for their roles in BBB dysfunction in AD. We selected SMAD3 in follow-up studies since it is markedly upregulated only in AD pericytes, the most AD vulnerable vascular cell type.

NicheNet based astrocytic ligands of pericytic *SMAD3* included *VEGFA*, a pro-angiogenic factor with critical roles in both vascular and neuronal processes of human CNS[94,95]. VEGFA, has also emerged recently as a target in AD and neurodegeneration with debated roles[68–70,96,97]. While previous studies have shown that high levels of serum and cerebrospinal fluid (CSF) VEGFA are associated with increased risk of AD[97,98], recent studies suggested increased VEGFA is protective against AD[68–70,96]. Also, a pathological hallmark of AD, Aβ, binds directly to VEGF through specific domains, hampers its binding capability to its receptor, VEGFR2[99].

To validate the impact of VEGFA signaling on *SMAD3* expression levels, we differentiated AD and control derived iPSCs into pericytes[79] and either activated or inhibited this signaling via recombinant VEGF or VEGFR2 inhibitor cocktail treatments, respectively. VEGF treatment decreases *SMAD3* expression levels at longer treatment duration, and blocking VEGF signaling consistently increases *SMAD3* expression levels. This inverse relationship is reminiscent of high pericytic *SMAD3*-low astrocytic *VEGFA* we observed in AD brains and is also corroborated by the zebrafish studies.

Comparative molecular studies from zebrafish amyloidosis model and human AD patients demonstrated transcriptional similarities in their response to Aβ toxicity[100]. Ours and other snRNAseq data from human donor brains demonstrate that *VEGFA* expression is reduced in AD astrocytes[23,101]. To test the impact of Aβ on astrocytic *VEGFA* in vivo, we injected Aβ42 to zebrafish model and generated scRNAseq data. Astrocytic *vegfaa* is downregulated upon Aβ injection, which validates our human postmortem findings. We also observed an increased trend in *smad3a* expression, albeit not statistically significant likely owing to the low number of zebrafish pericytes.

We also used the zebrafish model to experimentally assess the impact of VEGFA signaling on SMAD3 and BBB integrity. We pharmacologically treated zebrafish with *vegfr2* inhibitors in vivo and analyzed activation status of Smad3 signaling and zebrafish brain vascular integrity. Blocking VEGFA signaling causes elevation in the proportion of active signaling SMAD3 molecule, i.e. pSMAD3+ endothelia and pericytes, and impairs vascular integrity. This finding demonstrates that VEGF, does not only affect pericytic *smad3* expression, but also its signaling and BBB integrity. Interestingly, we also demonstrated increased pSMAD3+ pericytes in postmortem human AD brains, supporting increases in both transcript levels and activation status of SMAD3 in AD.

To assess whether our findings in postmortem brain samples, in vitro, and in vivo could be translated to living patient samples, we explored blood gene expression and neuroimaging data from two studies of longitudinally followed older participants[75,76]. *SMAD3* genetic locus variants associated with higher blood levels of this gene and with lower frequency of brain infarcts. Furthermore, higher blood *SMAD3* levels associated with less brain amyloid and less cortical atrophy, especially in brain regions typically affected by AD.

While these blood *SMAD3* level associations per se do not prove causality, they indicate a potential role of this molecule for AD in both brain and periphery. Taken together, our findings support a model wherein VEGFA reduction and signaling in the presence of Aβ (and possibly other AD neuropathologies) lead to increased *SMAD3* levels, signaling and BBB disintegrity. The specific receptors involved and whether elevated *SMAD3* levels and signaling are detrimental to BBB integrity or represent protective/reparative responses remains to be established. Our antemortem results suggest that higher *SMAD3* levels may be protective against vascular, Aβ and neurodegenerative outcomes in AD. In contrast, blocking of *Smad3* signaling in peripheral macrophages of mouse models of amyloidosis reduced brain Aβ in both parenchyma and blood vessels[102,103] via enhanced phagocytosis of Aβ, also reducing inflammation. Future studies on both expression and signaling of SMAD3 in brain and peripheral human samples, as well as in experimental models are required to further investigate functional consequences of this molecule for AD and its vascular, Aβ and neurodegeneration-related outcomes. More broadly, our results demonstrate the utility of our experimental and analytic approach in the discovery and prioritization of AD-related genes at the GVU.

Dysfunctions in cellular interactions and signaling in the GVU are critical to understand the mechanisms underlying BBB dysfunction that contributes to AD pathophysiology[1,92]. Our study demonstrates transcriptional alterations of vascular cells and astrocytes of GVU in AD at single cell resolution and discovers target-ligand relationships between these cell types. Validation of the VEGFA-SMAD3 ligand-target pair interactions using in vitro and in vivo model systems pave the way to uncover mechanistic interactions between pericytes, endothelia, and astrocytes and their perturbations in AD.

Despite these strengths, our study also has some weaknesses and limitations. In this study, we focused on predicted interactions of brain vascular target molecules with astrocytic ligands, given their known crosstalk at the BBB[1,67]. However, it will be important to also interrogate interactions with neurons, oligodendrocytes, and OPCs. Although we focused on one interacting pair (VEGFA-SMAD3), other predicted astrocytic ligands-vascular targets will also be worth following up in future experimental studies. Additional efforts are needed to identify any binding partner(s) of SMAD3 that responds to VEGFA signaling. Furthermore, our study focused on late-stage AD cases and a single brain region that has a relatively high burden of AD neuropathology. Our discovery cohort of 24 AD and control brain donors where we conducted snRNAseq of TCX also has limited number of participants, and hence limited statistical power. To address the limitation in power and determine the applicability of our findings in *VEGFA* and *SMAD3* in other brain regions, we analyzed external datasets from multiple different brain regions[14,15,20–23] resulting in an integrated dataset of 150,664 astrocyte and 4,730 pericyte nuclei from 6 or more brain regions. In these integrated datasets, we confirmed our findings of up-regulation in AD of *SMAD3* and down-regulation of *VEGFA* in the largest pericyte and astrocyte clusters, respectively. Notably, *VEGFA* was also down in AD brains in the largest clusters from enriched astrocytic nuclei in Sadick et al.[23] (Supplementary Data 45), which was not included in our integrated analyses due to the differences in the *APOE* ε4 distribution of this dataset and their enrichment approach. Importantly, in our integrated analyses, we were able to characterize the pericyte and astrocyte subclusters with these expression changes and demonstrate their applicability in different brain regions, as well as studies of both selected vascular and unselected nuclei from AD and control brains.

In summary, we identified three distinct cerebrovascular nuclear clusters and demonstrated their transcriptional perturbations in AD, which are most pronounced for pericytes. We uncovered computationally predicted interactions between astrocytic ligands and vascular targets, which underscore potential downstream effects of transcriptional changes at the GVU. We identified target-ligand interactions for genes, including those that are well-known for AD risk, such as ECE1 and APP. We validated our selected astrocytic ligand and vascular target interaction using in vitro iPSC-derived pericyte and in vivo zebrafish models. We demonstrate associations with peripheral levels of a perturbed pericyte signaling gene, SMAD3, with AD-related outcomes in living patients. Collectively, our study provides a prioritized list of perturbed brain vascular molecules and their astrocytic partners at the GVU in AD and offers mechanistic avenues to explore for deciphering the precise molecular mechanisms of BBB dysfunction in AD.

## Methods

This study was approved by the Mayo Clinic Institutional Review Board (IRB). Additional data used in this study from the AD Knowledge Portal (https://adknowledgeportal.synapse.org) were accessed under the data usage agreement. All personally identifiable information from the donors has been removed or de-identified. Written informed consent was obtained from all participants, their qualified caregivers or next of kin.

### Human Postmortem brain data generation and analysis

**Brain donors and samples.** From the Mayo Clinic Brain Bank for Neurodegenerative Disorders, frozen post-mortem brain tissues from 12 AD patients and 12 control donors, matched for age at death and sex, were obtained (Supplementary Data 1). We also selected and received 10 additional AD and 10 control donors for immunohistochemical validation. The neuropathological diagnosis was made by a neuropathologist (DWD) according to the published criteria[104]. Total RNA from ∽20 mg collected temporal cortex (TCX) from the superior temporal gyrus region was isolated to evaluate tissue quality. RNA integrity number (RIN) was determined using RNA Pico Chip assay (Agilent Biotechnologies, 5067-1513) via Agilent 2100 Bioanalyzer, and tissues that have RI*n* > 5.5 were utilized in nuclei isolation and single nucleus RNA sequencing (snRNAseq).

**Histology and immunohistochemistry.** Neuropathologic assessment that comprises evaluation of gross and microscopic findings, as well as quantitative analysis of Alzheimer type pathology was conducted[104]. Braak neurofibrillary tangle (NFT) stage and Thal amyloid phase were assigned as previously described[105,106]. Presence of TDP-43 inclusion bodies were determined by immunohistochemistry with antibodies directed against pathological TDP-43[107] (Supplementary Fig. 1). To assess vascular disease, a summary of pathological vascular lesion scores based on the presence and number of macroscopic vascular lesions (large infarct, lacunar infarct, and leukoencephalopathy) that correlate with neuroimaging during life were used[108]. We assessed Lewy pathology in the neocortices, cingulate gyrus, transentorhinal cortex, amygdala, basal forebrain, midbrain, pons, and medulla using α-syn immunohistochemistry (NACP, 1:3000 rabbit polyclonal, Mayo Clinic antibody)[109]. Lewy pathology was staged as following: brainstem, transitional or diffuse LBD according as previously established[110]. Immunohistochemistry (IHC) was performed on paraffin-embedded sections from the hippocampus and adjacent cortices, which were placed on glass slides. We used phospho-SMAD3 antibody (Thermo Fisher, S.434.0, MA5-14936, 1/100) for this procedure. The antigen retrieval process involved steaming the slides in Citrate buffer (pH 6) for 30 min. This was done after deparaffinization in xylene and rehydration in reagent alcohol. The immunohistochemical staining process was conducted using the IHC Autostainer 480 S (Thermo Fisher) and DAKO EnVision™ + reagents (Dako). We used 3,3'-diaminobenzidine as the chromogen (Dako). Finally, the immunostained slides were counterstained with hematoxylin and then coverslipped.

**Immunohistochemistry image acquisition and analysis.** The immunostained slides were scanned at a magnification of 20x using the Aperio AT2 (Leica Biosystems) to obtain whole slide images. We

manually annotated three blood vessels per case in each sample (specifically, blood vessels located in the entorhinal cortex and the adjacent white matter) using the Aperio ImageScope software (Leica Biosystems, ver 12.4.2.7000). With a custom-designed color deconvolution algorithm, we identified the immunopositive pixels and determined the proportion of the immunoreactive area, expressed as a percentage of the total area within the annotated region[74].

**Immunofluorescence.** We selected two cases each of Alzheimer's Disease (AD) and control samples from the Columbia University Brain Bank. IHC was performed on paraffin-embedded tissue slides. SMAD3 (Thermo Fisher, E.980.9, MA5-14939, 1:500), PDGFRB (Thermo Fisher, PR7212, MA5-28128, 1:500), VEGFA (R&D Biosystems, VG1, MAB2932-100, 1:500), and GFAP (Thermo Fisher, OPA1-06100, 1:500) were used. Deparaffinization and hydration steps were performed in xylene and alcohol, respectively. The antigen retrieval was done by using citrate buffer (pH:6.0) in pressure cooker for 18 min. Sections were washed in PBST and blocked in 10% normal goat serum for 30 min. Sections were then incubated with two primary antibody combinations (SMAD3-PDGFRB and VEGF-GFAP) overnight at 4 °C in a humidified chamber. Sections were again washed with PBST, and the secondary incubation was done with secondary antibodies. Each secondary antibodies applied respectively with 30 min incubation and three times washing. Slides were covered by mounting medium with DAPI.

**Image acquisition and analysis.** The images of immunostained slides were acquired using a Zeiss fluorescent microscope equipped with ZEN software (version blue edition, v3.2, Carl Zeiss, Jena, Germany). Images were analyzed and quantifications were performed on z-stacks images by ImageJ version 2.1.0/1.53c. To compare the two groups, a two-tailed t-test was performed, and GraphPad Prism software version 9.2.0. was used for the statistical analyses.

**Nuclei isolation.** Single nuclei suspensions were collected from human temporal cortex. Nuclei isolation was performed using an established protocol with adaptations[111]. 100 mg tissue was directly transferred from dry ice to dounce homogenizer containing homogenization buffer (0.25 M sucrose, 25 mM KCl, 5 mM MgCl2, 20 mM tricine-KOH, pH 7.8, 1 mM DTT, 0.15 mM spermine, 0.5 mM spermidine, protease inhibitors, 5 μg/mL actinomycin, 5 u/μL recombinant RNAase inhibitor, and 0.04% BSA). Twenty-five strokes with loose and tight pestle were sequentially performed. After strokes with tight pestle, 5% IGEPAL (Sigma, I8896) solution was added to reach a final concentration of 0.32%. Ten additional strokes were performed, and homogenate was filtered through 30 μm cell strainers. Filtrated homogenate was centrifuged (500 g, 5 min), and washed once with wash and storage buffer (1X PBS with 2% BSA and 5 U/μL recombinant RNAase inhibitor (Takara Bio, 2313 A). After washing, homogenate was filtered again through 30 μm cell strainer and centrifuged for 10 min at 500 g. The pellet was re-suspended in 700 μL cold PBS with 5 U/μL RNAse inhibitors. 300 μl Debris removal solution (Miltenyi Biotech, 130-109-398) was added, and the solution was gently mixed. The solution was carefully overlaid with 1 mL wash and storage buffer (WSB) and centrifuged for 10 min at 3000 g. Supernatant was removed, and the pellet was washed with WSB and centrifuged for 10 min at 1000 g.

**Flow cytometry and nuclei sorting (FANS).** Isolated nuclei were incubated with mouse anti-Human Nuclear Antigen (Abcam, 235-1, ab191181, 1/200) antibody for 1 h on ice. Mouse IgG1, kappa monoclonal isotype control (Abcam, 15-6E10A7, ab170190, 1/200) was included in the staining. Nuclei were incubated for 30 min on ice in secondary antibody solution that contains 1:200 goat anti mouse Alexa488 secondary antibody (Abcam, ab150113, 1/200). Nuclei were then resuspended in 200 μL WSB and sorted into WSB via BD FACSAria

II sorter, using the 70-micron nozzle with 70 psi sheath pressure and 1.5 ND filter. Our sorting strategy is shown in Supplementary Fig. 28.

**Quality control of isolated nuclei.** To assess the purity of the sorted nuclei, both RNA and protein profiles were analyzed. RNA was isolated via Qiagen RNeasy Mini Kit (Qiagen, 74004). In RNA level, disappearances of 18 S and 28 S rRNA peaks in Bioanalyzer (Agilent Technologies, 5067-1511) histogram were analyzed to confirm lack of cytoplasmic RNA contaminants in the nuclei preparations. Nuclear H3 (Abcam, Y47, ab32356, 1/200) and mitochondrial COX4 (Abcam, mAbcam33985, ab62164, 1/200) protein ratios were checked via western blot to confirm nuclear purity in protein level. Also, to confirm the preparation method does not cause bias in favor of a certain cell type, qPCR was performed with probes against RNU2.1 (Nuclear probe), AQP4, CD34, P2RY2, RBFOX3, and MOG (Supplementary Fig. 3). Nuclei integrity was checked by microscope with 20X objective of EVOS Cell Imaging System (Thermo Fisher) and through Z-stack images of the sorted HNA-Alexa488 labeled samples captured with a Plan-Apochromat 100x/1.4 Oil objective on a LSM880 Laser Scanning Confocal Microscope (Carl Zeiss Microscopy), using 488 nm Argon laser excitation and capturing 500-550 nm emission. A single example image plane from the isolated nuclei images is shown in Supplementary Fig. 2.

**10X cDNA library production and snRNAseq.** To quantify the number, sorted nuclei were stained with 0.04 % trypan blue and counted in a hemocytometer. Total nuclei solution was diluted to 1000 nuclei/μL. A total of 3000 estimated nuclei per sample were loaded and single cell gel beads-in-emulsion (GEMs) were generated on Chromium Controller (10X Genomics). Single cell RNAseq libraries were prepared using the Chromium Single Cell 3' Gel Bead and Library Kit v3 (10X Genomics, 120237) and the Chromium i7 Multiplex Kit (10X Genomics, 120262) according to the manufacturer's instructions. Library quality was checked using High Sensitivity DNA Kit (Agilent Technologies, 5067-1504).

DNA libraries were sequenced at the Mayo Clinic Genome Analysis Core (GAC) using the Illumina HiSeq4000 sequencer. Two samples were run on each lane of one flow cell and two flow cells were used in total. Samples were randomized prior to sequencing.

**Read alignment and quality control.** Cell Ranger Single Cell Software Suite (10X Genomics, v3.1.0) was used to demultiplex raw base call files generated from the sequencer into FASTQ files. Raw reads were aligned to human genome build GRCh38 and a premature mRNA reference file. Reads aligned to gene transcript locus, including both exonic and intronic regions, were counted to generate raw UMI counts per gene per barcode for each sample. The raw UMI matrices were filtered to only keep barcodes with ≥ 200 UMIs and those that were called a 'cell' by Cell Ranger's cell calling algorithm. The filtered barcodes from all 24 samples were pooled together and further filtering criteria were applied to exclude the following barcodes and genes. 1) barcodes with > 10% of UMI mapped to mitochondrial genome; 2) barcodes with <400 or > 8000 detected genes; 3) barcodes with <500 or > 46425 mapped UMIs; 5) genes that are detected in <5 cells (Supplementary Fig. 4). The above thresholds were determined by UMI or gene distribution to identify undetectable genes and outlier barcodes that may encode background, broken or multiple cells. 1355 doublets were performed using Scrublet[112] and were subsequently removed. Next, we extracted protein coding genes for further analysis. Recorded sex of samples was compared to the sex inferred from chromosome Y gene expression, which confirmed the correctness of sex information for all samples.

**Clustering nuclei.** After quality control, UMI counts of remaining cells and genes were normalized using NormalizeData function in R

package Seurat[113] v3.1.0, which gave natural log transformed expression adjusted for total UMI counts in each cell. The top 2000 genes whose normalized expression varied the most across cells were identified through FindVariableFeatures function with default parameters. Using those genes, cells from eight groups of samples (grouped by AD/normal, male/female and *APOEε4* positive/negative) were integrated using functions FindIntegrationAnchors and IntegrateData with default parameters. Principal components (PCs) of the integrated and scaled data were computed; and the first 31 PCs, which accounted for > 95% variance, were used in clustering cells. Cell clustering was performed using FindNeighbors and FindClusters with default parameters. All analyses described in this section were performed using Seurat v3.1.0.

**Identifying cluster marker genes and assigning cell types of each cluster.** Marker genes that were conserved in both AD and control nuclei were identified in each cluster using FindConservedMarkers in Seurat v3.1.0. Marker genes of one cluster must 1) be present in > 20% AD nuclei and > 20% control nuclei of the cluster; 2) the log(fold change) between their expression in AD (control) cells of this cluster and AD (control) cells of other clusters must be > 0.25; 3) the rank sum test p-value (Bonferroni adjusted) between AD (control) cells in this cluster and AD (control) cells in other clusters <0.05.

Two approaches were adopted and combined for cell type assignment. The first one utilized the marker gene lists reported in R BRETIGEA[31] for neurons (1000 markers), astrocytes (1000 markers), oligodendrocytes (1000 markers), microglia (1000 markers), endothelial cell (1000 markers) and OPCs (500 markers). Hypergeometric tests were performed for over-representation of our cluster markers in those reported markers. Each cluster was assigned one cell type that was most over-represented. The second approach was to check the existence of a handful of well-recognized cell type markers in top cluster markers. Those cell type markers are *SYT1, SNAP25, GRIN1* for neuron; *SLC17A7, NRGN* for excitatory neuron; *GAD1, GAD2* for inhibitory neuron; *VCAN, PDGFRA, CSPG4* for OPC, *MBP, MOBP, PLP1* for oligodendrocyte; *C3, CSF1R, CD74* for microglia; *AQP4, GFAP* for astrocyte; *FLT1, CLDN5* for endothelial cells; and *PDGFRB*, for pericytes. Combining the two approaches and scType[114], we assigned the following eight cell types/subtypes to each cluster - excitatory neuron, inhibitory neuron, oligodendrocyte, OPC, microglia, astrocyte, endothelia, pericytes and perivascular fibroblasts.

**Cell distribution association test.** For each cluster, the number of cells in an individual for that cluster, was divided by the total number of cells in all clusters for that individual. The resulting ratio gives the cell distribution that was used to test for association with characteristics using a Wilcoxon rank sum test for binary variables (AD vs. control, male vs. female, *APOEε4* positive vs. negative, and TDP-43 positive vs. negative) or Spearman's test of correlation for quantitative/semi-quantitative variables (age at death, Thal phase, and Braak stage). All statistical tests were two-sided. P-values < 0.05 were considered statistically significant.

**Differential expression and association analysis for each cluster.** For each cluster, we performed differential expression analysis for genes that were detected (UMI > =1) in >= 10% AD cells or >= 10% normal cells using R package MAST[49]. MAST employed a hurdle model to accommodate the so-called 0-inflation observed in scRNAseq/snRNAseq data, i.e., many cells had 0 UMI for a given gene. In the models depicted in Supplementary Data 44, AD cells were coded as 1, normal cells were coded as 0; males were coded as 1, females were coded as 0; *APOEε4* positive (44 or 24 or 34) were coded as 1, *APOEε4* negative were coded as 0; TDP-43 positive were coded as 1, TDP-43 negative were coded as 0; age, Braak stage, and Thal phase were numerical variables.

For selecting DEGs or genes associated with continuous variables from each cell cluster, we focused on the set of genes that were detected (UMI > 0) in at least 20% of the AD cells or of the normal cells in the cluster and have q < 0.05. The DEGs between binary variable (including AD and normal, male and female, *APOE4* positive and *APOE4* negative, or TDP-43 yes and TDP-43 no) for each cluster were the ones that have q < 0.05 and |logFC| > 0.1.

**Signature genes of clusters.** Signature genes of a cluster are genes that are highly expressed in one cluster of a cell type but not the other clusters of that cell type such that a) they are present in >= 50% cells of this cluster, b) average log2 fold change >= 1.0 and c) Wilcoxon rank sum test Bonferroni p-value < 0.05 when compared to each of the other clusters. FindMarker function of Seurat was applied to obtain such signature genes. Signature genes for vascular and astrocyte clusters were both determined as above.

**Enrichment of genes in MSigDB GO terms.** MSigDB v7.0 was used for Gene Ontology enrichment analyses. The enrichment of selected genes in MSigDB C5 category (i.e., gene ontology or GO) was performed using R enRichment package. The top 5 enriched GO terms and top 5 genes that occur most frequently in these terms were plotted for the main figures.

**Constellation plot.** This analysis used the script from Olah et al.[16]. which was used to generate the constellation plot. For every pair of clusters, cells from the two clusters were randomly divided into four groups. For each group, cells in the other three groups were used as training data, and the cells of this group were classified to be from one of the two clusters. This classification procedure was repeated 100 times and therefore each cell was classified 100 times. If a cell was misclassified > 25 times, it was considered as "ambiguous" or "intermediate". The percent of intermediate cells was calculated as 100* (num of intermediate cells)/(num.cell.clusterA + num.cell.clusterB).

**Ligand-target analyses.** NicheNet[51] analysis tool was used to study the interaction between astrocytic and vascular cells through NicheNetr R package. Prior knowledge of ligand-target interaction has been compiled and optimized by NicheNet from multiple data sources to give a prior model which contains the regulation strength of ligands towards target genes. In this study, we set astrocyte DEGs between AD and control brain cells from cl.8, cl.11, and cl.31 as potential ligands and DEGs from vascular clusters cl.25, cl.26, and cl.30 as target genes. The following description uses pericyte cl.25 as an example. Among the potential ligands, we identified genes satisfying the following: a) It is a ligand according to prior model; b) It has receptor genes expressed in the target cluster pericyte cl.25. For the resulting set of ligands, we identify the target genes satisfying that a) It is a DEG of pericyte cl.25 and b) It is among the top 250 regulated genes by one of the ligands according to prior model.

In this manner, we obtained cl.8-cl.25, cl.11-cl.25 and cl.31-cl.25 targets interacting with ligands in the aforementioned astrocyte cluster ligands. The union of these three sets of targets gives 22 target genes in pericyte cl.25. Using a similar approach, we identified 4 target genes from cl.26 and 2 target gene from cl.30. In addition, we noticed that gene *MALAT1* is a DEG in almost all clusters and removed it from the target gene.

**External snRNAseq datasets.** To increase the number of pericyte and astrocyte nuclei and the number of participants, we integrated pericyte and astrocyte nuclei of external datasets with ours, performed subclustering, and differential expression (DE) analysis of *SMAD3* or *VEGFA* in each subcluster. Supplementary Data 31 lists these datasets, namely Is et al. (this study), Grubman et al.[15], Mathys[14], Yang et al.[22], Mathys 2023 PFC region[20], Mathys 2023 multi-region[20], Sun et al.[21], and

Zhang et al.[23]. (GSE188545). All studies, except Zhang et al.[23]. have published post-QC, post-cell assignment data and meta data. In order to avoid any bias in our analyses, we used these post-QC, post cell type assignment data associated with the original publications. Mathys et al. 2023/multi-region[20] and Sun et al.[21] shared the dataset as indicated at compbio.mit.edu/ad_aging_brain/, albeit Sun et al.[21] mainly focused on analyses of vascular cell types (compbio.mit.edu/scADbbb/). Therefore, we only included Sun et al.[21] in pericyte related analyses, and Mathys et al.[20]/multi-region in astrocyte related analyses.

We performed QC and cell type assignments for the Zhang et al.[23] study as this was not available. For this study, the filtered UMI counts for each feature and each barcoded data were download from GSE188545[23]. We filtered out cells that contain less than 500 UMI, or greater than 32677 (the 98 percentile) UMI, or >10% UMI from mitochondria genome, or less than 400 genes, or more than 7136 genes. We filtered out genes that were in <5 cells and kept protein coding genes. These thresholds were determined either by distribution or consistent with those applied to our snRNAseq dataset. 3224 doublets were identified by Scrublet[112] and were removed. After QC and filtering steps, we retained 17,946 protein coding genes, and $6,120 \pm 3,109$ (mean $\pm$ standard deviation) cells. Next, we performed SCT transformation v2[115] for each sample, selected the top 2,000 most variable features, computed the first 50 principal components (PCs), and used the top 35 PCs, which accounted for $\geq$ 95% variance, to run Harmony[116] to integrate cells from each sample. Next, we utilized the integrated data to find neighbors and find clusters with resolution 0.5. 32 cell clusters were identified and were assigned cell types according to their marker gene expression. Cluster 28 is the pericyte cluster while cluster 3, 15 and 31 are the astrocytic clusters for the Zhang et al.[23] study.

**Integrating external snRNAseq datasets.** We took pericyte nuclei where this was available, i.e. from Is et al (this study), Sun et al.[21], Zhang et al.[23], Yang et al.[22] and Mathys[14], performed SCT transformation v2, integration using Harmony and cell clustering using Seurat function FindNeighbors and FindClusters. We noticed that the number of nuclei in each participant in Yang et al.[22] is much greater than that in other studies, as Yang et al enriched vascular nuclei using VINE-seq method[22]. We also noticed that in Sun et al.[21], PFC was over-represented compared to other brain regions with ~4000 pericytes from 375 participants (Supplemental Data 31). Therefore, we down-sampled nuclei as follows: For Sun et al.[21], we randomly selected 25 AD and 25 control participants and included all nuclei for them. For Yang et al, 400 pericytic nuclei were randomly selected from each brain region, i.e. SFX and HC, with even numbers in each participant. Given the increased number of participants in this integrated analysis, we applied negative binomial generalized linear mixed effects model for differential expression analysis[117]. R package glmmTMB was used for this analysis, where the participants and studies were coded as random effects, diagnosis, age at death and sex were coded as fixed effects. Numbers of and testing for over-representations of nuclei from donor(s) in each cluster in integrated astrocyte and pericyte integrated datasets are shown in Supplementary Data 42, 43, respectively.

We integrated astrocyte nuclei of external datasets with ours, performed clustering, and DE analysis of *VEGFA* in each cluster. Mathys 2023[20] multi-region study already included astrocyte nuclei from PFC region of 41 participants. Therefore, we didn't include Mathys 2023 PFC data in this integrated analysis, which encompassed 149,558 astrocytes from 427 participants that would overweigh the PFC region data. We performed SCT transformation v2, integration using Harmony and cell clustering using Seurat function FindNeighbors and FindClusters. Mixed effect models implemented in R package glmmTMB were used to perform differential expression analysis.

**Gene expression validation via RT-qPCR.** Total RNA was extracted from sorted nuclei using the miRNeasy Serum/Plasma Kit (QIAGEN;

217184). The Agilent BioAnalyzer RNA 6000 Pico Kit (Agilent; 5067-1514) was used to assess RNA concentration and quality. RNA was normalized to 0.5 ng/µL for cDNA synthesis using the SuperScript IV VILO Master Mix (ThermoFisher; 11756050). TaqMan PreAmp Master Mix (ThermoFisher; 4391128) was used to pre-amplify cDNA, followed by TaqMan Universal PCR Master Mix (ThermoFisher; 4304437) with the following gene expression probes: MOG, AQP4, RBFOX3, P2RY12, CD34, ANGPT2, AHNAK, ECE1, SMAD3, STAT3, TSC22D3, GAPDH, RNU2-1 (ThermoFisher; Hs01555268_m1, Hs00242342_m1, Hs01370654_m1, Hs00224470_m1, Hs00375822_m1, Hs00169867_m1, Hs01043735_m1, Hs00969210_m1, Hs00374280_m1, Hs00608272_m1, Hs99999905_m1, Hs03023892_g1). RT-qPCR was performed on a QuantStudio 7 Flex Real-Time PCR System (ThermoFisher). Comparative CT analysis ($\Delta\Delta$CT) was used to quantify gene expression with RNU2-1 used as the endogenous reference and brain homogenate as the calibrator. Two-sided Wilcoxon rank sum tests were performed to test whether these genes were expressed higher in AD compared to control nuclei. Validation was performed on 20/24 participants with sufficient tissue (Supplementary Data 27).

**Gene expression validation via RNAscope assay.** Nuclei were extracted from and purified from 50-100 mg of frozen human superior temporal gyrus from 9 AD and 9 control samples as previously described. 50,000 nuclei were seeded on poly-D-lysine (Thermo Fisher, A3890401) coated 96-well PhenoPlate Plates (Perkin Elmer, 6055302). Plates were centrifuged for 5 min at 500 g. Nuclei were fixed with 4% formaldehyde in PBS for 30 min. DAPI was used to mark and visualize the isolated nuclei samples. RNAscope Multiplex Fluorescent v2 kit (ACD Biotech, 323100) was used to stain the nuclei with selected probes following manufacturers recommendations. Following ACD RNAscope probes were used: *LEF1* (412991-C2), *SMAD3* (404241), *AGT* (459131), and *VEGFA* (423161-C2). Images were captured on Operetta CLS High Content imaging system through confocal mode under 20x objective. Cell Profiler (version 4.2.5) custom pipeline was established to relate and assign the RNAscope dots to respective nuclei. Positivity of nuclei staining were defined as having $\geq$1 assigned dot for each staining condition.

**Antemortem association of blood *SMAD3* levels with genetic variants and neuroimaging phenotypes**
Data used in the preparation of this article were obtained from the Alzheimer's Disease Neuroimaging Initiative (ADNI) database (adni.loni.usc.edu)[76,118] and Mayo Clinic Study of Aging[75,119]. The ADNI was launched in 2003 as a public-private partnership, led by Principal Investigator Michael W. Weiner, MD. The primary goal of ADNI has been to test whether serial magnetic resonance imaging (MRI), positron emission tomography (PET), other biological markers, and clinical and neuropsychological assessment can be combined to measure the progression of mild cognitive impairment (MCI) and early Alzheimer's disease (AD). The MCSA was launched in 2004 and is led by Principal Investigator Dr. Ronald C. Petersen. ADNI and MCSA data were used to evaluate the association of genetic variants within *SMAD3* locus with neuroimaging (NI) phenotypes and blood *SMAD3* levels, genotype data, blood gene expression and neuroimaging phenotypes available from the Alzheimer's Disease Neuroimaging Initiative[76,118] (ADNI) and from the Mayo Clinic Study of Aging[75,119] (MCSA) were used.

**Genotype data.** Following approval, genetic data available for ADNI participants was obtained through the Laboratory of Neuroimaging (LONI) Image & Data Archive (IDA). Genotypes from participants in two ADNI cohorts namely ADNI WGS ($n = 808$) and the non-overlapping ADNI2/GO GWAS ($n = 361$) were obtained in the form of VCF and PLINK files, respectively. While ADNI WGS genotypes were derived from Illumina Omni 2.5 M (WGS Platform) and subsequent variant calling

and genotyping with GATK, the ADNI2/GO GWAS genotypes were derived from Illumina HumanOmniExpress BeadChip array.

For the MCSA participants, genome-wide genotypes were generated for study participants in two batches, batch A (n = 528) and batch B (n = 1081), using the Infinium Omni2.5 Exome8 array v1.3 consisting of 2,612,357 SNPs (A) or v1.5 consisting of 2,617,655 SNPs (B) and exported to a comma-separated final report file using Illumina's GenomeStudio software v1.9.4 and v2.0.4, respectively. Final report files were converted to PLINK[120] (v1.9) formatted lgen, fam, and map files using in-house scripts.

**QC of genetic data.** Duplicate variants were evaluated for missingness and those with the best genotyping rate were retained. Variants with a genotyping rate equal to or greater than 98% and a minor allele frequency (MAF) of 2% or more were retained. Samples with a genotyping rate less than 98% or having discordant sex or those with a PLINK heterozygosity estimate (F) beyond three standard deviations ($\mu(F) \pm 3sd$) were excluded. One sample from each pair or family of related samples (PLINK PI_HAT > 0.125), with the best call rate was retained. Population outliers were excluded using Eigenstrat[121,122] which was set to remove outliers of up to 6 standard deviations of the top 10 principal components (PCs) over five iterations, while refitting PCs after each iteration of outlier removal. Given that the MCSA batches were genotyped on the same platform, samples and variants were merged after QC and any relatedness among the merged set was resolved and PCs for population substructure were recalculated. These merged genotypes were then utilized for imputation. Since the ADNI cohorts were genotyped on different platforms, utilizing a common set of variants, relatedness among the ADNI cohorts was resolved and PCs recalculated. Samples that were retained in ADNI WGS and ADNI2/GO GWAS cohorts were imputed separately but combined after imputation to have a common set of variants for analysis. In summary, 1508 subjects and 1,393,625 variants passed QC in combined MCSA cohort, 755 samples and 2,0375,599 variants passed QC in ADNI WGS and 325 subjects and 629,732 variants passed QC in ADNI2/GO GWAS.

**Imputation of genotypes.** Prior to imputation, variant strand, position and alleles were aligned to the HRC reference panel[123] using tools provided by the McCarthy Group (https://www.well.ox.ac.uk/~wrayner/tools/). Genotypes were uploaded to the Michigan Imputation Server[124] and run in "QC only" mode to identify and remove variants with mismatched allele frequencies. Genotypes were then imputed to the HRC (r1.1.2016) reference panel with Eagle (v2.3) phasing[125]. Since imputation replaces genotypes with imputed doses, original genotypes were reinserted back into the VCFs using in-house scripts. Dosages were then exported from the VCF using PLINK (v2.00a3LM). Since the ADNI cohorts were imputed separately, only variants with an imputation R2 ≥ 0.7 and a minor allele frequency (MAF) ≥ 2% in both cohorts were retained. In summary, 6,899,321 variants in MCSA and 6,644,298 in the combined ADNI cohorts with an imputation R2 ≥ 0.7 and a MAF ≥ 2% were retained for downstream analysis. Variants were annotated using ANNOVAR[126]. Within the post-QC samples (1,508 from MCSA and 1,080 from ADNI) and genotypes, 588 genetic variants in the *SMAD3* locus were analyzed for association with blood *SMAD3* gene expression (PaxGene) and neuroimaging infarct phenotypes in each cohort and also using meta-analysis.

**PAXgene RNAseq.** Blood PaxGene RNAseq data was available for 395 MCSA participants. Whole blood was collected in PAXgene Blood RNA tubes and RNA was isolated using the PAXgene Blood RNA kit PreAnalytiX (Qiagen, 762164) per manufacturer's protocol. RNA was further purified following the RNA Clean & Concentrator Kit (Zymo Research, R1013/R1014) manufacturer's protocol. Purified RNA was quantified using the Nanodrop 2000 and quality assessed using the

RNA 6000 Nano kit (Agilent Technologies, 5067-1511) run on the 2100 BioAnalyzer, following Agilent reference pamphlet.

The quality and quantity of double DNase I-treated total RNA were initially assessed using Qubit fluorometry (Invitrogen) and the Fragment Analyzer (Agilent). Two micrograms of the RNA underwent rRNA removal using Illumina's Globin-Zero Gold rRNA Removal Kit according to the manufacturer's instructions for the TruSeq Stranded mRNA Sample Prep Kit (Illumina, 20020595). The concentration and size distribution of the completed libraries was determined using an Agilent Bioanalyzer DNA 1000 chip (Agilent Technologies, 5067-1504) and Qubit fluorometry. Libraries were sequenced at six samples per lane following Illumina's standard protocol using the Illumina cBot and HiSeq 3000/4000 PE Cluster Kit. The flow cells were sequenced as 100 ×2 paired end reads on an Illumina HiSeq 4000 using HiSeq 3000/4000 sequencing kit and HCS v3.4.0.38 collection software. Basecalling was performed using Illumina's RTA version 2.7.7.

**QC of paxGene RNAseq data.** Raw paired-end reads were processed through MAP-RSeq pipeline v3.0[127]. MAP-RSeq removed reads of low base-calling Phred scores, aligned remaining ones to reference human genome build GRCh38 using STAR aligner v2.5[128], counted reads in genes and exons using featureCounts[129] in subread v1.5[130]. It obtained QC measures from both pre- and post-alignment reads using RSeQC toolkit[131] and fastQC[132]. Subsequently, we identified and excluded outlier samples of low mappability, or of discrepancies between estimated strandedness and know strandedness, or of disconcordance between recorded sex and estimated sex. Further, samples for which the principal components 1 or 2 were outside the mean +/− 4*SD were excluded. Raw RNA read counts were normalized using R package CQN[133], which generated library size, gene length, and GC content adjusted expression values in log2 scale. Based on the bimodal expression distribution, genes with median CQN values less than 1 were considered lowly expressed and filtered out.

**Blood *SMAD3* eQTL and brain imaging infarct associations.** To evaluate the association of *SMAD3* genetic variants with brain infarcts or with *SMAD3* blood expression values, 588 variants within a 1 Mb window of *SMAD3* were extracted from both ADNI and MCSA cohorts and tested using generalized linear models (infarcts) or linear mixed models (eQTL) in R v4.0.3. The presence or absence of infarcts were encoded as a binary phenotype, 1 representing presence and 0 absence. Infarcts were identified from brain MRI data according to extensively standardized methods[134]. The presence or absence of infarcts on MRI were detected from T2-weighted images by experienced readers using the last MRI examination for each participant. A total of 1,508 MCSA and 1,080 ADNI participants were analyzed for associations of infarcts with *SMAD3* locus variants. A subset of 395 MCSA participants with PAXgene blood *SMAD3* gene expression measures from RNAseq and 645 ADNI participants with PAXgene blood RNA expression quantified using Affymetrix Human Genome U219 Array (Affymetrix)[78] were utilized for eQTL analysis. *SMAD3* expression in the ADNI cohort was quantified with 5 probes. Correlation between these probes as well as the average expression across all 5 probes is shown in Supplementary Fig. 17. Of the 5 probes that measure transcripts of *SMAD3* in ADNI, probes 'p11754091_s_at' and 'p117118266_s_at' were most correlated with each other (Pearson r = 0.71) and with the average *SMAD3* expression across all 5 probes (0.79 and 0.81, respectively) (Supplementary Fig. 17). Variant dosages were tested for association with infarcts while adjusting for age (at time of neuroimaging), sex, batch and the first three principal components (PCs) accounting for population substructure. To identify eQTL, variant dosages were tested for association with *SMAD3* gene expression values derived from RNAseq (MCSA) or from array expression of each *SMAD3* probe, while adjusting for diagnosis, (encoded as a binary variable, 0 representing cognitively normal

controls and 1 representing subjects with mild cognitive impairment=MCI or AD), age (at time of PaxGene collection), sex, batch, RIN, flowcell/plate and the first three PCs. Flowcell (MCSA) or Plate (ADNI) was encoded as a random effects variable while all other covariates were treated as having fixed effects in the linear mixed model. Primary model also included allelic dosages for both *APOE* ε2 and ε4 (MCSA) or just *APOE* ε4 (ADNI). Secondary models were run excluding *APOE* or after sex or *APOE* ε4 stratification. Genetic associations with brain infarcts and with *SMAD3* gene expression in ADNI and MCSA were meta-analyzed in the PLINK v1.9 to obtain random effects beta, p-values and an estimate of heterogeneity (Q and $I^2$).

**Association of blood SMAD3 levels with brain amyloid β and cortical thickness.** There were 638 participants from ADNI with blood PaxGene microarray *SMAD3* expression, amyloid β (Aβ) positron emission tomography (PET) scan and magnetic resonance imaging (MRI). We performed whole-brain association analysis of blood *SMAD3* levels with brain Aβ deposition and cortical thickness, as previously described[135]. Briefly, an automated MRI analysis technique (FreeSurfer V5.1) was used to process T1-weighted structural MRI scans[134]. [18 F] Florbetapir PET scans for brain Aβ measurement were pre-processed as described previously and were intensity-normalized by the whole cerebellum[136]. The normalization yielded standardized uptake value ratio images. Gene expression profiling from peripheral blood samples collected using PAXgene tubes for RNA analysis was performed on the Affymetrix Human Genome U219 Array (Affymetrix). The processing and QC of the microarray expression for blood RNA samples was described previously[135]. All probe sets were mapped and annotated with reference to the human genome (hg19). After QC including for sex discrepancies, 21,150 expression probes remained. We used the average gene expression from 5 probes for *SMAD3* in the brain Aβ and cortical thickness association analyses as follows:

Multivariable analysis of cortical thickness and Aβ accumulation was performed to examine effects of blood *SMAD3* gene expression levels on vertex-by-vertex and voxel-by-voxel bases, respectively. In MRI scans, the cortical thickness was calculated by taking the Euclidean distance between the gray and white boundary and the gray and CSF boundary at each vertex on the surface[137]. The SurfStat software package (www.math.mcgill.ca/keith/surfstat/) was used to perform a multivariable analysis of cortical thickness on a vertex-by-vertex basis using a general linear model (GLM) approach, using age, sex, years of education, MRI field strength, and total intracranial volume as covariates. The processed [18 F]Florbetapir PET images were used to perform a voxel-wise statistical analysis across the whole brain using SPM12 (www.fil.ion.ucl.ac.uk/spm/). We performed a multivariable regression analysis using age and sex as covariates. Adjustment for multiple comparisons was performed using the random field theory (RFT) correction for whole brain surface-based analysis and FDR correction methods for whole brain voxel-based analysis[138].

## Human In Vitro data generation and analysis

**Human iPSC lines and pericyte differentiation.** Two fully characterized AD- and two control patient-derived iPSCs were kindly provided by Mayo Clinic Center for Regenerative Biotherapeutics (Supplementary Data 40). These cells were fully characterized previously[80,81,83] and validated for pluripotency and ectodermal differentiation capability (Supplementary Fig. 18). Mycoplasma contamination in iPSCs were checked via MycoAlert® PLUS Mycoplasma Detection Kit (Lonza, LT07-710) and compared our readouts with control samples from MycoAlert® Assay Control Set (Lonza, LT07-518) (Supplementary Data 41). iPSCs were maintained in mTesR1 medium (Stem Cell Technologies, 100-0276) on Matrigel (Corning, 354277) coated plates. All iPSCs were passaged when the lines reached 70 % confluency by either manual selection of healthy colonies or ReLeSR (StemCell Technologies, 05872). Pericyte differentiation was adapted from previous studies and

applied with slight modifications[79]. Prior to differentiation, iPSCs were passaged with Accutase (StemCell Technologies, 7920) and plated onto 6-well plates with mTeSR1 medium supplemented with 10 µM Rock Inhibitor Y27632 (Stem Cell Technologies, 72302) and plated at a density of 40,000 cells/cm². On the first day, the cells were washed with 1x PBS and maintained in differentiation N2B27 medium (1:1 DMEM/F12 + neurobasal medium, B27 and N2 supplements, penicillin, and streptomycin, and Beta-Mercaptoethanol) supplemented with 25 ng/mL of BMP4 (R&D Biosystems, 314-BP-050) and 8 uM CHIR99021 (R&D Biosystems, 4423/10) for three days. On both days 4 and 5, medium was changed to fresh N2B27 media supplemented with 2 ng/mL of Activin A (R&D Biosystems, 338-AC-010) and 10 ng/mL PDGF-BB (Stem Cell Technologies, 78097). On day 6, pericytes were passaged with Accutase, plated onto a new matrigel coated plate at a density of 18000 cells/cm², and cultured in N2B27 medium for 6 days. Medium is changed in every 2 days. On day 12, pericytes were passaged via Accutase and seeded for treatment experiments onto 24-well plates with the density of 50,000 cells/well.

**Validation of pericyte differentiation.** Pericyte differentiation was validated through flow cytometry, immunocytochemistry (ICC), and RT-qPCR. Pericytes were incubated with primary antibodies Anti-NG2 (BD Pharmingen, 554275, Clone 9.2.27, 1/300), Anti-PDGFRB (R&D Systems, MAB1263, PR7212, 1/300) and Anti TRA1-60 (Abcam, ab16288, 1/300) in blocking buffer (1x PBS, 0.5% BSA, 2% FBS, and 3 mM EDTA) for 1 h on ice. Cells were incubated for 30 min on ice in secondary antibody solution that contains goat anti-mouse Alexa488 secondary antibody (Abcam, ab150113, 1/200). 7-AAD (Sigma, A1310, 1/100) was used to stain live/dead cells. FCS files were acquired through Attune NxT Flow cytometer (Life Technologies) and processed and gated in FlowJo (BD Biosciences, v10). For ICC, cells were seeded and fixed with 4% paraformaldehyde on 96-well PhenoPlate Plates coated with matrigel (Perkin Elmer, 6055302). Then, cells were washed with 1X DPBS; permeabilized in 0.2% Triton X-100 in DPBS for 10 min at room temperature; blocked in blocking solution (1% BSA in DPBS containing 0.01% Triton X-100) for 1 h at room temperature; and stained with Anti-PDGFRB (R&D Systems, MAB1263, PR7212, 1/100) and Anti-Actin (Thermo Fisher, MA511869, ACTN05 (C4), 1/200) antibodies in blocking solution for 1 h. Anti-mouse Alexa488 secondary antibody (Abcam, ab150113, 1/00) was used for secondary staining and DAPI was used to stain nuclei. Images were captured on Operetta CLS High Content imaging with 20X objective. For RT-qPCR, RNA was isolated from iPSCs and differentiated pericytes, reverse-transcribed to cDNA through RT reaction and used in RT-qPCR as previously described. Brain cell type marker gene expression between pericytes and iPSCs were compared through comparative CT analysis method with GAPDH used as the endogenous reference. The purity of pericytes was also checked via qPCR after differentiation with a panel of comprehensive qPCR probes that include several brain cell type markers. (Supplementary Fig. 29).

**Treatment of pericytes with VEGF, KDR inhibitor cocktail and aggregated Aβ.** Pericytes were seeded for treatment experiments onto 24-well plates with density of 50,000 cells/well. Pericytic *SMAD3* expression is validated through RNAscope assay previously described (Supplementary Fig. 19). Next day, cells were treated with Recombinant VEGF (Three applied concentrations: 50 ng/mL, 100 ng/mL, and 200 ng/mL; R&D Biosystems, 293-VE-010/CF), KDR inhibitor cocktail (Semaxanib SU5416 (10 uM, SelleckChem S2845), Tivozanib AV- 951 (10 uM, SelleckChem S1207), and ZM 306416 (10 uM, SelleckChem, S2897) and aggregated Aβ (250 nM, AnaSpec, AS-20276). Treatments were applied for a total of three durations: 6, 12, and 24 h. Detailed pericyte differentiation and treatment strategy is depicted in Supplementary Fig. 20. *SMAD3* (Thermo Fisher, Hs00969210_m1), *PDGFRB* (Thermo Fisher, Hs01019589_m1), *LEF1* (Thermo Fisher, Hs01547250_m1), *GAPDH* (Thermo Fisher, Hs99999905_m1) expression were measured

via RT-qPCR as previously described. Effect of treatment was compared to medium change control. To perform statistical testing between non-treated and treated groups at a given treatment duration, we applied linear mixed effects model implemented in R package lmerTest. In the model below, Treatment is the variable of primary interest, Treatment and Diagnosis are the fixed effects whereas Batch and Subject are the random effects.

lmer(dCT of SMAD3 ∽ Treatment + Diagnosis + (1 | Batch) + (1 | Subject))

And for diagnosis-stratified analysis, the following model was applied:

lmer(dCT of SMAD3 ∽ Treatment + (1 | Batch) + (1 | Subject))

For VEGF treatment, to perform statistical testing between non-treated and treated groups at a given VEGF concentration (50 ng/mL, 100 ng/mL or 200 ng/mL), the following model was applied:

lmer(dCT of SMAD3 ∽ Treatment + Diagnosis + (1 | Batch) + (1 | Subject))

For visualization, we first calculated the ΔCT values of *SMAD3* by subtracting house-keeping gene *GAPDH* of the same well, i.e., ΔCT of *SMAD3* = *SMAD3* CT - *GAPDH* CT. Next, we obtained the median ΔCT of non-treated samples at each duration (6 h, 12 h or 24 h) separately as baseline points. For non-treated samples, the ΔCT values were normalized by dividing the baseline point of the same duration and then visualized. For treated samples, the 2 to the negative ΔΔCT values, i.e. $2^{(-(\Delta CT \text{ of treated samples} - \Delta CT \text{ of corresponding non-treated samples}))}$, were visualized.

### Zebrafish in vivo data generation and analysis

**Animal maintenance and experimentation.** Animals are maintained according to the Institutional Animal Care and Use Committee (IACUC) standards of the Institute of Comparative Medicine at the Columbia University Irving Medical Center and to the accepted guidelines[139–142]. The animal care and use program at Columbia University is accredited by the AAALAC International and maintains an Animal Welfare Assurance with the Public Health Service (PHS), Assurance number D16-00003 (A3007-01). Animal experiments were approved by the IACUC at Columbia University (protocol number AC-AABN3554).

**Single cell sequencing.** Amyloid toxicity was induced as described[84,85] in the adult telencephalon of double reporter transgenic zebrafish line – Tg(*her4:DsRed*)[86] and Tg(*fli1a:eGFP*)[26]. At 3 days after cerebroventricular injection, the brains were dissected, and single cell suspensions were generated as previously described[143,144]. After fluorescence-activated cell sorting (FACS) of GFP+ and DsRed+ cells in a separate tube via FACSAria II sorter (Gating strategy is depicted in Supplementary Fig. 24), Chromium Single Cell 3' Gel Bead and Library Kit v3.1 (10X Genomics, 120237) was used to generate single cell cDNA libraries. Generated libraries were sequenced via Illumina NovaSeq 6000 as described[28,85,143–145]. Cell Ranger Single Cell Software Suite (10X Genomics, v6.1.2) was used to demultiplex raw base call files generated from the sequencer into FASTQ files. In total, 22,396 cells were sequenced and analyzed. On average, 94.7% of the total 1,78 billion gene reads mapped to the zebrafish genome release GRCz11 (release 105). For quality control selection, we removed any cells with less than 200 expressed genes, having more than 5-fold ratio between nCount_RNA/nFeature_RNA, with more than 20% mitochondrial RNA genes, and with genes expressed in less than 3 cells. After filtering out the low-quality cells, 4960 cells with 22,031 genes for GFP+ and 14,230 cells and 24,790 genes for DsRed+ cohorts remained. The Seurat objects were created, normalized, and the top 2000 variable genes were used for further analyses. We used DoubletFinder[146] to identify and remove doublets. After identifying the anchors (FindIntegrationAnchors), the datasets were integrated (IntegrateData). The integrated Seurat object included 19,190 cells and 26,095 genes. The data were scaled using all genes, and 30 PCAs (RunPCA) were identified. Cell clustering, marker gene

analyses, differential gene expression and preparation of feature plots were performed using Seurat V4 as described[100,144,147–149]. The clusters were identified using a resolution of 1. In total, 20 clusters for GFP+, 26 clusters for DsRed +, and 34 clusters for integrated objects were identified. The main cell types were identified by using *s100b* and *gfap* for Astroglia; *sv2a, nrgna, grin1a, grin1b* for Neuron; *pdgfrb* and *kcne4* for Pericyte; *cd74a* and *apoc1* for Microglia; *mbpa* and *mpz* for Oligodendrocyte; *aplnra* for OPC; *myh11a* and *tagln2* for vascular smooth muscle cells, *lyve1b* for Lymph endothelial cells and *kdrl* for vascular cells[10,15]. To find signature genes, we used FindMarkers function of Seurat with 0.25 logfc.threshold. The zebrafish gliovascular single cell dataset can be accessed at NCBI's Gene Expression Omnibus (GEO) with the accession number GSE225721.

**Treatment, immunohistochemistry, quantification, and statistical analyses.** For zebrafish studies, 6 months old Tg(*kdrl*:GFP)[27] reporter fish of both genders were used. In every experimental set, animals from the same fish clutch were randomly distributed for each experimental condition. The fish were treated with a mixture of Semaxanib (SU5416) (10 μM, SelleckChem S2845), Tivozanib (AV- 951) (10 μM; SelleckChem S1207), and ZM 306416 (10 μM; SelleckChem S2845) in fish water for 3 h per day for three consecutive days. Euthanasia and tissue preparation were performed as described[85]. 12-μm thick cryo-sections were prepared from these brain samples using a cryostat and collected onto glass slides which were then stored at −20 °C. Immunohistochemistry was performed as previously described[85] using the following antibodies: chicken anti-GFP (Thermofisher, PA1-9533, 1:1000), rabbit anti-phospho-ERK (Cell Signaling, 9101, 1:500), rabbit anti-phospho-SMAD3 (Abcam, EP823Y, ab52903, 1:500), mouse anti-ZO-1 (Thermofisher, ZO1-1A12, 33-9100, 1:500). Images were acquired using a Zeiss AxioImager Z1 and Zeiss LSM800 confocal microscope. The quantification of the colocalization of markers was performed using ImageJ software's colocalization module by generating two-channel composite, R(and) colocalization analyses and Fay translation into correlation values (Supplementary Fig. 27). The statistical evaluation was performed using GraphPad Prism (GraphStats, 6.02). Pairwise comparisons were performed with unpaired parametric t test with Welch's correction. The effect sizes for animal groups were calculated using G-Power, and the sample size was estimated with n-Query. At least 4 animals from both sexes were used per group.

**Statistics and reproducibility.** Our main findings were successfully replicated and validated using a systematic approach. From human brain snRNAseq study, we discovered perturbed vascular and astrocytic transcript pairs, of which pericytic SMAD3 (up in AD) and astrocytic VEGFA (down in AD) were prioritized. In Vitro: We validated VEGFA-SMAD3 interactions in human iPSC-derived pericytes. Treatment of human pericytes with VEGF (encoded by VEGFA) reduces SMAD3, and blocking VEGF signaling increases SMAD3. In Vivo: To determine impact of VEGFA-SMAD3 interactions on the blood-brain-barrier experimentally, we utilized a well-established zebrafish model. Injection of amyloid beta-42 in this model decreased vegfaa (zebrafish ortholog to human VEGFA) expression in astroglia. Blocking vegfaa signaling pharmacologically increased phosphorylated Smad3, the active form of this signaling molecule and importantly also impaired blood-brain-barrier integrity. Postmortem Measures: Age and sex matched AD and control donors were used in snRNAseq data generation ($n = 24$). SMAD3 and VEGFA expression were validated independently from nuclei isolated on the original 20/24 donors with sufficient tissue (Supplementary Data 27). Each qPCR experiment contained three technical replicates for each gene expression. Expression of VEGFA in astrocytes and SMAD3 in vascular cells were validated in nuclei isolated from the original 20/24 donors with sufficient tissue via RNAscope ($n = 18$, Supplementary Data 28, 29). Total of 16 images were analyzed and all number of cells were annotated for the

marker expression. Immunohistochemistry experiments were performed in the TCX from another cohort of AD and control donors from the Mayo Clinic Brain Bank ($n = 20$, Supplementary Data 30). Each staining experiment was performed independently from other donors. SnRNAseq findings were validated using external datasets (Supplementary Data 31) independently. Antemortem Measures: Two independent study groups, ADNI and MCSA were used. For all analyses, we included relevant covariates such as *APOE* genotype, sex and batch in the model where appropriate. Five Different SMAD3 probes were used to assess blood PaxGene expression. Correlation between SMAD3 probes as well as the average expression across all 5 probes is shown in Supplementary Fig. 17. IPSC Measures: 4 independent patient derived iPSCs were utilized in functional experiments. Each differentiating pericyte batch contained 6 technical replicates. There have been at least four repetitions of each experimental finding. Experimental design is displayed in Supplementary Fig. 18. Zebrafish Measures: Power analyses in zebrafish was performed using G*Power. At least 4 animals were used from both sexes as biological replicates. Multiple tissue sections were used per animal.

We applied different randomization strategies in each section. For human snRNAseq, randomization was performed such that the AD patients and control donors were matched for age at death and sex. Each batch or flowcell contained a balanced proportion of males/females and diagnosis groups. For Human antemortem studies, variant dosages were tested for association with infarcts while adjusting for age (at time of neuroimaging), sex, batch and the first three principal components (PCs) accounting for population substructure. For in vitro IPSC studies, we applied linear mixed effects model implemented in R package lmerTest. In the model, Treatment is the variable of primary interest, Treatment and Diagnosis are the fixed effects whereas Batch and Subject are the random effects. In the last section, zebrafish were randomly assigned to experimental and control groups to minimize selection bias and ensure that each fish has an equal chance of being placed in any group, making the groups comparable at the start of the experiment. In studies involving transgenic reporter lines, randomization was also applied to the selection of lines for experimentation. In experiments where there were known or potential confounding variables (e.g., age, sex, or batch effects in transgenic lines), a randomized block design was used. Zebrafish were grouped into blocks based on known variables, and then within each block, individuals were randomly assigned to experimental groups. Alongside randomization, evaluators were blinded to the group assignments, especially when evaluating outcomes. The person analyzing the results was different from the person conducting the experiments.

Owing to the nature of each section, we applied different blinding strategies. For the snRNAseq experiments, samples were randomized and were assigned a unique identifier. The technicians were blinded in the workflow except for nuclei sorting in snRNAseq data generation, where the diagnosis of the specimen was known. The analysts were not blinded for the analysis of the data. Bioinformatics personnel is blinded to overall experimental goals in antemortem studies. During in vitro experiments, qPCR analyses were performed in a blinded fashion. IPSC maintenance, differentiation, and treatments were performed by an independent technician. RNA isolation and qPCR experimental data generation were performed by another technician. Data were analyzed and visualized by independent personnel. In the last section, image analyses in zebrafish was performed in a blinded fashion. Tissue staining and labeling was performed by one experimenter and IDs were revealed after quantification by another experimenter.

## Reporting summary

Further information on research design is available in the Nature Portfolio Reporting Summary linked to this article.

## Data availability

All generated human snRNAseq data in this manuscript is available via the AD Knowledge Portal (https://adknowledgeportal.synapse.org). The AD Knowledge Portal is a platform for accessing data, analyses and tools generated by the Accelerating Medicines Partnership (AMP AD) Target Discovery Program and other National Institute on Aging (NIA)-supported programs to enable open-science practices and accelerate translational learning. Data is available for general research use according to the following requirements for data access and data attribution (https://adknowledgeportal.synapse.org/DataAccess/Instructions). An overview of all the data generated and used in this study can be found on the manuscript landing page (https://doi.org/10.7303/syn52669545). The single-nucleus RNAseq data generated in this study is deposited in the AD Knowledge Portal under The Mayo Clinic Single Nucleus RNAseq Study (MC_snRNA) (https://doi.org/10.7303/syn31511672). The zebrafish gliovascular single cell transcriptomics dataset can be accessed at NCBI's Gene Expression Omnibus (GEO) with the accession number GSE225721. Data availability information of external snRNAseq datasets are in Supplementary Data 31. The full complement of clinical and demographic data for the ADNI cohorts are hosted on the LONI data sharing platform and can be requested at http://adni.loni.usc.edu/data-samples/access-data/. The Mayo blood RNAseq data used in this study have already been deposited in the AD Knowledge Portal under The Mayo Clinic Study of Aging (MCSA) (accession ID: syn22024536). The additional data generated in this study are provided in the Supplementary Information/Source Data file. Source data are provided with this paper.

## Code availability

Codes used for the analysis and visualization of snRNASeq data are available in the AD Knowledge Portal (https://doi.org/10.7303/syn58577320).

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

## Acknowledgements

We would like to thank the patients and their families for their participation, without whom these studies would not have been possible. This work was supported by the National Institutes of Health, National Institute on Aging [RF AG051504, U01 AG046139, R01 AG061796, U19 AG074879 to NET; P30 AG062677 to RCP, CJ, KK, MEM; R01 AG054449, R01 AG075802, U19 AG069701 to MEM; R01 LM012535 and U01AG072177 to KN; U19 AG024904, P30 AG072976, U01 AG068057, to AJS; NIA R01AG067501, RF1AG066107, R01AG072474 to RM, BNV, CK], Columbia University Schaefer Research Scholar Award, Thompson Family Foundation Program for Accelerated Medicines Exploration in Alzheimer's Disease and Related Disorders of The Nervous System (TAME-AD), and Taub Institute Grants for Emerging Research (TIGER) to C.K. NET is also supported by the Alzheimer's Association Zenith Fellows Award (ZEN-22-969810). We thank the Mayo Clinic Genome Analysis Core (GAC), Co-Directors, Julie M. Cunningham, PhD and Eric Wieben, PhD, and supervisor Julie Lau, for their collaboration in collection of omics data. We would like to thank the Mayo Clinic Center for Regenerative Biotherapeutics for providing iPSC lines and relevant patient data. We would like to thank Taub Institute for Research on Alzheimer's Disease and the Aging Brain Imaging Platform at Columbia University, Molecular Pathology (MPSR) and Flow Cytometry Core Facility (CCTI, supported in part by the Office of the Director, National Institutes of Health under awards S10OD020056) platforms of the Columbia University Herbert Irving Comprehensive Cancer Center for procedural support, and New York Brain Bank for post-mortem human brain sections. The single cell sequencing for zebrafish was performed by the Single Cell Analysis Core and Columbia Genome Center at the Sulzberger Genome Center, which was funded in part through the NIH/NCI Cancer Center Support Grant P30CA013696 and used the Genomics and High Throughput Screening Shared Resource. Part of the data generation for this publication at Columbia University was also supported in part by the National Center for Advancing Translational Sciences, National Institutes of Health, through Grant Number UL1TR001873. Data collection and sharing for this project was funded by the Alzheimer's Disease Neuroimaging Initiative (ADNI) (National Institutes of Health Grant U01 AG024904) and DOD ADNI (Department of Defense award number W81XWH-12-2-0012). ADNI is funded by the National Institute on Aging, the National Institute of Biomedical Imaging and Bioengineering, and through generous contributions from the following: AbbVie, Alzheimer's Association; Alzheimer's Drug Discovery Foundation; Araclon Biotech; BioClinica, Inc.; Biogen; Bristol-Myers Squibb Company; CereSpir, Inc.; Cogstate; Eisai Inc.; Elan Pharmaceuticals, Inc.; Eli Lilly and Company; EuroImmun; F. Hoffmann-La Roche Ltd and its affiliated company Genentech, Inc.; Fujirebio; GE Healthcare; IXICO Ltd.; Janssen Alzheimer Immunotherapy Research & development, LLC.; Johnson & Johnson Pharmaceutical Research & Development LLC.; Lumosity; Lundbeck; Merck & Co., Inc.; Meso Scale Diagnostics, LLC.; NeuroRx Research; Neurotrack Technologies; Novartis Pharmaceuticals Corporation; Pfizer Inc.; Piramal Imaging; Servier; Takeda Pharmaceutical Company; and Transition Therapeutics. The Canadian Institutes of Health Research is providing funds to support ADNI clinical sites in Canada. Private sector contributions are facilitated by the Foundation for the National Institutes of Health (www.fnih.org). The grantee organization is the Northern California Institute for Research and Education, and the study is coordinated by the Alzheimer's Therapeutic Research Institute at the University of Southern California. ADNI data are disseminated by the Laboratory for Neuro Imaging at the University of Southern. Data used in preparation of this article were obtained from the Alzheimer's Disease Neuroimaging Initiative (ADNI) database (adni.loni.usc.edu). As such, the investigators within the ADNI contributed to the design and implementation of ADNI and/or provided data but did not participate in analysis or writing of this report. A complete listing of ADNI investigators can be found at: http://adni.loni.usc.edu/wp-content/uploads/how_to_apply/ADNI_Acknowledgement_List.pdf.

## Author contributions

N.E.T. and M.A. designed the study; Ö.İ., X.W., J.S.R., Y.M., E.Y., P.B., C.K., and N.E.T. wrote the initial draft of the manuscript; X.W., J.S.R., Ö.İ., Y.M., E.Y., M.I.C., B.C.D., S.O., M.M.C., and Z.Q. performed data analysis; M.G.H., L.W., A.J.L., B.N.V. consulted on statistical methods; Ö.İ., X.W., J.S.R., Y.M., E.Y., P.B., M.I.C., J.B., K.D., T.C., K.N., C.K. generated tables and figures; N.K., D.W.D., M.E.M. provided fresh frozen brain samples with their pathology information; S.K. and D.W.D. provided performed and provided brain immunohistochemistry; R.M. and C.K. acquired human brain tissue samples from Columbia University for immunofluorescence; P.B. and C.K. performed immunofluorescence; Ö.İ., T.P., J.B., E.Y., P.B., F.Q.T.N., V.K., K.D., T.C., and T.N. performed experimental procedures from blood and brain samples; E.Y., P.B., N.N., M.I.C., C.K. designed and/or performed zebrafish experiments; E.Y., P.B., M.I.C., C.K. generated single cell transcriptomics dataset in zebrafish; L.J.L.T. and T.K. consulted on experimental data generation. Y.I. and T.K. provided iPSCs and expertize; J.G.R., R.C.P., C.J., and K.K. provided data from MCSA. K.N. and A.J.S. provided data and performed analyses for ADNI. All authors read the manuscript and provided input and consultation. Ö.İ., X.W., and N.E.T. finalized the manuscript. N.E.T. oversaw the study and provided direction, funding, and resources.

## Competing interests

The authors declare no competing interests.

## Additional information

Özkan İş [1,19], Xue Wang [2,19], Joseph S. Reddy [2], Yuhao Min [1], Elanur Yilmaz [3,4], Prabesh Bhattarai [3,4], Tulsi Patel[1], Jeremiah Bergman[1], Zachary Quicksall [2], Michael G. Heckman[2], Frederick Q. Tutor-New[1], Birsen Can Demirdogen [1,5], Launia White[2], Shunsuke Koga [1], Vincent Krause[1], Yasuteru Inoue[1], Takahisa Kanekiyo [1], Mehmet Ilyas Cosacak [6], Nastasia Nelson[3,4], Annie J. Lee[3,4,7], Badri Vardarajan [3,4,7], Richard Mayeux[3,4,7,8,9], Naomi Kouri [1], Kaancan Deniz [1], Troy Carnwath[1], Stephanie R. Oatman[1], Laura J. Lewis-Tuffin [10], Thuy Nguyen [1], for the Alzheimer's Disease Neuroimaging Initiative*, Minerva M. Carrasquillo [1], Jonathan Graff-Radford[11], Ronald C. Petersen [11,12], Clifford R. Jr Jack [13], Kejal Kantarci[12], Melissa E. Murray [1], Kwangsik Nho[14,15,16], Andrew J. Saykin [15,16,17], Dennis W. Dickson [1], Caghan Kizil [3,4,7], Mariet Allen [1] & Nilüfer Ertekin-Taner [1,18] ✉

¹Department of Neuroscience, Mayo Clinic, Jacksonville, FL, USA. ²Department of Quantitative Health Sciences, Mayo Clinic, Jacksonville, FL, USA. ³Department of Neurology, Columbia University Irving Medical Center, New York, NY, USA. ⁴Taub Institute for Research on Alzheimer's Disease and the Aging Brain, Columbia University Irving Medical Center, New York, NY, USA. ⁵Department of Biomedical Engineering, TOBB University of Economics and Technology, Ankara, Turkey. ⁶German Center for Neurodegenerative Diseases (DZNE) within Helmholtz Association, Dresden, Germany. ⁷The Gertrude H. Sergievsky Center, College of Physicians and Surgeons, Columbia University, New York, NY, USA. ⁸Department of Psychiatry, Columbia University Irving Medical Center, New York, NY, USA. ⁹Department of Epidemiology, Mailman School of Public Health, Columbia University, New York, NY, USA. ¹⁰Mayo Clinic Florida Cytometry and Cell Imaging Laboratory, Mayo Clinic, Jacksonville, FL, USA. ¹¹Department of Neurology, Mayo Clinic, Rochester, MN, USA. ¹²Mayo Clinic Alzheimer's Disease Research Center, Rochester, MN, USA. ¹³Mayo Clinic, Radiology, Rochester, MN, USA. ¹⁴Center for Computational Biology and Bioinformatics, Indiana University School of Medicine, Indianapolis, IN, USA. ¹⁵Department of Radiology and Imaging Sciences, Indiana University School of Medicine, Indianapolis, IN, USA. ¹⁶Indiana Alzheimer's Disease Research Center, Indiana University School of Medicine, Indianapolis, IN, USA. ¹⁷Department of Medical and Molecular Genetics, Indiana University School of Medicine, Indianapolis, IN, USA. ¹⁸Department of Neurology, Mayo Clinic, Jacksonville, FL, USA. ¹⁹These authors contributed equally: Özkan İş, Xue Wang. *A list of authors and their affiliations appears at the end of the paper. ✉e-mail: taner.nilufer@mayo.edu

## for the Alzheimer's Disease Neuroimaging Initiative

Ronald C. Petersen [11,12], Clifford R. Jr Jack [13], Kejal Kantarci[12], Kwangsik Nho[14,15,16] & Andrew J. Saykin [15,16,17]

A full list of members and their affiliations appears in the Supplementary Information.

