## [Peer Review File · Nature Communications]

Gliovascular transcriptional perturbations in Alzheimer's disease reveal molecular mechanisms of blood brain barrier dysfunctionEditorial Note: This manuscript has been previously reviewed at another journal that is not operating a transparent peer review scheme. This document only contains reviewer comments and rebuttal letters for versions considered at *Nature Communications*. Mentions of the other journal have been redacted.

REVIEWER COMMENTS

Reviewer #1 (Remarks to the Author):

The authors have provided a deluge of additional supplementary data in this revised version of the original manuscript. This takes a considerable amount of time to digest, and unfortunately it raises some additional concerns that are details below.

The original concern about experimental design/analysis, and low power remain (see below).

On a positive note, the authors now include validation of their identified DEGs, which is both a plus, and a requirement for publication. There is some ambiguity about the sample input for these validation steps (see comments below) that must be addressed/highlighted so that the reader can understand the importance and validity of these experiments.

If the editorial team would allow and deem appropriate, the authors should include a detailed description of caveats of this manuscript - including experimental design and analysis, low power, and missing details about patient cohorts (though perhaps these can be included). The standard required for sequencing manuscripts has increased considerably in recent years, and the investigation of a particular cell type using gross snRNAseq data that is underpowered for the cell of interest is simply not best practice. Combined with the lack of details in supplementary material, and that the authors have not addressed several of the major concerns raised in the earlier round of review, make it very difficult to endorse this manuscript.

Comments that require clarification/additional edits:

1. The power concern remains, and is most notable when considering summary data like that displayed in Supp Fig S10 - astrocytes are represented by >6000 nuclei (across all samples), while the pericyte/fibroblast/endothelial clusters (targets here) are represented by only dozens-to-hundreds of nuclei (across all samples). This makes the statistical testing of these putative receptor-ligand interactions unbalanced.

2. in response to reviewer concerns (R2, comment 2.1.2) these Quality control of nuclei statements do not detail factors that would be important for the original reviewer concern - that the output of snRNAseq was indeed reporting sequencing from cell nuclei as proposed in the authors narrative. Some ambient RNA cleanup, and ensuring doublet/mito/ribosomal RNA contamination is removed is integral. Regardless of the low number of nuclei input (suggested as a reason for not wanting to remove low quality nuclei by R2), these data must be properly cleaned and low quality nuclei removed. This is standard for the field, and has been omitted now from two versions of this manuscript.

3. Some major concerns from R2 about cell type identification have only been addressed by screenshot from the brainrnaseq.com website, rather than checking in their own system. Validation cannot be provided by using a different dataset - and considering the low power of nuclei sequenced, it was a real shame to see the authors not adequately address this concern using their own samples/system.

4. Response Fig 4 missing.

5. there are no purity measurements for iPSC-derived pericytes (see Supp Fig S17) - such measurements are imperative for interpretation of such assays

6. Supp Fig S8 - what are the gray circles and gene numbers (e.g. 4732) - are these total DEGs from the current study across all cells? This is another example of a poor legend making the data uninterpretable. Supp Figure S6 raises additional concerns (originally raised by R1 and R2) about the power of these studies. Given the low numbers of cells here (which contrary to arguments from the authors that the total glio-vascular unit number being high - are still very low for each individual cell type, making investigations of interactions quite difficult), the variability of different AD sub-types now drops the n values for each sub-type considerably. While it is too late to correct for this experimentally, it is a real shame that the authors did not choose a clearly defined patient cohort before embarking on this study. A limitations statement should be included now, as some of the differentially expressed genes presented may be present in only one patient group, or validated incorrectly in postmortem tissue from another patient cohort. (see gross differences in a small subset of genes in Supp Fig S12, panel B)

7. missing figure legends - e.g. Supp Fig S23, but also lacking considerable information on many (e.g. Supp Figs S13, S18, S20, S24-S26)

8. Supp Fig S5 would be better visualized by coloring variables in different colors and plotting integrated - e.g. ADvNormal, or male v female.

Also the authors should appropriately annotate these figure panels - leaving sex as '0' and '1' is not a difficult output to correct. Similarly, what does APOE4 positive/negative mean - are these homozygotes or heterozygotes?

>for an example of an easier to view output taking advantage of this suggestion - check Supp FigS23, panel B, which correctly plots samples on the same UMAP colors by treatment

9. Supp Fig S14 - how do these data show that AGT expression is restricted to human temporal cortex astrocytes? AGT is expressed by all three astrocyte clusters - suggesting it is expressed in all brain regions included in this dataset.

10. scale bars missing from micrographs in Supp Fig S3, S17

11. full blots should be included, with MW markers - e.g. Supp Fig S1

Reviewer #2 (Remarks to the Author):

The authors have addressed my comments and this is a solid body of work. Though some concerns

remain about the overall robustness of their lenient DEG analyses (perhaps necessary because of low vascular nuclei captured), the additional iPSC and zebrafish experiments strengthen their main claims of astrocyte-to-pericyte interactions in AD.

Minor comment:

-The correlation plot in Fig. 5C, right, is confusing: are the authors correlating the correlation coefficients? Is the intent to correlate the levels of pERK/GFP and ZO-1/GFP and provide one correlation coefficient R?

Reviewer #3 (Remarks to the Author):

Re-review: Identification of novel gliovascular interactions perturbed in Alzheimer's disease and conserved in cross-species models [REDACTED]

I appreciate the efforts made by the authors in addressing the previous comments and conducting additional validation experiments. The manuscript has certainly improved. However, I still have a two remaining concerns:

1) It's great to see the authors referencing the preprint by Zhang et al. to support their findings regarding SMAD3 gene expression in pericytes of AD and VEGFA downregulation in astrocytes. However, I'd like to suggest considering several other peer-reviewed brain/brain vascular atlases that are published in well-respected journals. Some of these include PMID: 35165441, PMID: 31042697, PMID: 31768052, PMID: 37774677, PMID: 31932797. It might be beneficial to explore and compare the expression of those findings across these atlases instead of focusing primarily on one preprint that confirms the own findings. This could provide a more comprehensive view of the subject.

2) I appreciate the validation experiments carried out by the authors. However, I do still have some concerns about the low nuclei numbers. It might be beneficial for the authors to use the available atlases that I mentioned above to further validate their claims and provide additional support for their findings. This could help reinforce the study's conclusions. It would also be beneficial to state more clearly the limitations of the study.

Identification of novel gliovascular interactions perturbed in Alzheimer's disease and conserved in cross-species models.

Responses to Reviewers' Comments

NCOMMS-23-47665-T

Our sincere gratitude goes out to the Editors and Reviewers for their careful consideration of our manuscript. We greatly appreciate that the Reviewers provided us with positive, constructive feedback and recommendations. Detailed below are the modifications we made to address the comments of our Reviewers, which we believe further enhanced the findings and impact of our study.

The Reviewers' individual comments are highlighted in *italic font* below. For each of our Main and Supplementary files, we provide both a version with all changes tracked and a version with accepted changes. Text changes to the manuscript we depict in this document is in Arial font.

To address all comments, we have re-conducted snRNAseq clean-up, QC and repeated all relevant analyses. This resulted in updating of all results in **Main Figures 1, 2, 3A, B, F; Supplementary Tables S1-S26; Supplementary Figures S4-S11, S13 and S14**. We have also downloaded, integrated and analyzed brain snRNAseq data from external datasets comprising 8 additional brain regions from 6 external studies yielding a total of 4,730 pericytes and 150,664 astrocytes. We replicated our main findings in these integrated datasets and include these new results in **new Main Figure 3F, new Supplementary Tables S31-S34, new Supplementary Figures S15 and S16**. Importantly, after these suggested analyses, none of our results change, and our conclusions are further strengthened and refined.

With these edits and others delineated in this document, we believe that we have addressed every comment from each of the Reviewers. We look forward to further feedback and a favorable decision from our Editor and Reviewers.

Responses to comments from two or more reviewers:

First, we thank Reviewer 1 for their suggestion to remove doublets. While this was not part of the analytic standards at the time of our project initiation, we agree that it is important to conduct this per current standards. Consequently, we have performed doublet removal using Scrublet¹ and re-performed all the downstream analyses. All of the results in the current submission, including Main and Supplementary Text, Figures and Tables are post-doublet removal. More details about removing doublets are described in the Responses to Reviewer 1 section, below. In summary, doublet removal has not changed any of our findings nor conclusions of the manuscript.

We thank and agree with Reviewer 1 that utilizing our dataset alone may raise concerns about statistical power. We also thank and agree with Reviewer 3's suggestion to analyze external datasets, including PMID: 35165441², PMID: 31042697³, PMID: 31768052⁴, PMID: 37774677⁵, PMID: 31932797⁶. Per these excellent suggestions, we have obtained all of these aforementioned dataset, as well as another study by Sun et al.⁷ in our revised manuscript. This enabled us to analyze 150,664 astrocyte and 4,730 pericyte nuclei (**new Supplementary Tables S31-S34, Response Tables R1-R3**) from these datasets and our study. This integrated dataset is >5x (for pericytes) to >24x (for astrocytes) larger than ours alone (926 pericyte; 6,165 astrocyte nuclei). We identified astrocyte and pericyte sub-clusters using these integrated data and characterized the subclusters that demonstrate *SMAD3* upregulation in AD pericytes and *VEGFA* downregulation in AD astrocytes. Importantly, we replicate our findings in this integrated dataset. In summary, these findings should alleviate the concerns about power and provide a more complete picture about our findings in other brain regions.

We provide detailed methods and findings below. In summary, the datasets are summarized in **Response Table R1**. This table is also provided as a **new Supplementary Table S31**, in the resubmission. In order to avoid any bias in our analyses, we used the post-QC, post cell type assignment data associated with the original publications. We note that Zhou et al.⁶ lacked post-QC, post-cell assignment data and meta data. Therefore, this dataset is not included in our analyses. Nevertheless, the brain region analyzed by Zhou et al., i.e. prefrontal cortex (PFC) is very well-represented (in fact over-represented) in other datasets both in terms of nuclei and participants. Another point is that the Mathys et al. 2023/multi-region⁵ and Sun et al.⁷ shared the dataset as indicated at compbio.mit.edu/ad_aging_brain/, albeit Sun et al.⁷ mainly focused on analyses of vascular cell types (compbio.mit.edu/scADbbb/). Therefore, we only included Sun et al.⁷ in pericyte related analyses, and Mathys et al. 2023⁵/multi-region in astrocyte related analyses.

In the revised manuscript, we include the methods describing these external datasets and their integration in **Methods Section 19. External snRNAseq datasets** and **Section 20. Integrating external snRNAseq datasets**.

Response Table R1: Summary of datasets

Datasets	Year of publication	Brain Region	Post-QC, clustered cell provided?	Ast/Per provided?	QC and Cluster by us?	PMID
Grubman et al	2019	EC	Yes	Yes/No	No	31768052
Mathys 2019	2019	DLPFC	Yes	Yes/Yes	No	31042697
Yang et al.	2022	HC; SFX	Yes	Yes/Yes	No	35165441
Mathys 2023	2023	PFC	Yes	Yes/Yes	No	37774677
Mathys multi-reg	2023	PFC;AG;TH;EC;MTX;HC	Yes	Yes/Yes	No	37774677
Sun et al	2023	PFC;AG;TH;EC;MTX;HC	Yes	No/Yes	No	37907809
Zhang et al	in medRxiv	MTX	No	Yes/Yes	Yes	GSE188545
Is et al	Under review	TCX		Yes/Yes	Yes	

Response Table R1 continued: Summary of datasets

Datasets	Brain region	# of participants/astrocyte nuclei	# of participants/pericyte nuclei
Grubman et al	EC	12/2129	0/0
Mathys 2019	DLPFC	48/3392	37/167
Yang et al	HC; SFX	17;8/16754;5941	17;8/15784;11412
Mathys 2023	PFC	427/149558	390/5308
Mathys multi-reg	PFC;AG;TH;EC;MTX;HC	41;41;38;40;41;41/ 14841;14899;26240;21222;13458;18808	41;41;37;40;41;38/545;553;519;462;497;434
Sun 2023	PFC;AG;TH;EC;MTX;HC	0/0	375;45;42;43;47;68/3999;441;503;315;346;400
Zhang et al	MTX	12/6773	12/226
Is et al	TCX	24/6265	24/926

Ast: astrocytes

Per: pericytes

TCX: temporal cortex

EC: entorhinal cortex

DLPFC: dorsolateral prefrontal cortex

PFC: prefrontal cortex

SFX: superior frontal cortex

Hippocampus: HC

AG: angular gyrus

TH: thalamus

MTX: midtemporal cortex

Up-regulation of *SMAD3* in the largest pericytic cluster of AD donor brains in integrated datasets:

To increase the number of pericyte nuclei and the number of donors, we integrated pericyte nuclei of external datasets with ours, performed clustering, and differential expression (DE) analysis of *SMAD3* in each subcluster.

We took pericyte nuclei in Is et al (our manuscript), Sun et al.⁷, Zhang et al. (in medRxiv, GSE188545), Yang et al.² and Mathys2019³, performed SCT transformation v2⁸, integration using Harmony⁹ and cell clustering using Seurat function FindNeighbors and FindClusters. We noticed that the number of nuclei in each participant in Yang et al.² is much greater than that in other studies, as Yang et al. enriched vascular nuclei using VINE-seq method². Additionally, in Sun et al.⁷, PFC was over-represented compared to other brain regions with ~4000 pericytes from 375 participants⁷ (**Response Table R1, Supplementary Table S31**). Therefore, we down-sampled nuclei as follows: For Sun et al.⁷, we randomly selected 25 AD and 25 control participants and included all nuclei for them. For Yang et al.², 400 pericytic nuclei were randomly selected from each brain region, i.e. SFX and HC, with even numbers in each participant. Given the increased number of participants and multiple studies in this integrated analysis, we applied a negative binomial generalized linear mixed effects model for

DE analysis¹⁰. R package glmmTMB was used for this analysis, where the participants and studies were coded as random effects, diagnosis, age at death and sex were coded as fixed effects.

We observed two pericyte subclusters from the integrated dataset of 4,730 pericyte nuclei described above. Per.0 cluster demonstrates upregulation of genes involved in solute transport, while Per.1 shows extracellular matrix organization, that resembles T-pericytes and M-pericytes from Yang et al.², respectively. We performed DE analysis of *SMAD3* for nuclei from each study and all studies in each subcluster, respectively. As shown in **Response Figure R1 (and new Main Figure 3F)**, in the larger subcluster Per.0, *SMAD3* was significantly up-regulated in AD participants in Is et al, Zhang et al and all cohorts combined, with a trend of up-regulation in HC of Yang et al.², and EC and MTX in Sun et al⁷. There is no significant down-regulation in any of the cohorts. As shown in **Response Table R2**, Per.0 subcluster contains ~2.5X number of nuclei compared to per.1. Interestingly, pericytes from Is et al (TCX region) are up-regulated in both subclusters. Signature markers of both pericyte clusters and GO enrichment profile of Per.0 have been added to Supplementary data as **new Supplementary Figure S15** and **new Supplementary Table S34**.

Response Figure R1: **A)** UMAP plot of pericytic nuclei from five studies. Colored dots: nuclei of subcluster Per.0. Gray dots: other nuclei. **B)** Expression of known pericyte marker genes in pericyte clusters. **C)** Forest plot of DE results of nuclei in subcluster Per.0 from each cohort by study and brain region, and all Per.0 nuclei, combined. Cohort column indicates the study and the brain region from which the results are obtained. The middle Per.0 column provides the number of nuclei in Per.0 in each cohort. P value is from DE analysis between AD and control nuclei using mixed effects model. The square indicates coefficient which is the natural log(fold change) of *SMAD3* expression in AD vs. control nuclei. The left bar in the forest plot indicates the 2.5% confidence interval. The right bar indicates the 97.5% confidence interval. For Sun et al, we randomly selected 25 AD and 25 control participants and included all nuclei for them. For Yang et al, 400 pericytic nuclei were randomly selected from each brain region, i.e. SFX and HC, with even numbers for each participant. **D)** Top Enriched GO terms of signature genes in pericyte cluster0 compared to cluster1 show downregulated genes in the

pathways associated with immune functions and upregulation of plasma membrane proteins. *: enrichment FDR < 0.05.
 ***: enrichment FDR < 0.001.

Response Table R2: differential expression of *SMAD3* in integrated dataset, all cohorts for pericyte subclusters.

cluster	cluster	Size	pValue	coef	97.5% confidence interval	2.5% confidence interval
Per.0	Per.0	3395	3.5E-02	0.16	0.30	0.01
Per.1	Per.1	1335	5.3E-01	-0.09	0.19	-0.38
All	All	4730	2.4E-01	0.10	0.27	-0.07

The coef is the natural log(fold change) in AD vs. control nuclei.

In summary, by integrating external datasets from multiple brain regions, we observed two pericyte subclusters, and *SMAD3* is significantly up-regulated in AD in the larger one characterized by solute transport genes. The full results of these analyses are presented in **new Supplementary Table S32, new Main Figure 3F, Results section Validation and replication of the prioritized pericyte target-astrocyte ligand pair SMAD3-VEGFA in human AD and control brains (3rd paragraph from last)**, added to the **Discussion (penultimate paragraph)** and **Abstract**.

Down regulation of *VEGFA* in astrocytic clusters of AD donor brains in integrated datasets:

To increase the number of astrocyte nuclei and the number of donors, we integrated astrocyte nuclei of external datasets with ours, performed clustering, and DE analysis of *VEGFA* in each cluster. We note that the Mathys 2023⁵ multi-region study already included astrocyte nuclei from PFC region from 41 participants. Therefore, we did not include Mathys 2023 PFC dataset in this integrated analysis, which encompassed 149,558 astrocytes from 427 participants that would outweigh the PFC region.

As described previously, we performed SCT transformation v2⁸, integration using Harmony⁹ and cell clustering using Seurat function FindNeighbors and FindClusters. Mixed effect models implemented in R package glmmTMB were used to perform differential expression analysis. In the integrated dataset, 150,664 astrocytic nuclei formed 14 clusters. Notably, the astrocytic subcluster 0 (Ast.0) contained the majority of astrocytic nuclei from the integrated datasets as well as that of astrocytic cluster 8 of our own data where *VEGFA* is downregulated in AD (**Response Table R3**). In **Response Figure R2** below (also in **new Main Figure 3F**), we show the DE results of nuclei in Ast.0 from each study and all studies combined. **Response Figure R2** (and **new Supplementary Figure S16, new Supplementary Table S34**) also demonstrates signature markers of all astrocyte clusters and GO enrichment profile of Ast.0.

Ast.0 is enriched for genes involved in synaptic assembly and organization compared to other clusters. These cells display a gene expression profile similar to those in the Mathys et al.³ GRM⁺ astrocyte subcluster. In Ast.0, *VEGFA* was significantly down regulated in AD donors in Yang et al.² HC region, Zhang et al and all cohorts combined, and a trend of down regulation in Is et al (our study), Grubman et al.⁴, Yang et al.² SFX region, Mathys 2023⁵ EC region. There is no significant up regulation of *VEGFA* in any of the cohorts.

For all cohorts, **Response Table R3** listed the DE results from Ast.0 to Ast.12. Please note that Ast.13 only had 2 nuclei. *VEGFA* is significantly downregulated in AD donors in the two most abundant astrocyte subclusters Ast.0 and Ast.1 as well as all astrocyte nuclei combined. In summary, by integrating external datasets from multiple brain regions, we observed 14 astrocyte subclusters, and *VEGFA* is significantly down-regulated in AD in the largest two clusters, where the largest cluster Ast.0 is characterized by synapse assembly genes. The full results of these analyses are presented in **new Supplementary Table S33, new Main Figure 3F, Results section Validation and replication of the prioritized pericyte target-astrocyte ligand pair SMAD3-VEGFA in human AD and control brains** (penultimate paragraph), added to the **Discussion (penultimate paragraph)** and **Abstract**.

Response Figure R2: A) UMAP plot of astrocytic nuclei from six studies. Colored dots: nuclei of subcluster Ast.0. Gray dots: other nuclei. **B)** Expression of known astrocyte marker genes in astrocyte clusters. **C)** Forest plot of DE results of nuclei in subcluster Ast.0 from each cohort by study and brain region, and all Ast.0 nuclei combined. Cohort column indicates the study and the brain region from which the results are obtained. The middle Ast.0 column provides the number of nuclei in Ast.0 in each cohort. P value is from DE analysis between AD and control nuclei using mixed effects model. The square indicates coefficient which is the natural log(fold change) of *VEGFA* expression in AD vs. control nuclei. The left bar in the forest plot indicates the 2.5% confidence interval. The right bar indicates the 97.5% confidence interval. **D)** Top Enriched GO terms of signature genes in Ast.0 compared to all other astrocyte clusters are displayed in the plot. Upregulated genes are enriched in cell signalling and tube morphogenesis and downregulated genes are enriched in the extracellular matrix related pathways. *: enrichment FDR < 0.05. **: enrichment FDR < 0.01. ***: enrichment FDR < 0.001 ****: enrichment FDR < 1E-4.

Response Table R3: Differential expression of *VEGFA* in AD vs control donor brains from integrated datasets, all cohorts combined.

cluster	size	p value	coef	# nuclei in our ast cl.8	# nuclei in our ast cl.11	# nuclei in our ast cl.31	model
Ast.0	81431	0.013	-0.289	2700	667	74	Negative binomial generalized linear mixed effects by glmmTMB
Ast.1	42880	0.004	-0.451	306	1283	18	
Ast.2	4737	0.987	-0.003	129	110	4	
Ast.3	3406	0.402	-0.161	16	24	12	
Ast.4	2917	0.917	0.018	80	16	2	
Ast.5	2637	0.592	0.105	50	26	2	
Ast.6	2329	0.182	0.349	0	18	11	
Ast.7	2209	0.131	-0.297	20	15	236	
Ast.8	2166	0.479	-0.149	21	78	13	
Ast.9	2075	0.744	-0.078	14	74	5	
Ast.10	1595	0.375	-0.198	3	64	3	
Ast.11	1347	0.41	-0.231	4	35	2	
Ast.12	933	0.866	-0.043	0	29	1	
Ast.13	2	NA	NA	0	0	0	
All Astrocyte	150664	0.007	-0.334521	3443	2439	383	

The coef is the natural log(fold change) in AD vs. control nuclei.

With the external datasets which increased pericyte nuclei analyzed by >5x, astrocyte nuclei analyzed by >24x, we defined the nuclei subclusters for these cell types in the integrated dataset, confirmed our original findings of pericyte *SMAD3* upregulation and astrocyte *VEGFA* downregulation in AD, characterized the largest nuclei subclusters which demonstrated these changes according to their marker genes and GO term enrichment. These findings further strengthen our manuscript, support our conclusions, refine these discoveries and contextualize them in the landscape of all snRNAseq brain data from AD and control donors available to date, to our knowledge.

We next provide point-by-point responses to each of the Reviewers' comments.

Reviewer #1

Reviewer 1: *The authors have provided a deluge of additional supplementary data in this revised version of the original manuscript. This takes a considerable amount of time to digest, and unfortunately it raises some additional concerns that are details below.*

Response: We appreciate this Reviewer's recognition of the considerable amount of experiments, analyses and overall work that we put into the prior revision to address all of the comments from the prior review. We hope and expect that our revision detailed herein will likewise address each of the comments from the current review.

Reviewer 1: *The original concern about experimental design/analysis, and low power remain (see below).*

Response: We agree with Reviewer 1 that our study alone may raise the concern of low power. We addressed this concern as detailed above, by downloading and integrating external datasets which include 150,664 astrocyte and 4,730 pericyte nuclei from 6 or more brain regions. Our revised manuscript has updated **Methods, Results and Discussion** including this integrated analyses which support our original results as outlined above. We now include a **new Main Figure 3F, new Supplementary Tables s31-s34, new Supplementary Figures s15-s16** in this revised submission. To facilitate the review of these extensive edits, we also provide **Response Tables R1-R3, Response Figures R1-R2** as well as detailed text above. We thank the first and third Reviewers for their comments which led to these edits that further strengthened our manuscript.

Reviewer 1: *On a positive note, the authors now include validation of their identified DEGs, which is both a plus, and a requirement for publication. There is some ambiguity about the sample input for these validation steps (see comments below) that must be addressed/highlighted so that the reader can understand the importance and validity of these experiments.*

Response: We thank the reviewer's positive note on our experimental validation. Please see the following responses to each specific comment.

Reviewer 1: *If the editorial team would allow and deem appropriate, the authors should include a detailed description of caveats of this manuscript - including experimental design and analysis, low power, and missing details about patient cohorts (though perhaps these can be included). The standard required for sequencing manuscripts has increased considerably in recent years, and the investigation of a particular cell type using gross snRNAseq data that is underpowered for the cell of interest is simply not best practice. Combined with the lack of details in supplementary material, and that the authors have not addressed several of the major concerns raised in the earlier round of review, make it very difficult to endorse this manuscript.*

Response: We thank the reviewer's concrete suggestions. We would like to note that most of the demographics of the patients and their core neuropathologies were already listed in **Supplementary Table S1**, including age of death, sex, *APOE* genotype, and AD related neuropathology scores of Braak stage, Thal phase, and the presence of TDP43 aggregates. In addition, based on reviewer's suggestions, we added race, dementia duration, brain weight at death, presence of α -synuclein/Lewy bodies, and vascular disease (VaD) to **Supplementary Table S1**. We modified text in **Methods, Section Tier 1 Methods: Human Postmortem brain data generation and analysis, Sub-section 2. Histology and immunohistochemistry, paragraph 1** by adding the following text:

"To assess vascular disease, a summary of pathological vascular lesion scores based on the presence and number of macroscopic vascular lesions (large infarct, lacunar infarct, and leukoencephalopathy) that correlate with neuroimaging during life were used¹¹⁴. We assessed Lewy body pathology in the neocortices, cingulate gyrus, transentorhinal cortex, amygdala, basal forebrain, midbrain, pons, and medulla using α -syn immunohistochemistry (NACP, 1:3000 rabbit polyclonal, Mayo Clinic antibody)¹¹⁶. Lewy pathology was staged as follows: brainstem, transitional or diffuse LBD according as previously established¹¹⁷."

We also thank the reviewer's suggestion about including a paragraph of caveats. We note that we had already included a paragraph of study limitations in the prior submission. In this, submission, we expanded this section and also added text to reflect the findings from the external studies now integrated into our manuscript. This modified text is in **Discussion, penultimate paragraph** where the added text is highlighted **in yellow**.

"Despite these strengths, our study also has some weaknesses **and limitations**. In this study we focused on predicted interactions of brain vascular target molecules with astrocytic ligands, given their known crosstalk at the BBB^{1,70}. However, it will be important to also interrogate interactions with neurons, oligodendrocytes, and OPCs. Although we focused on one interacting pair (*VEGFA*-*SMAD3*), other predicted astrocytic ligands-vascular targets will also be worth following up in future experimental studies. Furthermore, our study focused on late-stage AD cases and a single brain region that has a relatively high burden of AD neuropathology. **Our discovery cohort of 24 AD and control brain donors where we conducted snRNAseq of TCX also has limited number of participants, and hence limited statistical power. To address the limitation in power and determine the applicability of our findings in *VEGFA* and *SMAD3* in other brain regions, we analyzed external datasets from multiple different brain regions^{15,16,21-24} resulting in an integrated dataset of 150,664 astrocyte and 4,730 pericyte nuclei from 6 or more brain regions. In these integrated datasets, we confirmed our findings of up-regulation in AD of *SMAD3* and down-regulation of *VEGFA* in the largest pericyte and astrocyte clusters, respectively. Importantly, we were able to characterize the pericyte and astrocyte**

subclusters with these expression changes and demonstrate their applicability in different brain regions, as well as studies of both selected vascular and unselected nuclei from AD and control brains.”

Reviewer 1: *The power concern remains, and is most notable when considering summary data like that displayed in Supp Fig S10 - astrocytes are represented by >6000 nuclei (across all samples), while the pericyte/fibroblast/endothelial clusters (targets here) are represented by only dozens-to-hundreds of nuclei (across all samples). This makes the statistical testing of these putative receptor-ligand interactions unbalanced.*

Response: We thank the reviewer’s question about unbalanced number of nuclei from pericyte and astrocyte. We have already addressed the issue of power in our extensive modifications described above. In addition, we would like to point out that the NicheNet¹¹ tool we used for predicting ligand-target interaction is not based on our dataset, but rather has a built-in prior model with evidence of ligand-target interactions from multiple existing databases. In this study, we provided to NicheNet differentially expressed (DE) genes in astrocyte as candidate ligands and DE genes in pericyte as candidate targets. While the candidate ligands and targets we provide are based on DE results in our dataset, NicheNet predicts top ligand-target pairs independent of our data. We should also indicate that NicheNet is a tool to prioritize ligand-target pairs that must then be validated further, as we did in our study. Indeed, the top ligand-target pair, which we nominated based on our data, now replicated in a much larger integrated snRNAseq dataset and prioritized by NicheNet, is also experimentally validated in both iPSC derived pericytes in vitro (**Main Figure 4**), and in transgenic zebrafish models in vivo (**Main Figure 5**).

Reviewer 1: *in response to reviewer concerns (R2, comment 2.1.2) these Quality control of nuclei statements do not detail factors that would be important for the original reviewer concern - that the output of snRNAseq was indeed reporting sequencing from cell nuclei as proposed in the authors narrative. Some ambient RNA cleanup, and ensuring doublet/mito/ribosomal RNA contamination is removed is integral. Regardless of the low number of nuclei input (suggested as a reason for not wanting to remove low quality nuclei by R2), these data must be properly cleaned and low quality nuclei removed. This is standard for the field, and has been omitted now from two versions of this manuscript.*

Response: We thank the reviewer’s emphasis on RNA cleanup and consideration of potential contamination. We agree that more QC checks should be performed, some of which were not done due to the standards at the time when the project was initiated.

In this revision, we have now conducted the RNA cleanup and QC suggested by this reviewer and updated all of our results, including updating of all results in **Main Figures 1, 2, 3A, B, F; Supplementary Tables S1-S26; Supplementary Figures S4-S11, S13 and S14**. Further, these new post-QC data was used in the integration with external datasets, the results of which are shown in **new Main Figure 3F, new Supplementary Tables S31-S34, new Supplementary Figures S15 and S16**. All text has also been updated accordingly. Importantly, after this suggested QC, none of our results change. Further, as detailed above, the new integrated data provide additional support for our findings. We describe the RNA cleanup and QC below.

To address reviewer’s comments and to strengthen our study, we have identified and removed 1355 doublets out of 79,751 nuclei (~ 1.6%) from our dataset using python package Scrublet¹. This package was applied in a recent snRNAseq study by Mathys et al.⁵. We assumed a lenient prior doublet ratio 0.06, identified doublets for each participant separately using the following command and parameters (`scrub_doublets(min_counts=1, min_cells=3, min_gene_variability_pctl=85, n_prin_comps=30)`). For Zhang et al.¹² dataset, where we conducted QC, we also ran Scrublet and removed 3224 doublets out of 76664 nuclei (~4.2%).

Subsequently, for our and external post-QC datasets, we calculated and compared the percent of doublets identified by Scrublet¹ (**Response Table R4**). We further calculated the percent of UMI counts in mitochondrial (MT) genes and ribosomal (RPS and RPL) genes. As shown in **Table R4**, our percents of doublets, MT gene UMI and ribosomal gene UMI are comparable to or better than the other published studies.

Response Table R4: percent of doublets, and UMI counts from mitochondrial and ribosomal genes.

Datasets	% doublets	% MT gene UMI	% RPS gene UMI	% RPL gene UMI
----------	------------	---------------	----------------	----------------

		mean±SD, [min, max]	mean±SD, [min, max]	mean±SD, [min, max]
Grubman et al	0.67	0.53±0.86, [0, 9.42]	0.48±0.37, [0, 5.99]	0.63±0.50, [0, 7.46]
Mathys 2019*	0.47	0±0, [0, 0]	0.42±0.32, [0, 8.67]	0.49±0.44, [0, 10.30]
Yang et al	1.4	0.35±0.42, [0, 5.23]	0.20±0.19, [0, 4.14]	0.12±0.15, [0, 7.19]
Mathys 2023	0.08	2.05±2.28, [0, 14.12]	0.24±0.17, [0, 9.27]	0.21±0.24, [0, 12.57]
Mathys multi-reg	0.03	1.90±2.72, [0, 20.00]	0.22±0.14, [0, 2.86]	0.17±0.19, [0, 3.77]
Sun et al	0.12	1.25±1.16, [0, 5.0]	0.38±0.32, [0, 2.77]	0.39±0.43, [0, 3.78]
Zhang et al	0.10	1.27±1.55, [0, 10.01]	0.25±0.35, [0, 7.59]	0.20±0.44, [0, 8.46]
Is et al [#]	0.16	0.95±1.35, [0, 10]	0.18±0.10, [0, 3.74]	0.08±0.09, [0, 4.53]

The statistics is from both protein-coding and non-coding genes, to be comparable with other datasets.

* This data is from protein coding genes, deposited by the authors. Therefore, the % RPS and RPL UMI might be inflated.

After doublets removal, we re-performed all the downstream analysis using remaining 78,396 nuclei, including cell clustering, differential gene analysis and NicheNet analysis. Corresponding changes have been made and tracked in the main text. All main figures and supplementary figures/tables have been updated accordingly as described above.

Reviewer 1: *Some major concerns from R2 about cell type identification have only been addressed by screenshot from the brainrnaseq.com website, rather than checking in their own system. Validation cannot be provided by using a different dataset - and considering the low power of nuclei sequenced, it was a real shame to see the authors not adequately address this concern using their own samples/system.*

Response: This concern was originally raised as, "... Fig 1C shows expression of contaminating neuron-specific SYT1 in their vascular nuclei and contaminating pericyte-specific PDGFRB in their astrocytes ...". We responded to this by demonstrating SYT1 and PDGFRB expression using external dataset from sorted cell expression at brainrnaseq.com.

We thank the reviewer's request for further demonstration of the expression pattern of SYT1 and PDGFRB using internal datasets. Below, we show the expression pattern of these two genes in two internally generated datasets, as well as additional external datasets, as follows:

- 1) A snRNAseq study from our group, recently published in *Nature Communications*¹³
- 2) Our snRNAseq data generated from vascular and glial enriched nuclear fractions.
- 3) External snRNAseq datasets described in **Response Table R1**

In summary, in all these datasets we demonstrate expression of SYT1 in some non-neuronal and that of PDGFRB in some non-vascular clusters, highlighting that these genes are expressed to some degree in these other clusters and arguing against contamination in our dataset. We describe these results as below.

In 1), we utilized post-mortem TCX samples from 33 donors with progressive supranuclear palsy, a neurodegenerative tauopathy, and control participants¹³. This study used the same experimental methodology as the current study to generate data from frozen brain from the Mayo Clinic Brain Bank samples. As shown in **Response Figure R3**, although PDGFRB is highly expressed in pericytes, it is also expressed in astrocytes. SYT1 has some expression in endothelia. More details can be viewed at the web tool we generated as part of our study¹³ at tools.mayo.edu/PSP_RNAseq_Atlas/.

Response Figure R3: Gene expression profile of *PDGFRB* (Upper) and *SYT1* (Lower) genes in major brain cell types from TCX of 33 donors from Mayo Clinic Brain Bank¹³.

In 2), we generated new snRNAseq data from glial enriched or vascular enriched nuclei from frozen human temporal cortex. Nuclei isolation and purification from frozen human brain were performed using established protocols. NEUN- and LEF1 antibodies were used to enrich for glial cells and vascular cells in FANS, respectively. FANS gating strategies are illustrated in **Response Figures R4A** and **R5A** for sorting LEF1+ and NEUN- nuclei, respectively. We applied standard alignment and quality control pipelines to the samples, separately. After QC, we obtained the transcriptome profile of 1,973 NEUN- nuclei and 2,487 LEF1+ nuclei. We annotated the nuclei using commonly used brain cell type markers (**Response Figures R4B-C** for glial fractions; **Figure R5B-C** for vascular fractions). We observed expression of *SYT1* in glial cells (**Response Figure R4D**) and *PDGFRB* expression in astrocytes (**Response Figure R5D**).

Response Figure R4: The expression of *SYT1* in major brain cell types based on vascular enriched snRNAseq data. Nuclei isolated and purified from human temporal cortex were enriched for vascular nuclei using LEF1 as vascular target via FANS. A) Dot plots demonstrate gating strategy to obtain LEF1⁺ population from nuclear samples. B) snRNAseq data were generated and QCed as previously described. Cells were annotated using commonly used brain cell type markers and C) visualized in UMAP. D) Although the expression is not as strong as observed in neurons, *SYT1* expression could be detected in other brain cell types such as oligodendrocytes, microglia, perivascular fibroblasts, endothelia and pericytes (Oligo: oligodendrocytes; Micro: microglia; Astro: astrocytes; PV.F: Perivascular fibroblasts; OPC: oligodendrocyte precursor cells, Peri: pericytes; Endo: endothelia; CAM: capillary associated microglia)

Response Figure R5: The expression of *PDGFRB* in major brain cell types based on glia enriched snRNAseq data. Nuclei isolated and purified from human temporal cortex were enriched for glia nuclei using NEUN- as glial target via FANS. A) Dot plots demonstrate gating strategy to obtain NEUN⁻ population from nuclear samples. B) snRNAseq data were generated and QCed as previously described. Cells were annotated using commonly used brain cell type markers and C) visualized in UMAP. D) *PDGFRB* expression could be detected in astrocytes and OPCs (Oligo: oligodendrocytes; Micro: microglia; Astro: astrocytes; OPC: oligodendrocyte precursor cells)

Regarding 3), we illustrated in **Response Figure R6** the expression of *PDGFRB* and *STY1* in external datasets that were described previously. For *PDGFRB*, while pericytes/vascular cells have the highest expression, it is also expressed in astrocytes in Is et al, Grubman et al⁴, Mathys et al. 2019³, Mathys et al. 2023 PFC region⁵, Mathys et al. 2023 multi-region⁵, and Zhang et al.¹² datasets. For *STY1*, its expression is observed in pericyte/vascular cells in Is et al, Mathys 2023 et al. PFC region⁵, Sun et al.¹⁴ and Zhang et al.¹², and in astrocytes in Is et al, Mathys et al. 2023 PFC region⁵, Mathys et al. 2023 multi-region⁵ and Zhang et al.¹².

Response Figure R6: Distribution of log-normalized gene expression of *PDGFRB* (left) and *SYT1* (right) in different studies.

Reviewer 1: Response Fig 4 missing.

Response: We checked the previous rebuttal letter and identified the Response Figure 4. For reviewer's convenience, we show that figure below.

Previous Response Figure 4: Expression profile of *PLCG2* across all clusters in all donors (lower panel) and split by diagnosis (upper panel).

Reviewer 1: there are no purity measurements for iPSC-derived pericytes (see Supplementary Fig S17) - such measurements are imperative for interpretation of such assays

Response: We agree with the reviewer that adding purity measurements for iPSC-derived pericytes would strengthen our experimental validation. Meanwhile, we would like to make a note that the goal of supplementary Figure S17 (**new Supplementary Figure S19**) is to demonstrate *SMAD3* RNA localization in iPSC derived pericyte cells using RNAscope. To address this reviewer's concerns, we examined three possible contaminations - mycoplasma contamination, feeder cell contamination, and differentiating into wrong cell types.

Regarding mycoplasma contamination: The iPSC lines have already been characterized in published studies (**Supplementary Table S40**) and we note that the absence of mycoplasma contamination is part of our standard operating procedures prior to differentiation. We checked the mycoplasma contamination in iPSCs via MycoAlert® PLUS Mycoplasma Detection Kit (Lonza, LT07-710) and compared our readouts with control samples from MycoAlert® Assay Control Set (Lonza, LT07-518). Using MycoAlert™ PLUS Assay, mycoplasma activity can be detected by exploiting certain enzymes. ADP is converted to ATP through the reaction of enzymes with MycoAlert™ PLUS Substrate. As a result of measuring ATP levels before and after the addition of MycoAlert™ PLUS Substrate, a ratio can be determined whether mycoplasma is present or not. With uninfected samples, the test is designed to produce ratios less than 1; with mycoplasma-infected samples, it produces ratios greater than 1. A summary of our mycoplasma test is presented in **Response Table R5**. All of our readings from luminometer were below 1, which is an indication of lack of mycoplasma

contamination. This is also confirmed by very high reading from positive control sample. **Table R5** was added as new **Supplementary Table S41**.

Response Table R5: Results of mycoplasma contamination test.

		1st Read	2nd Read	2nd Read/1st Read	mean
	positiveC.	7901	473366	59.91216	
	negativeC.	7024	1534	0.218394	
iPSC	MC0031	17939	6680	0.372373	0.386693
		17194	6895	0.401012	
iPSC	MC0430	23653	7434	0.314294	0.357269
		25407	10169	0.400244	
iPSC	MC0429	8805	3089	0.350823	0.375728
		8549	3425	0.400632	
iPSC	MC0018	8889	4318	0.485769	0.49529
		9665	4879	0.504811	

Readings at 2nd/1st reads higher than 1 at this assay is an indication of mycoplasma contamination. None of our iPSCs reached higher values.

We have added the following sentence to the **Tier 3 Methods: Human In Vitro data generation and analysis, section 30. Human iPSC lines and pericyte differentiation, first paragraph**:

“Mycoplasma contamination in iPSCs were checked via MycoAlert® PLUS Mycoplasma Detection Kit (Lonza, LT07-710) and compared our readouts with control samples from MycoAlert® Assay Control Set (Lonza, LT07-518) (**Supplementary Table S41**).”

Regarding feeder cell contamination: Our differentiation protocol does not include feeder cells to support the growth of iPSCs and only comprises iPSCs and follow-up differentiating cells. Our protocol has been adapted from (<https://doi.org/10.1038/s41591-020-0886-4>) and based on sequential addition of growth factors to iPSCs seeded on matrigel.

Regarding iPSC differentiation purity: From the same batch of cells, we performed extensive validation experiments to confirm the differentiation of iPSCs to pericytes through qPCR, ICC, and flow cytometry that are summarized in **Figure 4**. In **Figure 4B**, we show the differentiation of iPSCs to pericytes through upregulation of pericyte markers, PDGFRB, and NG2 while downregulation of iPSC marker, TRA1-60 via flow cytometer. We demonstrated that these cells express PDGFRB, a well-known protein expressed highly by pericytes. We also showed upregulation of pericytic *PDGFRB* and vascular *LEF1* expression via qPCR in **Figure 4D**. These results validate the identity of differentiated iPSCs as pericytes. Further, additional interrogation experiments were completed. We checked the expression of major brain cell type markers in iPSC derived pericytes through qPCR to confirm the identity of pericytes following differentiation. Compared to iPSCs, we observed around 50-fold increase in pericytic marker expression after differentiation (**Response Figure R6**, n=3). Except for the OPC marker, *PDGFRA*, none of the brain markers showed consistent increase in the gene expression upon differentiation. We are not concerned about this increase in *PDGFRA* as OPCs are known to also express *OLIG2* gene, which is not expressed in our iPSC-derived pericytes.

In summary, our iPSC lines are devoid of mycoplasma, and our pericyte differentiation protocol yields pericytes with high efficiency. **Response Figure R6** was added to our revised manuscript as **Supplementary Figure S29**. We have added the following sentence to the Main Text, Tier 3 Methods: Human In Vitro data generation and analysis, Section 31. Validation of pericyte differentiation, last paragraph:

“The purity of pericytes was also checked via qPCR after differentiation with a panel of comprehensive qPCR probes that include several brain cell type markers. (**Supplementary Figure S29**).”

Response Figure R6: Fold change in gene expression profile of iPSCs following differentiation. To confirm the identity of differentiated cells’ profile, expression of brain cell type marker genes was analyzed via qPCR: *AQP4* and *GFAP* for astrocytes; *P2RY12* and *PTPRC* for microglia; *PDE4A* and *RBFOX3* for neurons; *MOBP* and *MOG* for oligodendrocytes; *OLIG2* and *PDGFRA* for OPCs; *ESAM*, *PECAM1* and *VWF* for endothelia; *PDGFRB* for pericytes; and *LEF1* for vascular cells.

Reviewer 1: *Supp Fig S8* - what are the gray circles and gene numbers (e.g. 4732) - are these total DEGs from the current study across all cells? This is another example of a poor legend making the data uninterpretable.

Response: We would like to thank the reviewer for this question. We added the following sentences to the figure legend of Supplementary Figure S8:

“**Supplementary Figure S8: Comparison of our study with previous study.** Number of overlapping DEGs between astrocytic clusters in our study (cl.8, cl.112 and cl.31) and those from Lau et al. study¹. The gray circles and the numbers indicate the detected genes in both Lau et al astrocyte cluster and our astrocytic clusters cl.8, cl.11 and cl.31 respectively. Enrichment p-value is from Fisher’s exact test.”.

Reviewer 1: *Supp Figure S6* raises additional concerns (originally raised by R1 and R2) about the power of these studies. Given the low numbers of cells here (which contrary to arguments from the authors that the total glio-vascular unit number being high - are still very low for each individual cell type, making investigations of interactions quite difficult), the variability of different AD sub-types now drops the *n* values for each sub-type considerably. While it is too late to correct for this experimentally, it is a real shame that the authors did not choose a clearly defined patient cohort before embarking on this study. A limitations statement should be included now, as some of the differentially expressed genes presented may be present in only one patient group, or validated incorrectly in postmortem tissue from another patient cohort. (see gross differences in a small subset of genes in *Supp Fig S12*, panel B)

Response: We have extensively addressed the power issue as detailed above. We note that we had already included a

paragraph of study limitations in the prior submission. In this, submission, we expanded this section and also added text to reflect the findings from the external studies now integrated into our manuscript. This modified text is in **Discussion, penultimate paragraph**, already added above to address another comment by this reviewer.

We would like to emphasize that validation of gene expression on postmortem brain tissue were completed on adjacent brain regions from the initial snRNAseq patient cohort, i.e. separate brain tissue but from the same donors. Donor information and as well as Delta Ct values for each validated gene are summarized and provided in **Supplementary Tables S1 and S27**.

In addition, we would like to highlight that the six genes selected for qPCR validation in **Supplementary Figure S12** met several criteria 1) they are differentially expressed (DE) genes in vascular cell clusters, 2) they are the top targets of ligands in each of the three astrocytic clusters, where the ligands were again DE genes in astrocytic clusters; 3) they have known important vascular biological functions. Indeed, the top vascular target-astrocytic ligand pair, *SMAD3-VEGFA* are further replicated in external human brain RNAseq datasets, through in vitro iPSC and in vivo zebrafish experimental studies.

Reviewer 1: *missing figure legends - e.g. Supp Fig S23, but also lacking considerable information on many (e.g. Supp Figs S13, S18, S20, S24-S26)*

Response: We thank the reviewer's requests on modifying the supplementary figure legends, have made changes to these legends which currently read as follows:

“Supplementary Figure S13: SMAD3 is a predicted target for multiple astrocytic ligands in AD from NicheNet. DEG (differentially expressed gene) is from comparing AD and control nuclei using MAST. The figure outlines the connections between astrocytic ligand genes to *SMAD3* based on the interaction strengths in three astrocytic clusters. *SMAD3* ligands include well-known AD genes in astrocytes such as *APOE*, *MAPT*, *PSEN1*, and *APP*. Line thickness: the Astrocytic gene's connection strength to *SMAD3*; Red Blue gradient heatmap: Fold change of differential expressed gene expression in AD vs control (downregulated (blue) and upregulated (red)); Radius of circle: FDR corrected significance between AD vs controls.”

(Old Supplementary Figure S18): “Supplementary Figure S20: Experimental design for the treatment of iPSC derived pericytes. The flowchart outlines the iPSC experimental design. Four patient-derived iPSC lines ($n_{AD}=2$, $n_{CONTROL}=2$) were used to differentiate into pericytes. Each patient derived iPSC was seeded on all wells of a 6 well plate. Seeded cells were differentiated into pericytes using established protocols as depicted. Cells in each well is accounted as one technical replicate and merged into one single tube to a total of 6 tubes. Collected cells were reseeded onto a 24 well plate for treatment experiments. Concentrations that were used for each stimulant were as follows: 50 ng/mL, 100 ng/mL, 200 ng/mL recombinant VEGF; 250 nM aggregated A β ; and VEGFR2 inhibitor cocktail that contains (Semaxanib SU5416 (10 uM, SelleckChem S2845), Tivozanib AV-951 (10 uM, SelleckChem S1207), and ZM 306416 (10 uM, SelleckChem, S2897). Treatment durations were as follows for each treatment: 6 hours, 12 hours, and 24 hours. Following treatment, RNA was collected and expression of selected genes were checked via qPCR.”

(Old Supplementary Figure S20): “Supplementary Figure S22: Pericyte SMAD3 levels at different VEGF treatment concentrations. All timepoints from treatment with VEGF are compiled together and checked against untreated controls. 50 ng/mL and 100 ng/mL VEGF treatment caused significant downregulation of SMAD3 expression in iPSC derived pericytes compared to control conditions ($p<0.05$).”

(Old Supplementary Figure S24): “Supplementary Figure S26: Structural comparison of VEGFR2 in human and zebrafish. We made a structural comparison of human VEGFR2 with zebrafish *vegfr2* using Swiss 3D Model prediction algorithm. Our *in silico* analysis predicts that the catalytic domain is highly conserved in 3D and drugs targeting this domain would be effective for both species.”

(Old Supplementary Figure S25): “Supplementary Figure S28: FANS Gating strategy of intact nuclei from frozen human brain. Dot plots demonstrate gating strategy to obtain HNA+ population from nuclear samples. Isolation of nuclei were performed as previously established and purified via FANS as depicted. Sorted nuclei were directly used to generate snRNAseq data, or to seed for validation experiments such as RNAscope and qPCR.”

(Old Supplementary Figure S26): “Supplementary Figure S27: Colocalization of A) pERK and B) ZO-1 on GFP fcy is visualized. In order to quantify the colocalization of markers, ImageJ software's colocalization module was used to generate two-channel composites, R(and) colocalization analyses, and Fay translation into correlation values. Pairwise comparisons were performed with unpaired parametric t test with Welch's correction.”

Reviewer 1: *Supp Fig S5 would be better visualized by coloring variables in different colors and plotting integrated - e.g. ADvNormal, or male v female. Also the authors should appropriately annotate these figure panels - leaving sex as '0' and '1' is not a difficult output to correct. Similarly, what does APOE4 positive/negative mean - are these homozygotes or heterozygotes? for an example of an easier to view output taking advantage of this suggestion - check Supp FigS23, panel B, which correctly plots samples on the same UMAP colors by treatment.*

Response: We agree with the reviewer that the figure legend should be more explanatory. We have indicated on the figure the labels of AD vs Normal, Male vs Female, APOE4 Yes vs No. In addition, in the figure legend, we stated “APOE4 Yes means either homozygotes or heterozygotes”. Per reviewer's request, we have plotted the integrated view colored by variables, instead of split view. We include the modified **Supplementary Figure S5** below for ease for the reviewer.

Supplementary Figure S5: snRNAseq nuclei clusters on UMAP by features: Top left: Nuclei clusters displayed on UMAP reduced dimension space, annotated as cluster number followed by cell type. Exc: excitatory neuron; Inh: inhibitory neuron; Oli: oligodendrocyte; OPC: oligodendrocyte progenitor cell; End: endothelia; Per: pericyte; Fib: fibroblast; Ast: astrocyte; Mic: microglia. Top right and bottom: Distributions are colored by variables. “APOE4 Yes” means either homozygotes or heterozygotes.

Reviewer 1: *Supp Fig S14 - how do these data show that AGT expression is restricted to human temporal cortex astrocytes? AGT is expressed by all three astrocyte clusters - suggesting it is expressed in all brain regions included in this dataset.*

Response: We would like to thank the reviewer for their attention. The sentence has been modified to “In human temporal cortex, *AGT* expression is restricted to astrocytes”. Because of the order of the words, the sentence could be easily misunderstood, which was the case. Only human temporal cortex has been utilized in this analysis and we detected *AGT* expression only in astrocytes.

Reviewer 1: scale bars missing from micrographs in Supp Fig S3, S17

Response: We thank that the reviewer caught the problem of missing scale bars. For **old Supplementary Fig S3** (new **Supplementary Figure S1**), we have replaced it with white scale bar in each panel (shown below to the right lower corner of each panel) and added the following sentence to the figure legend “The white scale bar in each panel equals 200 μm ”. For **old Supplementary Fig S17** (new **Supplementary Figure S19**), we have replaced it with white scale bar in each panel (shown below to the right lower corner of each panel) and added the following sentence to the figure legend “The white scale bar in each panel equals 100 μm ”.

Supplementary Figure S1: Neuropathological measures in AD and control samples. Left panels depict representative immunohistochemistry pictures from an Alzheimer’s patient and control brain amygdala tissue slides for H&E, thioflavin-S and TDP-43 staining. The white scale bar in each panel equals 200 μm . Right sided panels from top to bottom show distribution of Thal phase, Braak stage and presence or absence of TDP-43 in the cohort of 12 AD and 12 control brains

Supplementary Figure S19: Validation of *SMAD3* gene expression through RNAscope at iPSC derived pericytes. Negative Control: Bacterial *dapb*, Positive Control: *PPIB*, Orange: *SMAD3*, Green: Actin, Blue: Nuclei (DAPI). The white scale bar in each panel equals 100 μm .

Reviewer 1: full blots should be included, with MW markers - e.g. Supp Fig S1

Response: We thank the reviewer's request. We have added the full blots with MW markers (the following panel) to new Supplementary Figure S2.

Reviewer #2

Reviewer 2: *The authors have addressed my comments and this is a solid body of work. Though some concerns remain about the overall robustness of their lenient DEG analyses (perhaps necessary because of low vascular nuclei captured), the additional iPSC and zebrafish experiments strengthen their main claims of astrocyte-to-pericyte interactions in AD.*

Response: We thank the reviewer's recognition of our work and their positive comments.

Reviewer 2: *The correlation plot in Fig. 5C, right, is confusing: are the authors correlating the correlation coefficients? Is the intent to correlate the levels of pERK/GFP and ZO-1/GFP and provide one correlation coefficient R?*

Response: The reviewer's assumption is correct since our intent has been to correlate the co-localization coefficient of pERK/GFP with ZO-1/GFP. Through binding to VEGFR2, VEGF activates RAS-mediated S/T protein kinase and downstream MAP2/ERK kinase (MEK); leading further phosphorylation of ERK1/2, whose activation have pleiotropic effects on several pathways. Through this figure, we demonstrate that decrease of pERK signaling deteriorates the integrity of zebrafish brain blood vessels that are mediated by tight junctions (ZO-1).

For clarity, we have added the label "E" to the following figure in **main Figure 5**, with figure legend "**E**) Colocalization coefficients of pERK/GFP and ZO-1/GFP are correlated. With decreased pERK, ZO-1 also reduces, similarly, high pERK expressing vascular cells also has high ZO-1 levels. GFP always marks the vasculature, and therefore is common to both separate correlations".

Reviewer #3

Reviewer 3: *I appreciate the efforts made by the authors in addressing the previous comments and conducting additional validation experiments. The manuscript has certainly improved. However, I still have a two remaining concerns:*

Response: We appreciate the reviewer's positive feedback of our efforts in conducting experimental validations and addressing previous comments.

Reviewer 3: *It's great to see the authors referencing the preprint by Zhang et al. to support their findings regarding SMAD3 gene expression in pericytes of AD and VEGFA downregulation in astrocytes. However, I'd like to suggest considering several other peer-reviewed brain/brain vascular atlases that are published in well-respected journals. Some of these include PMID: 35165441, PMID: 31042697, PMID: 31768052, PMID: 37774677, PMID: 31932797. It might be beneficial to explore and compare the expression of those findings across these atlases instead of focusing primarily on one preprint that confirms the own findings. This could provide a more comprehensive view of the subject. I appreciate the validation experiments carried out by the authors. However, I do still have some concerns about the low nuclei*

numbers. It might be beneficial for the authors to use the available atlases that I mentioned above to further validate their claims and provide additional support for their findings. This could help reinforce the study's conclusions. It would also be beneficial to state more clearly the limitations of the study

Response: We greatly thank the reviewer's suggestion of adding external snRNAseq datasets to strengthen our study. We have collected and listed in **Response Table R1** of these external datasets. As described at the beginning of this document, we integrated pericytes and astrocytes from external datasets with ours and performed integration, subclustering and differential gene expression analysis. We have added these results to the **Main text, Main Figure 3F, Supplementary Tables S31-S34, Supplementary Figures S15-S16, Main Results, Abstract and Discussion**. Our manuscript thus now includes analyses of an integrated dataset of 150,664 astrocyte and 4,730 pericyte nuclei from 6 or more brain regions. In these integrated datasets, we confirmed our findings of up-regulation in AD of *SMAD3* and down-regulation of *VEGFA* in the largest pericyte and astrocyte clusters, respectively. Importantly, we were able to characterize the pericyte and astrocyte subclusters with these expression changes and demonstrate their applicability in different brain regions, as well as studies of both selected vascular and unselected nuclei from AD and control brains. We also expanded the paragraph on the limitations of our study as described above.

References used in this document:

- 1 Wolock, S. L., Lopez, R. & Klein, A. M. Scrublet: Computational Identification of Cell Doublets in Single-Cell Transcriptomic Data. *Cell Syst* **8**, 281-291.e289 (2019). <https://doi.org:10.1016/j.cels.2018.11.005>
- 2 Yang, A. C. *et al.* A human brain vascular atlas reveals diverse mediators of Alzheimer's risk. *Nature* **603**, 885-892 (2022). <https://doi.org:10.1038/s41586-021-04369-3>
- 3 Mathys, H. *et al.* Single-cell transcriptomic analysis of Alzheimer's disease. *Nature* **570**, 332-337 (2019). <https://doi.org:10.1038/s41586-019-1195-2>
- 4 Grubman, A. *et al.* A single-cell atlas of entorhinal cortex from individuals with Alzheimer's disease reveals cell-type-specific gene expression regulation. *Nat Neurosci* **22**, 2087-2097 (2019). <https://doi.org:10.1038/s41593-019-0539-4>
- 5 Mathys, H. *et al.* Single-cell atlas reveals correlates of high cognitive function, dementia, and resilience to Alzheimer's disease pathology. *Cell* **186**, 4365-4385.e4327 (2023). <https://doi.org:10.1016/j.cell.2023.08.039>
- 6 Zhou, Y. *et al.* Human and mouse single-nucleus transcriptomics reveal TREM2-dependent and TREM2-independent cellular responses in Alzheimer's disease. *Nat Med* **26**, 131-142 (2020). <https://doi.org:10.1038/s41591-019-0695-9>
- 7 Sun, N. *et al.* Single-nucleus multiregion transcriptomic analysis of brain vasculature in Alzheimer's disease. *Nature Neuroscience* **26**, 970-982 (2023). <https://doi.org:10.1038/s41593-023-01334-3>
- 8 Hafemeister, C. & Satija, R. Normalization and variance stabilization of single-cell RNA-seq data using regularized negative binomial regression. *Genome Biology* **20**, 296 (2019). <https://doi.org:10.1186/s13059-019-1874-1>
- 9 Korsunsky, I. *et al.* Fast, sensitive and accurate integration of single-cell data with Harmony. *Nat Methods* **16**, 1289-1296 (2019). <https://doi.org:10.1038/s41592-019-0619-0>
- 10 Squair, J. W. *et al.* Confronting false discoveries in single-cell differential expression. *Nature Communications* **12**, 5692 (2021). <https://doi.org:10.1038/s41467-021-25960-2>
- 11 Browaeys, R., Saelens, W. & Saeys, Y. NicheNet: modeling intercellular communication by linking ligands to target genes. *Nature Methods* **17**, 159-162 (2020). <https://doi.org:10.1038/s41592-019-0667-5>
- 12 Zhang, L. *et al.* Single-cell transcriptomic atlas of Alzheimer's disease middle temporal gyrus reveals region, cell type and sex specificity of gene expression with novel genetic risk for MERTK in female. *medRxiv* (2023). <https://doi.org:10.1101/2023.02.18.23286037>
- 13 Min, Y. *et al.* Cross species systems biology discovers glial DDR2, STOM, and KANK2 as therapeutic targets in progressive supranuclear palsy. *Nat Commun* **14**, 6801 (2023). <https://doi.org:10.1038/s41467-023-42626-3>
- 14 Sun, N. *et al.* Single-nucleus multiregion transcriptomic analysis of brain vasculature in Alzheimer's disease. *Nat Neurosci* **26**, 970-982 (2023). <https://doi.org:10.1038/s41593-023-01334-3>

REVIEWERS' COMMENTS

Reviewer #1 (Remarks to the Author):

The authors have addressed many concerns as raised by the reviewers in the second version of the manuscript. I am still not 100% happy with the response to the power questions, particularly as it relates to individual cell types - which I detail with respect to astrocytes below:

1. the inclusion of additional datasets is a great improvement on the manuscript. It was curious to note that the authors comment on additional datasets at line 648 with high astrocyte capture, but did not include these in the integration analyses - while the Zhang dataset is include (~6500 astrocytes), the Sadick dataset was not (~41,000 astrocytes). Similarly, the statement that "We note that the Mathys 20235 multi-region study already included astrocyte nuclei from PFC region from 41 participants. Therefore, we did not include Mathys 2023 PFC dataset in this integrated analysis, which encompassed 149,558 astrocytes from 427 participants that would overweigh the PFC region." is both confusing and concerning - if these astrocytes are from the same brain region, there is no valid reason to make the statement that the added numbers would overweigh the PFC region - do the authors mean that including non-PFC astrocytes would be a problem (which could quite possibly be true)? If they mean increasing numbers of astrocytes from the same brain region would somehow alter results - is only true to the level that artifacts of underpowered clustering would be removed. This should be clarified, and if the later, these analyses should be re-done to test this concern.

Given the total astrocyte datasets (on average) provide only 70-985 astrocytes from each donor, one would argue that these numbers are still quite low. (it should be noted that while the Mathy's datasets are large at the conglomerate level, they provide on 350-452 astrocytes/donor, making them among the least powerful of these datasets integrated, so caution should be taken when relying on their numbers alone. There still remains no QC metrics of data plots showing representation across donors in each cluster - are they equally represented across the datasets/donors? This remains a major oversight.

I appreciate that this is a very field-specific concern, but it remains, and limits the scope of this research to the community.

Similarly, there is no validation of key insights from these data integration - for example the receptor-ligand interaction calculations should be validated with true interaction experiments (e.g. proximal ligation assay), not in situs of marker genes in adjacent cells (see Figure 3, panel C). I will not however that these concnrs are somewhat mitigated by the elegant experiments completed in fish that are depicted in Fig 5 (however the exact interaction of SMAD3 with putative binding partners is still lacking).

I am happy to relinquish these concerns if other reviewers are more positive on the revised manuscript, but these doubts remain for me personally.

Reviewer #2 (Remarks to the Author):

The authors have addressed my comments.

Reviewer #3 (Remarks to the Author):

I thank the authors for thoroughly addressing my comments and undertaking further validation experiments. The manuscript has significantly improved, and I have no further comments.

Identification of novel gliovascular interactions perturbed in Alzheimer's disease and conserved in cross-species models.

Responses to Reviewers' Comments

NCOMMS-23-47665A

We are delighted to receive the communication from the Editors of the decision to publish our manuscript. Furthermore, we are excited to receive the very positive feedback from all our Reviewers, where second and third reviewers have no further comments. Our Editors requested one final revision from us to address the remaining comments from Reviewer #1 and also the editorial requests, all of which we have completed in this final revised version.

The Reviewers' individual comments are highlighted in *italic font* below. For each of our Main and Supplementary files, we provide both a version with all changes tracked and a version with accepted changes. Text changes to the manuscript we depict in this document is in Arial font. We also completed the Editorial Checklist and provide a Featured Image in this submission.

With these edits and others delineated in this document, we believe that we have addressed every comment. We look forward to the next steps in the publication of our manuscript.

Reviewer #1:

The authors have addressed many concerns as raised by the reviewers in the second version of the manuscript. I am still not 100% happy with the response to the power questions, particularly as it relates to individual cell types - which I detail with respect to astrocytes below:

We appreciate this reviewer's acknowledgement of our addressing of the concerns.

Reviewer 1: *The inclusion of additional datasets is a great improvement on the manuscript. It was curious to note that the authors comment on additional datasets at line 648 with high astrocyte capture, but did not include these in the integration analyses - while the Zhang dataset is include (~6500 astrocytes), the Sadick dataset was not (~41,000 astrocytes).*

Response: We thank the reviewer's recognition of the great improvement with the inclusion of additional datasets. We did not include Sadick et al. dataset ¹ based on the following considerations. **1)** Their AD and control participants are all APOE ϵ 2/3, whereas ours is a mixture of APOE ϵ 2/3/4 having ~40% (10 out of 24) participants with one or two copies of APOE ϵ 4. **2)** They performed enrichment of astrocytic nuclei whereas ours is un-enriched libraries. **3)** The brain region of their study is prefrontal cortex (PFC), which is already represented in Mathys's 2023 study from un-enriched library preparation. **4)** Sadick et al. post-QC UMI count matrix with information of cell type assignment was not available.

Nevertheless, we pulled the differential expression results of *VEGFA* expression in AD participants in Sadick et al. from their supplemental table "1-s2.0-S0896627322002446-mmc7.xlsx" at ncbi.nlm.nih.gov/pmc/articles/PMC9167747/. As shown in Response **Table R1**, in four largest astrocyte clusters, *VEGFA* expression is down in AD participants. This is consistent with our finding in our dataset and in the integrated dataset, where *VEGFA* is down in the largest astrocyte cluster in AD participants.

To further address the critique from Reviewer#1, we added the following sentence in our manuscript with citation in Discussion, penultimate paragraph:

"Notably, *VEGFA* was also down in AD brains in the largest clusters from enriched astrocytic nuclei in Sadick et al.¹⁰⁷, which was not included in our integrated analyses due to the differences in the *APOE* ϵ 4 distribution of this dataset and their enrichment approach. Importantly, in our integrated analyses, we were able to...".

Table R1: DE results of *VEGFA* in Sadick et al.

Downregulated_genes_in_AD_patients	Pval	log2FC	LEN_so_astro_r2_cluster
VEGFA	1.42E-34	-0.452381	0
VEGFA	2.69E-12	-0.267842	1
VEGFA	4.09E-18	-0.623803	2
VEGFA	2.86E-20	-0.713271	3
VEGFA	1.18E-07	-0.653581	6

Reviewer 1: *Similarly, the statement that "We note that the Mathys 20235 multi-region study already included astrocyte nuclei from PFC region from 41 participants. Therefore, we did not include Mathys 2023 PFC dataset in this integrated analysis, which encompassed 149,558 astrocytes from 427 participants that would outweigh the PFC region." is both confusing and concerning - if these astrocytes are from the same brain region, there is no valid reason to make the statement that the added numbers would outweigh the PFC region - do the authors mean that including non-PFC astrocytes would be a problem (which could quite possibly be true)? If they mean increasing numbers of astrocytes from the same brain region would somehow alter results - is only true to the level that artifacts of underpowered clustering would be removed. This should be clarified, and if the later, these analyses should be re-done to test this concern.*

Response: We thank the reviewer's questions. We would like to clarify our statement and logic. Astrocytes are known to be heterogenous across different brain regions⁴. To have a more complete picture of *VEGFA* expression in astrocytes in AD, we integrated external datasets from different brain regions by clustering all astrocytic nuclei, and then performing differential gene expression analysis in each cluster separately. If astrocytes from a particular brain region, e.g. PFC, is heavily over-represented in terms of number of nuclei/participants/studies, we expect that **1)** astrocyte clustering will be largely determined by PFC astrocytes, i.e., the clusters will reflect more PFC astrocyte subtypes but less from other regions, and **2)** the subclusters will be mostly composed of PFC astrocytes from multiple studies that will bias the joint-analysis results. Due to these considerations, we aimed to have a balanced representation of nuclei from different brain regions.

Reviewer 1: *Given the total astrocyte datasets (on average) provide only 70-985 astrocytes from each donor, one would argue that these numbers are still quite low. (it should be noted that while the Mathy's datasets are large at the conglomerate level, they provide on 350-452 astrocytes/donor, making them among the least powerful of these datasets integrated, so caution should be taken when relying on their numbers alone.*

Response: We appreciate that the reviewer pointed out this aspect of number of astrocytes per participants. It is true that hundreds of astrocyte nuclei per participant may not reveal the complete picture of astrocytic subtypes/states. On the other hand, we found in our dataset the down-regulation of *VEGFA* in AD in the largest astrocyte cluster. Therefore, we expect the current number of astrocytes per participant (around hundreds) should not be concerning. In fact, the total astrocytic nuclei in the largest cluster is 81,431 nuclei.

Reviewer 1: *There still remains no QC metrics of data plots showing representation across donors in each cluster - are they equally represented across the datasets/donors? This remains a major oversight.*

Response: We agree that showing the representation across donors in each cluster would be helpful. For the analysis of Is et al. dataset alone, we had already listed comprehensively in **Supplemental Table S3** the number of nuclei from each donor in each cluster. For the integrated analysis, we added **Supplemental Tables S42-S43** to show these numbers in astrocyte and pericyte clusters, respectively. We added the following text to Methods, **Section 20. Integrating external snRNAseq datasets**, penultimate paragraph, last sentence.:

"Numbers of and testing for over-representations of nuclei from donor(s) in each cluster in integrated astrocyte and pericyte integrated datasets are shown in **Supplemental Tables S42-S43**, respectively."

To test if there are over-representations of nuclei from certain donor(s) in each cluster in our dataset, we followed the approach of Wang et al.² to perform one-sided Fisher's Exact Test (FET). By applying threshold of 0.05 for

corrected FET p-value < 0.05 and enrichment fold change (EFC) > 4, in astrocyte clusters 8, 11, 31 and vascular clusters 25, 26, 30, there are 1, 0, 1, 0, 0, 0 donor with over-representation of their nuclei, respectively. The results for all clusters are added to **Supplemental Table S3**. The following stacked bar plot (**Response Figure R1**) for visualization was added to **Supplemental Figure S4**. In summary, there is no donor over-representation in any of the vascular clusters. Amongst the astrocyte clusters, a single donor was over-represented in 2 of 3 clusters, which is unlikely to influence the results.

Response Figure R1: Stacked bar plot of nuclei from each donor in each cluster in Is et al. dataset.

For the integrated analysis of astrocytes, we generated the enrichment data and added it as new **Supplemental Table S42**. In astrocyte cluster 0, we did not observe significant (adjusted p < 0.05 and EFC > 4) over-representation of nuclei of any donor. Similarly, for the integrated analysis of pericytes, we generated and included the enrichment data in **Supplemental Table S43**. We did not observe significant over-representation of nuclei of any donor in pericyte cluster 0.

Reviewer 1: *I appreciate that this is a very field-specific concern, but it remains, and limits the scope of this research to the community.*

Response: We hope that our responses above have alleviated the reviewer’s concerns, regarding Sadick’s dataset, the reasons to not overweigh PFC astrocytes in the integrated analyses, the number of astrocyte nuclei per participants, and the representation across donors in each cluster.

Reviewer 1: *Similarly, there is no validation of key insights from these data integration - for example the receptor-ligand interaction calculations should be validated with true interaction experiments (e.g. proximal ligation assay), not in situ of marker genes in adjacent cells (see Figure 3, panel C). I will not however that these concerns are somewhat mitigated by the elegant experiments completed in fish that are depicted in Fig 5 (however the exact interaction of SMAD3 with putative binding partners is still lacking).*

Response: We thank the reviewer’s question and their acknowledgement of the elegance of the in vivo experiments. We should again emphasize that the predicted interactions between VEGFA-SMAD3 does not imply an actual physical interaction between these molecules but rather an interaction along a signalling axis. This signal transduction does not need to involve a physical ligand-receptor interaction between these two molecules, i.e. there might be multiple steps of signal transduction between VEGFA to SMAD3. Therefore, for validation, we designed the zebrafish experiments where we showed that by blocking VEGF receptor VEGFR2, VEGF signaling in astrocyte is reduced and phosphorylated SMAD3 is increased in pericytes. Further, the brain vasculature integrity is impaired, confirming the ligand-target relationship of VEGF and SMAD3 along a signalling axis in zebrafish. In addition, our iPSC experiment (Figure 4) showed that blocking VEGFR2 (KDR) increased SMAD3 expression in iPSC differentiated pericytes. While we agree that discovery of putative binding partners for SMAD3 would be of interest, this is out of the scope of this study. We added the following sentence to address this point in Discussion, penultimate paragraph:

“Additional efforts are needed to identify any binding partner(s) of SMAD3 that responds to VEGFA signaling.”

Reviewer 1: *I am happy to relinquish these concerns if other reviewers are more positive on the revised manuscript, but these doubts remain for me personally.*

Response: We thank the reviewer's comments. Indeed, Reviewers 2 and 3 have no further comments.

Reviewer #2: *The authors have addressed my comments.*

Response: We thank the reviewer's recognition of our efforts addressing the comments.

Reviewer #3: *I thank the authors for thoroughly addressing my comments and undertaking further validation experiments. The manuscript has significantly improved, and I have no further comments.*

Response: We thank the reviewer's positive comments.

References Used in this Response Document:

- 1 Sadick, J. S. *et al.* Astrocytes and oligodendrocytes undergo subtype-specific transcriptional changes in Alzheimer's disease. *Neuron* **110**, 1788-1805.e1710 (2022). <https://doi.org:10.1016/j.neuron.2022.03.008>
- 2 Wang, M. *et al.* Guidelines for bioinformatics of single-cell sequencing data analysis in Alzheimer's disease: review, recommendation, implementation and application. *Mol Neurodegener* **17**, 17 (2022). <https://doi.org:10.1186/s13024-022-00517-z>